# A physical wiring diagram for the human immune system

Jarrod Shilts[1✉], Yannik Severin[2], Francis Galaway[1], Nicole Müller-Sienerth[1], Zheng-Shan Chong[1], Sophie Pritchard[3], Sarah Teichmann[3], Roser Vento-Tormo[3], Berend Snijder[2] & Gavin J. Wright[1,4✉]

The human immune system is composed of a distributed network of cells circulating throughout the body, which must dynamically form physical associations and communicate using interactions between their cell-surface proteomes[1]. Despite their therapeutic potential[2], our map of these surface interactions remains incomplete[3,4]. Here, using a high-throughput surface receptor screening method, we systematically mapped the direct protein interactions across a recombinant library that encompasses most of the surface proteins that are detectable on human leukocytes. We independently validated and determined the biophysical parameters of each novel interaction, resulting in a high-confidence and quantitative view of the receptor wiring that connects human immune cells. By integrating our interactome with expression data, we identified trends in the dynamics of immune interactions and constructed a reductionist mathematical model that predicts cellular connectivity from basic principles. We also developed an interactive multi-tissue single-cell atlas that infers immune interactions throughout the body, revealing potential functional contexts for new interactions and hubs in multicellular networks. Finally, we combined targeted protein stimulation of human leukocytes with multiplex high-content microscopy to link our receptor interactions to functional roles, in terms of both modulating immune responses and maintaining normal patterns of intercellular associations. Together, our work provides a systematic perspective on the intercellular wiring of the human immune system that extends from systems-level principles of immune cell connectivity down to mechanistic characterization of individual receptors, which could offer opportunities for therapeutic intervention.

The human immune system must maintain the same coordination and cohesion as the body's other homeostatic organ systems despite being composed of highly migratory and circulating cell types that are distributed throughout the body. Diverse arrays of cell-surface proteins organize immune cells into interconnected cellular communities, linking cells through physical interactions that act both for signalling communication and for structural adhesion[5]. The immune system has been described from one perspective as carefully coordinated networks of cell types[6,7], where by extension it is these physical linkages that hold the network together[3]. Consequently, immune receptors regulate virtually all stages of cellular activation and are appreciated as critical mediators of a variety of homeostatic and pathological processes, which range from tumour surveillance, to autoimmunity, to infection control. For these reasons, along with their accessibility to systemically administered medicines, immune surface proteins and their interactions are particularly attractive therapeutic targets[2,8].

Although the interaction networks that involve secreted proteins have already been systematically catalogued[9,10], in the immune system and more generally across existing protein interaction databases, there remains a substantial under-representation of the interactions between cell-surface proteins[11,12]. Specialized methods have been developed that tackle individual challenges stemming from membrane-embedded surface proteins, such as their typically weak binding affinities[13,14] and the low tractability of these proteins for many classic biochemical approaches[15,16]. These methods, however, generally lack the throughput to systematically characterize whole cell-surface proteomes, or have only had success for specific protein families rather than the full diverse spectrum of surface protein topologies and complexes[17,18]. Thus, how complete our understanding is of extracellular immune receptor interactions has remained unknown. Moreover, many immune receptors of clinical importance have been left as 'orphans', with their physiological ligands undiscovered despite in some cases decades of study[19–22]. Without a systematic picture of the physical interactions that link immune cells, any efforts at present to generate truly systems-level views of immune function will remain patchwork at best.

[1]Cell Surface Signalling Laboratory, Wellcome Sanger Institute, Cambridge, UK. [2]Institute of Molecular Systems Biology, ETH Zurich, Zurich, Switzerland. [3]Cellular Genetics Programme, Wellcome Sanger Institute, Cambridge, UK. [4]Department of Biology, Hull York Medical School, York Biomedical Research Institute, University of York, York, UK. ✉e-mail: js2320@cantab.ac.uk; gavin.wright@york.ac.uk

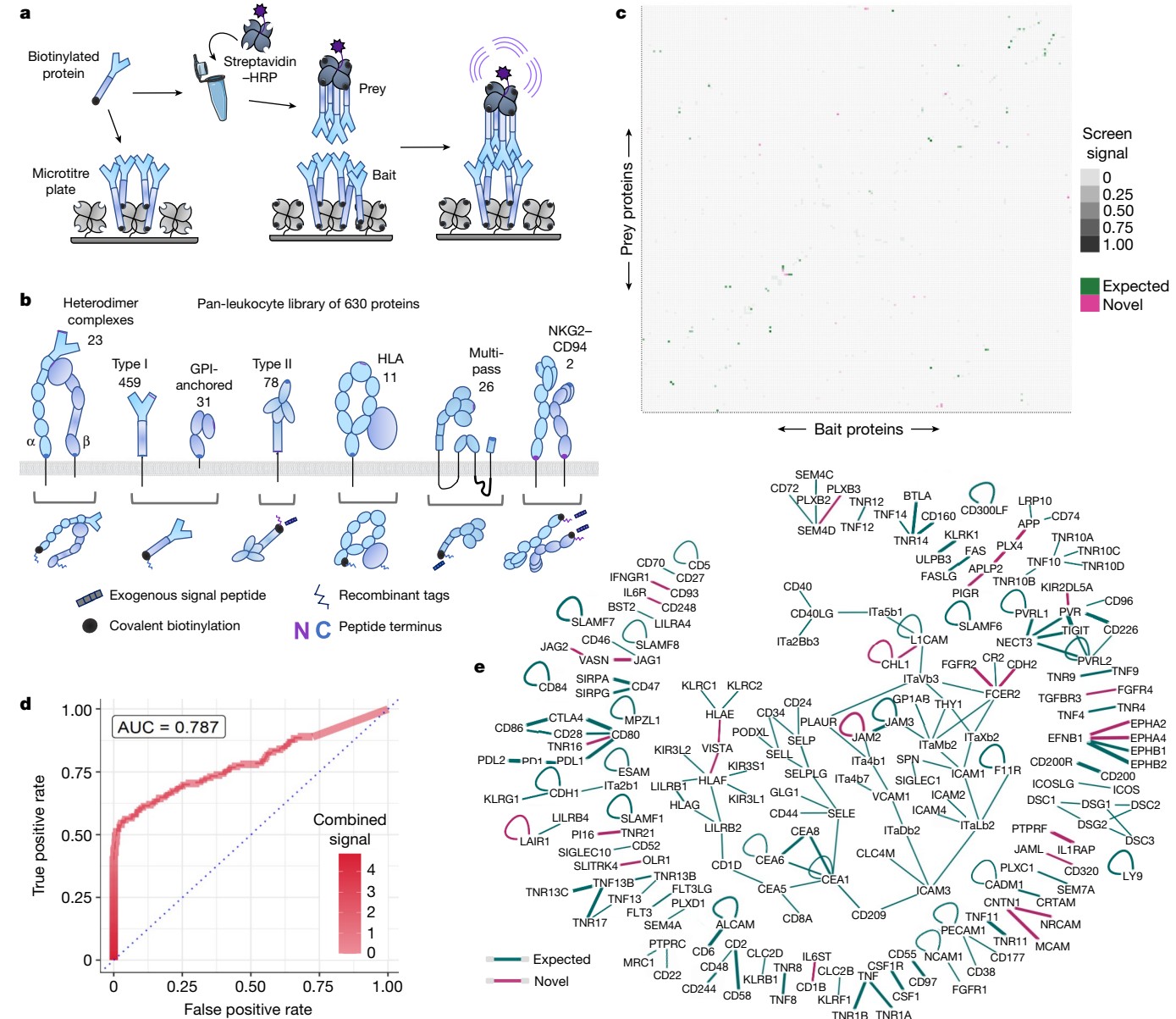

**Fig. 1 | A leukocyte receptor network by systematic protein interaction mapping. a**, SAVEXIS enables efficient and high-throughput screening for protein binding interactions between recombinant extracellular domains. **b**, Schematic showing the diverse structural architectures of leukocyte surface proteins within the pan-leukocyte library of 630 proteins. The number of proteins from each class is noted above, and the recombinant expression strategy is illustrated below. **c**, Summarized matrix of protein–protein pairs for immune receptors with interactions either identified by screening or previously reported in the literature. The average signal intensity for a given bait–prey measurement orientation across the primary and secondary screens is indicated by the shaded intensity, and the colour indicates which interactions are novel. **d**, Screening successfully finds most previously reported interactions with minimal false positives. Receiver operating characteristic (ROC) curve for average measurements of protein–protein pairs against reference sets of expected positive and randomized negative interactions. AUC, area under the curve. **e**, Organized interaction network of immune receptor interactions. The colour indicates which interactions are novel, and the line thickness is proportional to the magnitude of evidence from the screening measurements.

## Building a surface protein interactome

To enable a systematic survey of the surface protein interactions between immune cells at scales that approach the size of whole cell-surface proteomes, we first developed an optimized method for testing binary interactions of all possible pairings of recombinant surface proteins (Fig. 1a). Our method, the scalable arrayed multi-valent extracellular interaction screen (SAVEXIS), simultaneously addresses several key limitations of previous methods to make it possible to screen hundreds of thousands of interactions while consuming minute amounts of protein (Extended Data Fig. 1). By exploiting multimerization around streptavidin, both immobilized 'baits' and reporter-linked 'preys' for screening can be produced from a single construct instead of two, and the design is attuned for detecting even low-affinity interactions across the range of structural classes that cell-surface proteins span.

We assembled a detailed library that encompassed the full ectodomains of cell-surface proteins detectable in a previous high-resolution proteomics survey of peripheral immune cells[1] plus all CD-numbered proteins that were compatible with our recombinant expression platform (Supplementary Table 1). The design of each expression

construct was tailored to accommodate the structural class of the protein, with six different bespoke designs for functionally expressing different receptor topologies and multicomponent complexes (Fig. 1b). This library of 630 different proteins or protein complexes (such as all known integrin combinations) was expressed in human cells, purified and quality-checked (Extended Data Fig. 2a,b and Supplementary Fig. 1). We tested all possible protein pairings in both bait–prey orientations for every protein expressed (Extended Data Fig. 2c and Supplementary Table 2). Positive interactions identified from this comprehensive primary screen were then re-tested in a secondary screen with independent protein preparations from a separate cell strain, consisting of an all-versus-all matrix of 187 proteins (Extended Data Fig. 2d) to give a final matrix of reproducible interactions (Fig. 1c). In benchmarks against a hand-curation of the published literature (Supplementary Table 3), our screen independently captured a majority of all previously reported interactions at a false positive rate below 1 in 10,000 (Fig. 1d and Extended Data Fig. 3a,b). We identified 28 new interactions that were not in our literature curation, expanding the total number of known high-confidence interactions in the human immune system by 20% (Fig. 1e). Notably, these include endogenous non-tumour ligands for the previously orphan immune checkpoint receptor VISTA, comprising the non-classical MHC molecules HLA-E and HLA-F. As these measurements constitute a highly systematic view of surface interactions for an exemplar cell system, our data also suggest answers to the general properties of surface receptor networks. In the set of immune receptors, 57% of binding pairs are unique, without either protein having another binding partner. Exclusivity is particularly common among proteins that are generally considered to have primarily signalling roles, whereas the largest interconnected group features integrins and other adhesion molecules.

## Quantifying total receptor engagement

To validate these discoveries, every interaction was assessed by orthogonal approaches. First, we tested protein binding to the complementary receptor when displayed on a human cell surface following transfection with a cDNA that encodes the receptor (Fig. 2a). This was followed by two rounds of surface plasmon resonance (SPR) to characterize direct binding (Fig. 2b and Supplementary Fig. 2). All of the top-ranked 28 interactions were supported by at least one additional method (Extended Data Fig. 3c). We realized that we could combine the binding affinities measured in our SPR data with measurements that we methodically extracted from the literature to assemble not only a systematic physical interaction network, but also a uniquely quantitative one (Fig. 2c and Extended Data Fig. 4). We integrated this quantitative receptor interaction network with proteomics expression in leukocytes to gain insight into the patterns of binding kinetics across the immune system (Fig. 2d and Supplementary Table 4). For example, in their contacts with antigen-presenting cells, we found that circulating T lymphocytes show a subtle preference for higher-affinity receptors when pairing with B cells compared to dendritic cells (Extended Data Fig. 5a). We also established that the overall distribution of surface interactions has affinities centred in the low micromolar range, although with a long tail of higher-affinity interactions (Extended Data Fig. 5b). In a test of previous theoretical predictions[23,24] we found that higher expression levels do negatively correlate with binding strength, although only weakly (Extended Data Fig. 5c,d). Of note, we also found that immune activation is accompanied by a broad transition in cellular interaction strengths. Higher-affinity interactions predominate in an inflamed state, and these are replaced with more transient interactions in a resting state, possibly to support the need for more dynamic responses when sensing potential threats (Fig. 2e). We confirmed this 'affinity switch' in human leukocyte receptor preferences using an independent transcriptomic dataset (Extended Data Fig. 6).

Our quantitative wiring diagram, if truly systematic to the point of approaching completeness, should make it possible to derive a reductionist model that explains how circulating immune cells associate with each other solely from receptor-binding mechanisms and physics-based formulas. We built a coarse yet principled mathematical model that integrates quantitative proteomics expression, binding kinetics and published cell parameters to summarize the contributions of individual protein interactions to a given cellular interaction (Fig. 2f and Supplementary Equations). Using equations based on the law of mass action, the model then computes how the overall probability of binding between two cell types emerges from the distinct spectrum of cell-surface receptors that connect them (Extended Data Fig. 7). Although greatly simplistic, our model could still infer the relative frequencies at which human immune cells physically interact with sufficient precision to be consistent with published empirical measurements[25] (Fig. 2g and Extended Data Fig. 7d).

## Integrating multicellular networks

Although our library used circulating immune cells as a source, the immune system traverses a broad repertoire of organs, each of which may be key to understanding the biological role of an interaction. We therefore sought to contextualize the interactions of our network by creating an interactive atlas that charts where these receptor and ligand pairs have been detected across single-cell expression datasets of human tissues (Fig. 3a). Our integrated atlas is available at https://www.sanger.ac.uk/tool/immune-interaction/immune-interaction and allows multiple kinds of analysis, which range from summarizing the overall cellular connectivity of different tissue immune populations to inferring which cell–cell pairs are capable of carrying out a particular receptor interaction (Fig. 3b and Extended Data Fig. 8a).

Through this systematic multi-organ atlas, we could determine whether immune receptor interactions proceed through shared structures or are distinct between tissues. We found a recurring motif in which myeloid-lineage cells act as hubs across several cellular interaction networks. Quantified across multiple primary and secondary lymphoid tissues, myeloid cells have consistently higher network centrality scores (Fig. 3c), despite expressing similar numbers of surface ligands to other cell types (Extended Data Fig. 8b). This suggests that resident myeloid cells may adapt their receptor repertoire to serve as central integrators of local interactions in their tissue niche. Considering the breadth of pathological conditions that exhibit immune dysregulation, we reasoned that these same integrated approaches could also inform how the physical interactions that we catalogued between immune cells may change in disease. We incorporated paired diseased and reference samples where available in our atlas, which can be used to generate hypotheses on which interactions (cellular or molecular) may differentially appear in diseased states. For example, we could see phagocyte populations shifting a large fraction of their total cellular contacts within the tumour microenvironment of kidney samples (Extended Data Fig. 8c), including upregulation of APLP2 and APP ligands that we characterized (Supplementary Table 6). Our novel receptor interactions could also be integrated with known signalling pathways to infer cellular communication pathways that appear differentially active in diseased states[26], which implicated the newly discovered JAML interaction in potentially regulating anti-tumour immunity (Extended Data Fig. 8d). To investigate whether the interactions that we discovered in our network manifest themselves under physiological settings in the human body, we finally examined the spatial colocalization of receptors and ligands in a lymph-node spatial transcriptomics dataset. Although this reflects only a snapshot of these dynamic cellular populations, both our set of new interactions and the previously reported interactions were significantly more likely to be spatially proximal than randomized networks (Fig. 3d). In follow-up experiments, we could verify that novel interacting pairs we identified, such as JAG1–VASN, are

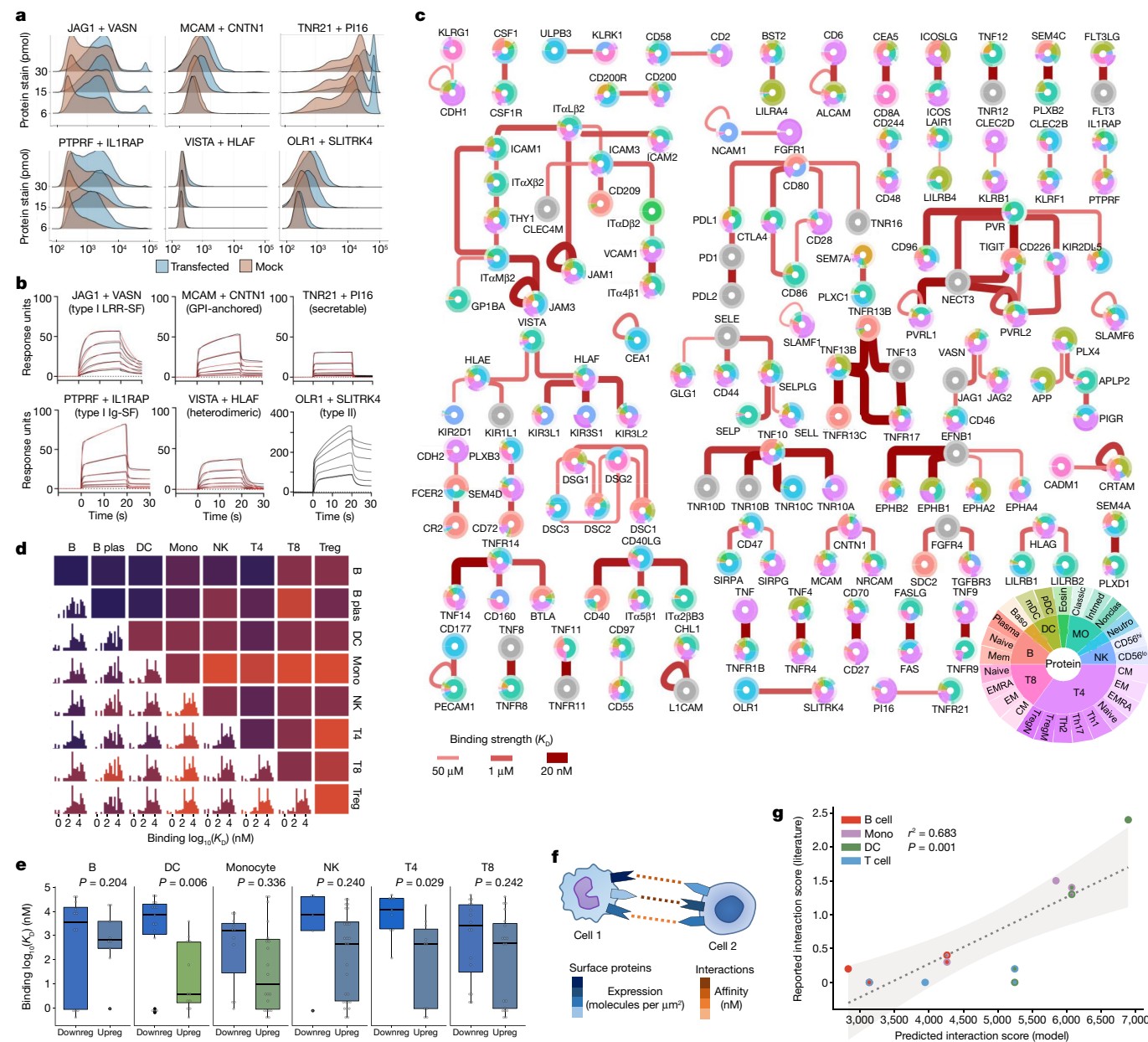

**Fig. 2 | Validating and assembling a quantitative immune interactome.**
**a**, Novel receptor interactions are detectable on the surfaces of live human cells. For six examples that encompass different architectural classes, flow cytometry traces are shown for the binding of fluorescent-conjugated protein to HEK293 cells overexpressing its identified counter-receptor (blue) or control cells (red). **b**, SPR substantiates and quantifies the binding of novel leukocyte receptor–ligand pairs. For the same six example protein pairs, sensorgram data (black) are shown with Langmuir model fitting curves overlaid (red) for all interactions for which a robust fit could be calculated. Ig-SF, immunoglobulin superfamily; LRR-SF, leucine rich repeat superfamily. **c**, The quantitative interactome of immune cell-surface proteins. Proteins are shown as circular charts indicating the proportion of expression in each leukocyte population. Binding affinity between proteins is indicated by the size and intensity of red edges (expressed in terms of the binding dissociation constant ($K_D$), where smaller values reflect stronger binding). Abbreviations for cell type names are defined in Supplementary Table 5. **d**, Immune cell subsets use related but varying distributions of binding affinities

when communicating with other cell types. For each pairing of two cells, a histogram of inferred interactions is shown alongside a colour shade that indicates the average affinity. **e**, Inflammatory activation broadly reconfigures receptors towards those with less-transient binding kinetics. After differential expression testing of surface proteins between activated and stimulated leukocytes ($n = 4$ samples per condition), the binding affinities of interactions involving downregulated (downreg) receptors are compared to the binding affinities of upregulated (upreg) receptors. Data are shown as Tukey box plots with Holm-corrected $P$ values from a two-sided Welch's $t$-test. **f**, Intercellular connectivity can be mathematically predicted by integrating protein expression, binding affinity and cell parameters using physics-based equations. A detailed description of the model can be found in the Supplementary Equations. **g**, Model predictions for baseline rates of immune cell association agree with published data measuring in vitro immune cell association. Each data point has two colours that correspond to the two physically interacting cell types. Shading depicts the 95% compatibility interval for the least-squares linear regression fit.

distributed in bordering regions of the immune-cell-rich compartments of human lymph nodes (Fig. 3e). This confirms that these interactions have the potential to occur between immune cells in vivo (Extended Data Fig. 9a).

## Assigning functions to binding targets

We finally asked whether the reagents and receptor interactions that we characterized could have a potential clinical use in modulating

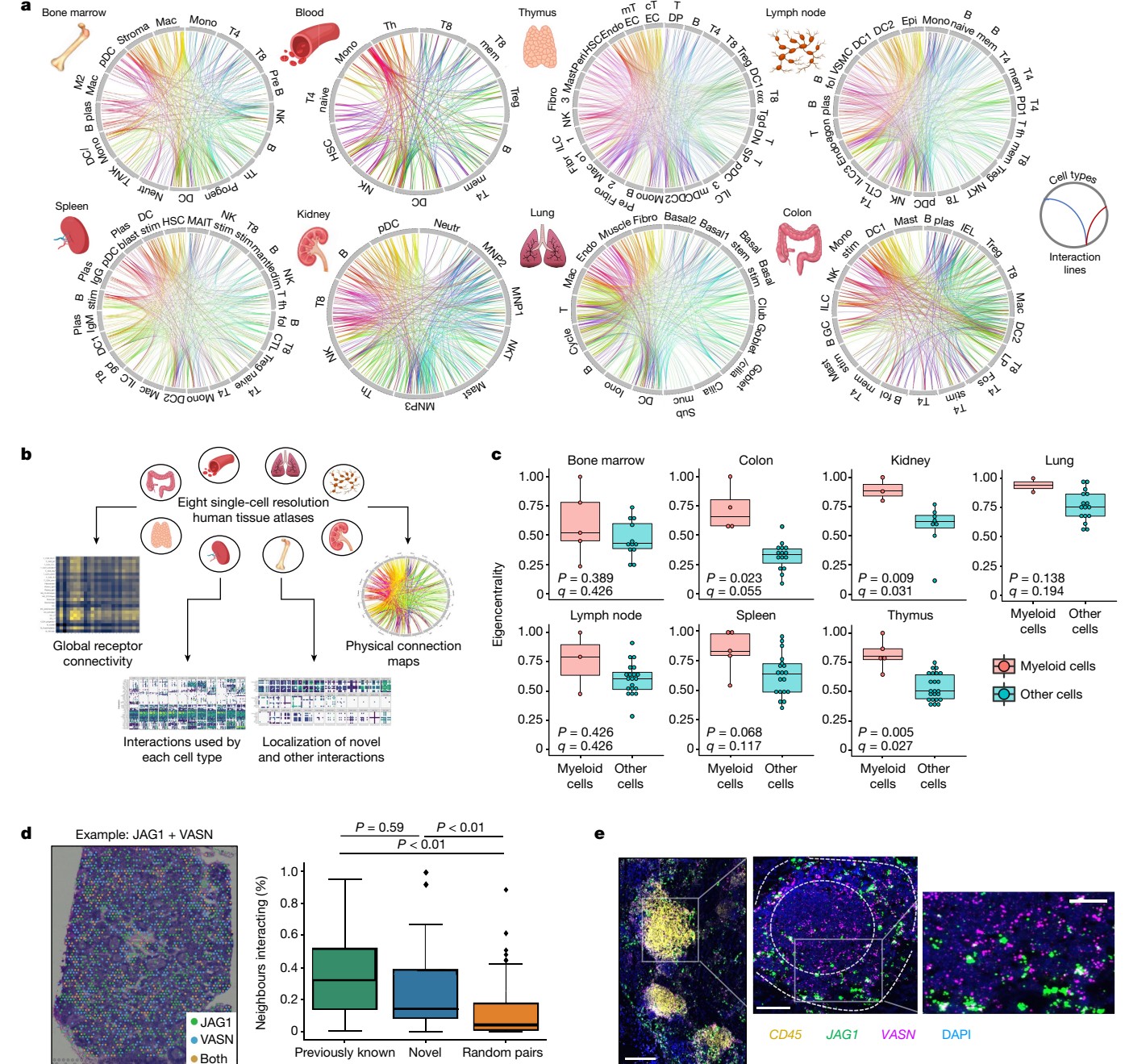

**Fig. 3 | An interactive atlas of immune cell connections across the human body. a**, Systematic integration of single-cell datasets to map cellular connectivity across tissues with substantial immune populations. Cell types are positioned around each circle, with each position along it marking a cell-surface protein. Linkages formed by physically interacting surface proteins between cell types are marked by curved lines, coloured by interaction identity. Full abbreviations are listed in Supplementary Table 5. **b**, Functionalities available through our interactive atlas of physical immune interactions (https://www. sanger.ac.uk/tool/immune-interaction/immune-interaction). **c**, Myeloid cells act as interaction hubs in immune tissues. Eigenvector centrality metrics of myeloid cells compared to all other populations after converting the total interaction count for all cell–cell pairs into a weighted undirected graph. Data are shown as Tukey box plots with Benjamini–Hochberg $P$ values calculated from a two-sided Welch's $t$-test. **d**, Spatial transcriptomics of a human lymph node confirms that

our identified interaction partners are physically colocalized in situ. An example data point of a tissue section analysed for the JAG1 + VASN interaction is shown. The percentage of measured spots in which the expression of one protein of an interacting pair is spatially connected to the other protein of the pair is compared for previously reported interactions (green), novel interactions (blue) and a negative control of the same proteins with interaction links randomly permuted (yellow) ($n = 100$). Data are shown as Tukey box plots with $P$ values calculated from a Tukey's honest significance test. **e**, Single-molecule RNA hybridization on human lymph nodes defines regions in which newly identified interaction partners are expressed in spatially bordering cells. A single lymphoid follicle enriched in *CD45*+ leukocytes is magnified (left), showing the zonation of *JAG1*- and *VASN*-expressing cells into the corona and the germinal centre, respectively (middle). An inset (right) highlights a region of bordering cells expressing each marker. Scale bars, 200 μm (left); 100 μm (middle); 50 μm (right).

the immune system. Soluble recombinant proteins with activity to bind specific immune cell receptors have shown promise as immuno-therapies[27,28]. As a first step towards potential therapeutic applications

and functional classifications on modes of action, we performed high-throughput cellular phenotyping assays on isolated human immune cells that were treated with recombinant forms of the proteins

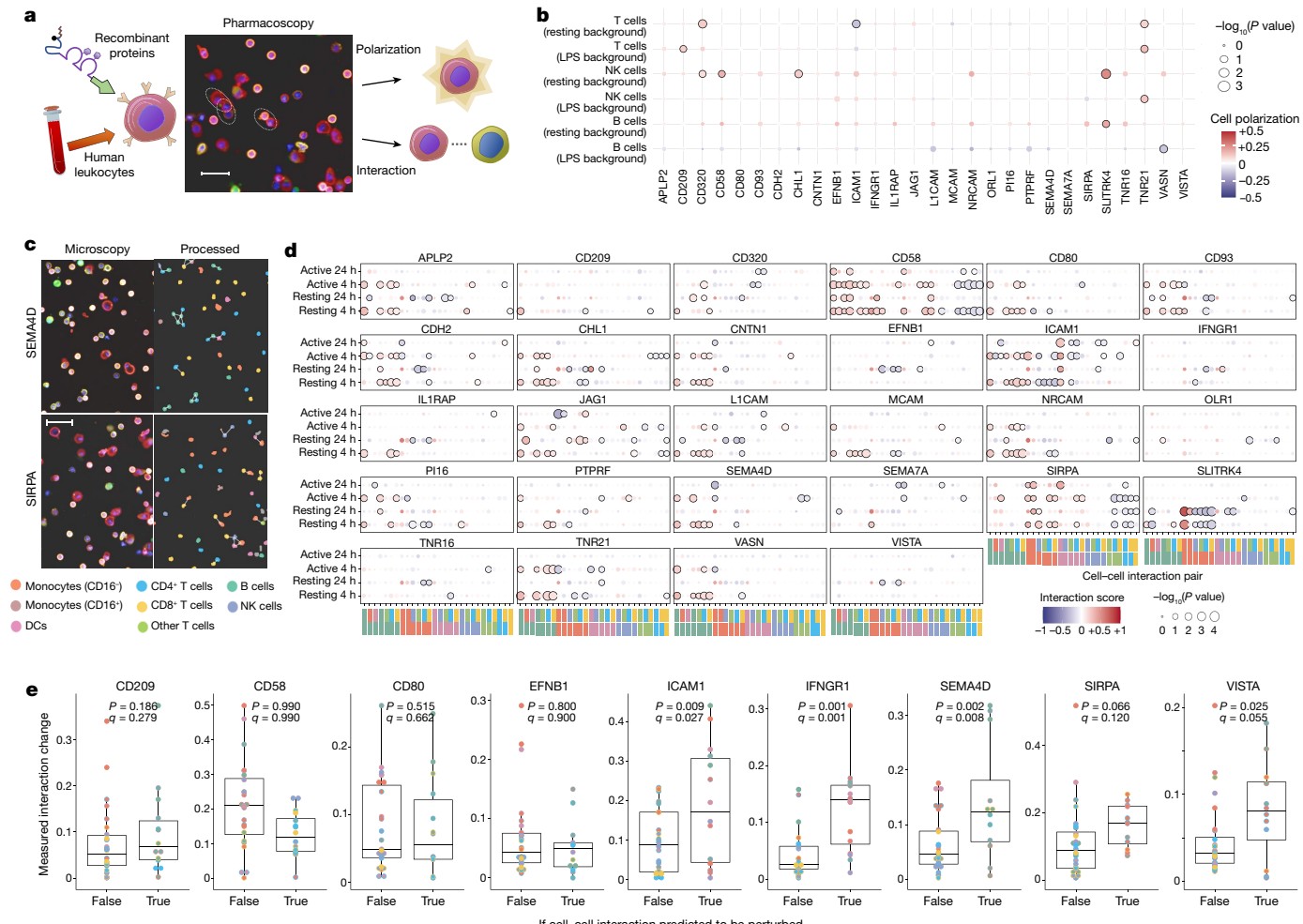

**Fig. 4 | Multiplex leukocyte assays identify functional pathways for receptor proteins. a**, High-content microscopy set-up for perturbing human peripheral blood mononuclear cells (PBMCs) with recombinant proteins and measuring changes in cellular activation and connectivity. Scale bar, 30 μm. **b**, Proteins with identified receptor interactions elicit responses on lymphocyte action. Polarization of lymphocyte populations in resting and weakly activating background conditions (*y* axis) after addition of soluble protein extracellular domains (*x* axis). Stimulation (red) or inhibition (blue) relative to control is shown of a cell polarization marker of lymphocyte activation. Phenotypes that have *P* values below the adjusted significance threshold are outlined in bold. *n* = 10 samples. **c**, Interacting cellular communities can be extracted from high-content imaging data. Representative microscopy fields (left) and computed physical cell contacts (right, white lines) are depicted for leukocytes perturbed with recombinant SEMA4D and SIRPA as

examples. Scale bar, 30 μm. **d**, Rewiring of cellular interactions by perturbing receptor pathways. Measured changes in cell–cell interactions (*x* axis) induced by recombinant proteins (panels) across measurement time points and background conditions (*y* axis). The same colour scale as in **c** is used to identify cell pairs along the *x* axis. *n* = 10 samples. **e**, Observed interaction changes conform to mathematical model predictions. Average magnitudes of cell–cell interaction changes (*y* axis) after the addition of recombinant proteins are compared for cell pairs predicted by the model to be likely to change after perturbing that surface protein ('true') and those predicted not to change ('false'). Each panel considers a different recombinant protein added in the experiment and the corresponding model predictions for that same protein. The two colours for each data point depict the identity of the cell pair according to the colour scale in **c**. *n* = 10 samples for the experimental data.

identified in our molecular wiring map. To reflect the wide array of interactions and leukocyte subtypes included in our investigation, we adapted an approach for measuring leukocyte activation and cellular interaction phenotypes by high-content microscopy[29], which captures all major cell populations in a single multiplex experiment (Extended Data Fig. 9b,c). We incubated pools of human leukocytes in the presence of purified proteins from our recombinant surface receptor library, imaging at 4 and 24 h to measure changes in both cell–cell interactions and the proportions of activated cells elicited by each protein (Fig. 4a and Supplementary Fig. 3). Because the function of interactions may not be revealed unless in the proper context, we measured resting leukocytes and those stimulated by low levels of lipopolysaccharide (LPS). Proteins from our set of novel interactors elicited diverse responses, including generalized T cell activation by the formerly orphan[30] TNFRSF21, as well as the adhesion proteins CHL1 and

CD320 (along with CD58 as previously described[31]) facilitating natural killer (NK) cell activation (Fig. 4b and Extended Data Fig. 10a). These phenotypes were generally consistent with the cell-type expression of each protein's newly identified receptor partners, suggesting direct effects as opposed to indirect mechanisms mediated by an intermediate leukocyte type within the mixed pools of cells (Extended Data Fig. 10b).

The perturbation experiments also provided high-dimensional data on changes in cellular connectivity triggered by infusing soluble receptor-binding proteins (Fig. 4c). The landscape of immune interactions across these conditions provides a rare view into the functional roles of previously described and novel immune surface proteins. This includes large-scale shifts away from cellular contacts between T cell populations from CD58 along with increased monocyte contacts from SIRPA (consistent with our previous biological understanding[32,33]); an inhibition of interactions between monocyte–lymphocyte pairs from

SLITRK4; and several ligands that we implicated in adhesion such as NRCAM, CD93 and CDH2 converging on facilitating early B cell interactions (Fig. 4d). In line with results reported previously for changes in the interactions of leukocytes after drug administration[25,29], infusing our recombinant proteins triggered changes that generally fit within a distinct set of modules of action (Extended Data Fig. 10c). Our in silico model of immune connectivity allowed us to rationalize which molecular changes would have been likely to lead to the observed phenotypes. We compared these experimentally measured perturbations to mathematical predictions as to which cell–cell pairs would have the greatest perturbations. Although as before there were some gaps in the accuracy of our model for particular conditions, nevertheless, for most stimuli we observed significantly greater magnitudes of connectivity perturbations in our experimental data for cell pairs that the model predicted would be most perturbed (Fig. 4e).

## Discussion

The immune system is distinctive for being a distributed system. It is not fixed to a single localized organ in the body, but rather is made up of numerous specialized cell types that must adaptably organize their intercellular connections to respond to pathogens and other threats wherever they may appear. We provide a systematic and quantitative view of the cell-surface proteins that enable immune cells to dynamically wire their interactions. The receptor interactions that we report in our network each merit further individualized study to characterize their full roles in health and disease. Of particular note are our discovery of HLA-E and HLA-F (but probably not HLA-G) as endogenous non-tumour ligands for the immune checkpoint receptor VISTA; the ability of vasorin to act as a receptor for Jagged ligands; and immunoglobulin family receptors binding members of the amyloid precursor protein family. Notably, members of this same family of non-classical MHC class I molecules that includes HLA-E and HLA-F have previously been identified as key ligands for maintaining innate immune quiescence[34], which our findings extend to raise the possibility that they may similarly act as regulators of adaptive immune quiescence through VISTA[35]. Our functional screening on blood immune cells further points to pathways worth greater consideration; for example, a role of SLITRK4 in lymphocyte responses. While we were preparing this study, independent groups have provided supporting evidence for several of the interactions that we characterized here[17,18,36], including PVR as the ligand for the formerly orphan KIR2DL5A and CD146 as an adhesive ligand for CNTN1.

Because our physical wiring diagram encapsulates the diversity of surface protein architectures found across all major subsets of leukocytes, it can be integrated with publicly available expression data both qualitatively and quantitatively. The physical interaction landscape that we provide in our single-cell expression atlas offers an interactive platform for deriving insights out of our systematic receptor network. Although other studies have provided useful views of interactions that are differentially regulated in particular cell types[37,38], we provide a systematized catalogue of all biological contexts in which an interaction is inferred to be possible, including complementary views directed at particular cell subsets or receptor proteins. This provides material for generating hypotheses about multicellular immune circuits. These same integrated datasets can resolve broader questions in the field. For example, we have shown that immune cell activation is accompanied by broad shifts in the affinity profiles of cell-surface interactions, with rapid kinetics predominating during the early stages of immune responses (matching previous conceptual models for optimal antigen sampling[39] and propagating signals to other cells[40,41]), which switch to higher-affinity contacts in an inflamed state to match the changing demands of a forceful inflammatory response. This raises parallels with findings of disease-associated variants that modulate the balance of adhesive receptor affinity[42–44]. We could also show that the previously

theorized link between receptor affinity and abundance—in which low-affinity receptors are compensated for by their high abundance[24]—may indeed exist, but appears not to be sufficient to account for why certain receptors have evolved the binding kinetic rates we observe.

Out of this integrative approach, we could construct a proof-of-concept mathematical model that predicts the behaviour of collections of leukocytes from first principles. Although considerable study has been devoted to particular specialized cell-to-cell contacts such as the immunological synapse[45], the overall connectivity of immune cells and their dynamic approaches and disengagements has been neglected. Our core model opens up considerable scope for dissecting in a principled way the mechanisms by which immune cells directly associate. In addition, the discrepancies between the model compared to experiments offer opportunities for refining our mechanistic understanding of cell–cell interactions by comparing how elaborations to this core model further improve prediction accuracy.

More broadly, the integrated approaches that we pioneered here for disentangling the immune system provide a framework for future systematic investigations. Using our high-throughput biochemical method for interaction screening (SAVEXIS) and the strategies that we describe here to characterize interactions by combinations of multiplex cellular assays and genomics datasets, a range of other cellular communities in the human body could similarly be quantitatively mapped. To our knowledge, our study is among the first to systematically map and model how the collective actions of individual receptor molecules through physical laws could explain and predict cellular connectivity on a scale as large as the circulating immune system. Our analysis and the methods that we developed provide a template for future studies looking at physical cell wiring networks in detail. From these combined approaches, we may finally begin to disentangle cellular circuits in immunity and beyond, bridging from individual protein molecules to multicellular behaviour.

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

# Methods

## Culture and transfection of HEK293 cells

Human embryonic kidney 293 (HEK293) cells were maintained in suspension culture with Freestyle medium (Gibco 12338018) supplemented by 1% heat-inactivated fetal bovine serum (Sigma F2442) at 37 °C, 5% carbon dioxide and 70% humidity as previously described[46]. For all transient transfections, cells were seeded 24 h before transfection at a density of $2.5 \times 10^5$ cells per ml, then transfected with 0.5 µg DNA per ml cells as previously described[46]. Protein expression was done in vented conical flasks (Corning) ranging from 30 ml cells for the primary screen to 100 ml cells for other applications. Transfections for cell-binding assays were typically done in a volume of 1 ml cells in 96-well deep plates (Corning 3960). Where expressed proteins were to be enzymatically biotinylated, the culture medium was supplemented with D-biotin (Sigma B4501) to 100 µM and a plasmid encoding secreted BirA biotin ligase was co-transfected at 30 ng per ml cells[47]. Transfections for protein complexes such as integrins were done using equimolar ratios of plasmids encoding both chains, except for HLA-related complexes for which only 10% of the total plasmid encoded $\beta_2$ microglobulin. For most experiments, the EBNA1-containing HEK293-E cell line was used, except for proteins produced for the secondary binding screen, which were expressed in the serum-free modified HEK293-6E line[48]. Both cell lines were provided by Y. Durocher. Cell lines were regularly tested for mycoplasma (Surrey Diagnostics) and found to be negative.

## Purification of recombinant proteins

After transfected cells were incubated for times ranging from 90 to 120 h, they were centrifuged at 2,000$g$ for 20 min. Supernatants were filtered through 0.22-µm filters and purified by nickel-ion affinity chromatography, with the exact procedure differing slightly by the intended downstream experiment. Proteins for high-throughput interaction screening were purified using His MultiTrap plates (GE Healthcare 28-4009-89) via a previously described 96-position pneumatic press[49]. Each supernatant was supplemented to a final imidazole concentration of 16 mM and an NaCl concentration of 250 mM before purification, and the plates were prepared by rinsing with 500 µl pure water and washing twice with 500 µl 20 mM imidazole phosphate buffer following the manufacturer's instructions. Once all samples had loaded, plates were washed again twice with 500 µl 20 mM imidazole phosphate buffer and eluted with 200 µl 200 mM imidazole phosphate buffer. Proteins for the secondary interaction screen and SPR were purified using an ÄKTA Pure automated chromatography instrument (GE Healthcare) loaded with 1 ml HisTrap HP columns (GE Healthcare) as previously described[50]. Proteins for cell-based experiments were manually purified with HisPur Ni-NTA resin (Thermo Fisher Scientific 88221). Resins were pre-washed twice in 20 mM imidazole phosphate buffer, then resuspended in supernatant supplemented with 1 mM imidazole. The slurry was incubated rotating overnight at 4 °C, before two further washes in 20 mM imidazole phosphate buffer and finally eluted in 200 mM imidazole phosphate buffer. After initial purifications, proteins intended for use in immune cell assays were dialysed against phosphate-buffered saline overnight at 4 °C using a 12–14-kDa molecular mass cut-off D-tube dialyser (Millipore 71505). Proteins intended for use as SPR analytes were further resolved by gel filtration with an ÄKTA Express machine on a Superdex 200 Increase 10/300 GL size-exclusion column (GE Healthcare) to remove any aggregated protein that may interfere with kinetic measurements. To avoid precipitation on the column, analyte samples were pre-dialysed in HBS-EP buffer (GE Healthcare BR100669). All proteins were stored at 4 °C until use.

## Protein gel electrophoresis

Samples were denatured in lithium dodecyl sulfate (NuPAGE NP0007) and dithiothreitol (NuPAGE NP0004) to a total volume of 10 µl before heating at 70 °C for 10 min. Samples were resolved on Bis-Tris 4–12% polyacrylamide gradient SDS−PAGE gels (NuPAGE NP0329) at 200 V for 50 min in MOPS buffer (NuPAGE NP0001). As a marker, 4 µl of pre-stained SeeBlue Plus2 protein standard (Invitrogen LC5925) was added to each gel. For Coomassie staining of total protein, gels were briefly rinsed in water then incubated in brilliant blue G-250 dye (Abcam ab119211) over 24–36 h. Before imaging under visible light, gels were briefly rinsed twice in water. For immunoblots, proteins were transferred to a methanol-activated PVDF membrane (GE Healthcare 10600029) in transfer buffer (NuPAGE NP0006) over 60 min at 300 mA. Blots were briefly washed in HEPES buffered saline (HBS) with 0.1% (v/v) Tween-20 (Sigma P2287), blocked with 2% (m/v) bovine serum albumin (BSA; Sigma A9647), stained with 1:5,000 anti-His tag C terminus antibody conjugated to HRP (Invitrogen R931-25, clone 3D5) for 16 h at 4 °C, washed an additional three times, and exposed to chemiluminescent substrate (Thermo Fisher Scientific PI34577). Images were developed on photographic film (GE Healthcare 28906835) for 45–90 s. To automate our comparisons of the molecular masses observed by electrophoresis to the computationally expected masses, we made a custom Python script that translates each expression construct using the Biopython library and identifies post-translational processing sites through automated queries to Uniprot.

## Bradford concentration normalization

Standard curves of pure BSA (Pierce 23209) spanning from 1,200 to 7 ng µl$^{-1}$ were prepared in clear flat-bottom 96-well plates (Thermo Fisher Scientific 11349163). In general, 20 µl of each standard curve and purified protein sample was added to wells of the plate. All wells had 250 µl of Bradford reagent (Pierce 23236) added and were gently agitated for 60 s, then incubated for 30 min before measuring the absorbance at 595 nm with a Tecan Spark plate reader. All raw absorbance signals had the average background of buffer-only controls subtracted before further processing. The standard curve data were fit with fifth-order polynomials with the intercept fixed at (0,0). The curves were manually inspected for their fits and residuals before using the polynomial equation to calculate the concentration of each sample. These values were converted to molar concentrations by using the molecular masses calculated through the automated Python script, except excluding the added mass of glycan modifications.

## ELISA tetramerization assays

Both as a check on Bradford-based molarity calculations and for determining how to form cell-staining tetramers, biotinylated recombinant proteins were quantified by a competitive ELISA. Dilution series of the target proteins were made in 96-well plates, then incubated with a known constant quantity of a conjugated avidin (SAV−HRP for binding assay calculations, SAV−PE for cell-binding cytometry reagents and neutravidin for immune cell perturbation assays). After at least 45 min at 20 °C, these pre-incubated samples were transferred to a BSA-blocked streptavidin-coated 96-well plate (Greiner 655990). After another 45 min or longer incubation, plates were washed three times with HBS and 0.1% (v/v) Tween-20. Primary antibody, typically 1 µg ml$^{-1}$ of an OX68 monoclonal against the protein epitope tags[51], was prepared in 100 µl 2% (m/v) BSA in HBS, incubated for 60 min and then washed three times. Secondary antibody of goat anti-mouse IgG conjugated to alkaline phosphatase (Sigma A3562) was similarly incubated at 0.2 µg ml$^{-1}$ for 30 min before another three washes were performed. Finally, 60 µl of 1.5 mg ml$^{-1}$ $p$-nitrophenyl phosphate (Sigma P4744) in diethanolamine buffer was added and the absorbance at 405 nm was measured after 30–60 min by a Tecan Spark plate reader.

## High-throughput SAVEXIS screening

The following is the procedure for the final SAVEXIS technique used for screening, as established by extensive optimization of each assay parameter following several iterations during its development. Streptavidin-coated 384-well plates (Greiner 781990) were washed once

briefly in 80 µl HBS with 0.1% Tween-20 (HBS-T), then blocked in 2% (m/v) BSA in HBS (10 mM HEPES, 1 mM MgCl₂, 2 mM CaCl₂, 5 mM KCl, 140 mM NaCl, pH 7.4) for at least 30 min at 20 °C. Purified bait proteins were diluted in 2% BSA in HBS such that each 50-µl well would contain 100 femtomoles of biotinylated protein. All screening plates were arrayed from a set of stock source plates using a custom-programmed Biomek FXp robot (Beckman Coulter) and manually inspected to correct any wells missed. Baits were left to be captured for 16 h at 4 °C. In parallel, multimeric preys were assembled by mixing 6.25 fmol of streptavidin–HRP (Pierce 21130) with the calculated stoichiometric equivalent of 25 fmol recombinant protein. Preys were prepared in fresh 2% BSA in HBS for at least 30 min at 20 °C before applying to plates (or alternatively at least 60 min at 4 °C). After removing baits from plates, they were washed three times in 50 µl HBS-T supplemented with 0.8 µM desthiobiotin (Sigma D1411) to gently block any unoccupied biotin-binding sites. To each well, 50 µl of prepared prey multimers were then added. After a 60-min incubation at 20 °C, two further 75-µl washes in HBS-T with desthiobiotin were followed by a final wash in 75 µl HBS. Immediately after, 30 µl of TMB chromogenic substrate (Millipore ES001) was dispensed and allowed to incubate at 20 °C for 40 min. To stabilize the signal at this standardized time point, the reaction was halted by adding an additional 30 µl of 0.3% (m/v) NaF (Sigma 201154). Plates were then measured on a Tecan Spark plate reader for absorbance at 650 nm. The identity of each plate and well was then unblinded by a custom R script that stitched all measurements together and matched their numeric barcodes to the proteins they correspond to. All washing steps were performed with multidrop dispenser units (for example, Thermo Fisher Scientific 5840300), and in between washes, plates were centrifuged at 10*g* for 30 s upside-down with absorbent padding (Kimberly-Clark 7338) layered underneath to remove all trapped bubbles. As controls, every plate had at least one well with no bait added, one well with a tag-only bait construct and two wells of the *Plasmodium falciparum* P41 protein over which dilutions of the complementary protein P12 presented as a prey were manually added as positive controls for sensitivity[52].

## Cell-based binding assays

HEK293-E cells were transiently transfected with full-length human cDNA expression constructs (OriGene) for 42–46 h before staining. Ninety microlitres of cells were transferred to U-bottom 96-well plates (Greiner 650161), then washed on ice in 110 µl DPBS (HyClone SH30264). Cells were centrifuged at 300*g* for 3 min, the supernatant was removed, then resuspended in tetramers of biotinylated recombinant protein complexed with R-phycoerythrin-conjugated streptavidin (BioLegend 405245). Typically, a series of four tetramer quantities were used, ranging from 30 pmol to 1 pmol in a volume of 100 µl DPBS (including around 1 mM calcium and magnesium ions) with 1% BSA. Cells were incubated with tetramers for 45 min on ice before washing with DPBS. Washing consisted of topping each well with an additional 150 µl DPBS, centrifuging, resuspending in 250 µl cold DPBS, then centrifuging again to remove supernatant. Cells were resuspended in 100 µl DPBS with 1% BSA before being analysed on a LSR Fortessa flow cytometer (BD). Recorded events were gated for size and to remove doublets by their forward scatter and side scatter profiles using FlowJo software (v.10.6.1) as previously described[46]. Generally, around 20,000 gated events were collected for each well condition.

## Surface plasmon resonance

Both kinetic and equilibrium measurements were done on a Biacore 8K SPR instrument as previously described[53]. Biotinylated protein ligands were immobilized on streptavidin-coated Series S CAP chips (GE Life Sciences 28920234) to approximately 200 response units or the closest achievable level. A tag-only negative control was immobilized to an approximately equimolar level. Within 24 h of being purified by size-exclusion chromatography, analytes were injected at 100 µl per minute to derive kinetic data, or at 20 µl per minute for equilibrium measurements. Experiments were performed at 37 °C in HBS-EP buffer (GE Healthcare BR100669). A dilution series of analytes were tested along with at least one concentration in duplicate to check for consistency, plus a buffer-only cycle as a negative control. Response traces were analysed using the manufacturer's evaluation software (v.1.1). Response units were reference-subtracted and sensorgram data was fit using default parameters.

## Assembly of expression constructs

The full library of constructs for recombinant expression was assembled by a combination of cloning and gene synthesis. For cloned sequences, cDNA templates (OriGene) were amplified by PCR with primers delineating the extracellular domain. Overhangs on the primers introduced NotI and AscI restriction sites, which enabled ligation into the appropriate vector backbone. All assembled inserts not produced de novo by synthesis were verified by Sanger sequencing. Constructs that were not cloned from existing DNA templates were ordered as synthetic DNA (Twist Biosciences and Thermo Fisher Scientific GeneArt). Synthesized codons were optimized for human cell expression. Optimal Kozak consensus nucleotide sequences were included in all constructs, which occasionally required mutating the second amino acid of the endogenous signal peptide to alanine. Plasmids were prepared to transfection-grade quantity using midiprep or maxiprep kits (for example, Invitrogen K210007).

## Protein library design

The cell-surface proteomes of blood immune cells were defined by two sources. First, the full dataset of a previous high-resolution proteomics survey of 28 leukocyte populations in resting and activated states (for 44 cell types and states total, covering all major categories)[1] was merged against a previously established manually curated list of every cell-surface protein in the human genome[54,55]. The cell-surface proteome list was manually reviewed to verify that each protein does not have publications measuring its localization that contradict a presence on the cell surface. Every protein that was detected was included regardless of how low the expression counts were, with the exception of highly polymorphic proteins such as HLA-A. Second, we added all proteins with a designated CD number as of the 10th human cell differentiation molecule workshop[56]. The amino acid sequences and topologies[57] of these proteins were manually inspected to determine the extracellular regions and in which structural class the protein belonged (out of type I single-pass/GPI-anchored, type II single-pass, multi-pass, and proteins that function as obligate dimers such as integrins). Proteins that lacked a single contiguous extracellular region of at least 20 amino acids after signal peptide processing were excluded. Similarly, multi-pass proteins without a clear contiguous extracellular domain to express were excluded as incompatible with our recombinant expression system. Constructs were produced as synthetic DNA sequences by Twist Biosciences or Thermo Fisher Scientific GeneArt, optimizing codons for human cell expression. As previously described, proteins were cloned into pTT3 expression vectors matching the intended topology[49]. Single-pass proteins with N-terminal extracellular domains retained their endogenous signal peptides in cases in which a well-annotated or SignalP-predicted signal peptide could be found. Otherwise, an exogenous signal peptide based on the mouse kappa antibody secretion sequence was inserted. Proteins with N-terminal domains had tags attached to their C terminus, whereas the inverted design was used for type II C-terminal proteins[58]. All proteins were produced as fusions with an established recombinant linker comprising domains 3 and 4 of rat CD4 in place of the original transmembrane sequence (termed rCD4)[59], along with a biotin-acceptor site for covalent modification and a hexahistidine tag for purification purposes as described[14]. In the case of proteins that exist as dimeric complexes, including integrins, HLA-related molecules, CD1, CD8, GPIb, CD79 and CD94 family NK receptors, we relied on a previously determined design in which one

chain would lack any tags, thus directing the purification of full complexes, particularly in cases in which the tagged chain is not secreted on its own[60]. Proteins with an intrinsic multimerizing ability owing to intermolecular disulfide bonds, including the TNF-superfamily CD27 and HVEM trimers or B7-family ICOS and CD28 dimers, were expressed with the appropriate cysteine residues intact and allowed to form functional complexes after cell secretion.

### Processing of data from the binding screen

The matrix of bait versus prey raw absorbance values from the SAVEXIS screen was processed through a two-way median polish. The two separate phases of screening were processed separately then combined to remove per-phase batch effects. Median-polished signals for each protein pair were summed across the two bait–prey orientations (or doubled in the case of homophilic interactions), then all interactions that gave a signal of at least 1.0 were selected to be re-measured in the secondary screen, with the exception of proteins that gave a highly variable background signal in the primary screen. Median-polished signals from the secondary and primary screens were combined by a weighted sum that valued the lower-throughput secondary screen measurements three times more than the high-throughput primary screen measurements. Every protein that gave a clearly reproducible signal in at least one orientation (32 interactions total, identified using our ROC analysis) was then followed-up with validation assays. We excluded interactions with proteins that were highly promiscuous in the screen; that is, appearing more than 20 times in the top 1,000 ranked interaction pairs. Most of these have known mechanisms explaining their lack of specificity, such as being known lectins. Our list of proteins that frequently recur in binding assays may be useful in guiding the interpretation of future screen results, as we note that many of them (for example, CLEC receptors, certain CEACAMs, NRP1, IGF2R, FGFRs and LDLR) are frequently also reported to be binding partners in the context of other published studies.

### Manual curation of literature interactions

Interactions involving proteins in our immune library were systematically compiled in a three-step process. First, every protein was checked across published reference manuals of immune surface molecules to see whether any interactions were claimed[61,62]. Second, for each protein, its name and applicable synonyms were searched in Google and PubMed using standardized search terms including <"protein name" AND (binding OR interaction OR affinity)> and <"protein name" AND (SPR OR kinetics)>. Finally, existing databases were evaluated, including CellphoneDB, IntAct, PCDq, BioGRID, OmniPath and other published lists[63,64]. Claims identified through these methods were manually verified by identifying the original publications behind each claim. Only interactions supported by citable experimental results were included. In the process, false positives in these databases were removed, such as common falsely claimed interactions based on mouse experiments that are demonstrated not to be conserved in humans, mistakes made by databases mapping protein names, issues arising from considering single genes instead of functional surface protein complexes, or outdated interaction claims that have since been rejected by a consensus of later studies. Wherever quantitative measurements of monomeric binding affinities were available, these were extracted from the original papers. In the process of this manual curation, other relevant results mentioned in the papers were also documented, including if an interaction measurement gave a demonstrably negative result.

### Benchmarking screen results

Processed binding signals from the arrayed screens were benchmarked against reference sets derived from the detailed manual curation of the literature. Positive reference sets were defined either as every interaction with a claim in the literature, or more stringently as every claim with validation by either a quantitative method (for example, SPR, analytical ultracentrifugation or radiolabelling) or a co-crystal structure (for example, X-ray crystallography or cryogenic electron microscopy). Negative reference sets were either based on experimentally measured negative interactions, or by defining a random negative reference set as previously recommended[11]. ROC and precision-recall curves were calculated using the PRROC package in R. For these ROC and precision-recall curves, the performance of every possible threshold for converting the median-polished absorbance measurements from the screens into a binary classification of 'interacting' or 'not interacting' is evaluated against the respective benchmark. The area under the curve (AUC) is reported as an overall summary of screen performance. Only proteins with detectable evidence of recombinant expression were considered when calculating classification performance in the main figures.

### Integration of expression data and binding matrices

Because our protein–protein interaction network largely represents molecular connections that would occur between cells, we integrated our interaction matrices identifying which cell-surface proteins bound each other with expression data identifying which cell-surface proteins were present on different cells. Expression data included proteomics from bulk-sorted immune cell types and single-cell RNA datasets. We iterated through all possible pairs of cell types in the expression dataset and all pairs of proteins with identified interactions in our binding dataset between those cell-type pairs. From this, we created a master data key that lists all detected molecular interactions between all cell pairs. The mappings of gene identifiers to the Uniprot accessions used in the interaction network files were manually verified to ensure no errors or missed values. From the full listing of cell–cell interactions and the molecules mediating them, we could then perform quantitative or qualitative analyses by either using the expression values or binarized lists of detected interactions. For binarization, gene or protein expression matrices were generally expressed in the form of percentages of replicates (for bulk datasets) or cells (for single-cell datasets) in which expression was detectable in that cell type. Binarization of expression then could be standardized by setting a minimum per cent threshold (that is, for most bulk datasets, expression detected in at least a majority of replicates; or for single-cell datasets, following a common precedent of thresholding at 10% detection[54,65]). When comparing diseased to control tissue samples, this process would be repeated separately for the control and disease expression data, and then for each interaction whether it was detected according to these standards in one, both or neither condition could be determined. All integrations were performed in R.

### Processing of single-cell RNA-sequencing data

Single-cell RNA-sequencing (RNA-seq) datasets[38,66–70] were processed following a standard data-cleaning pipeline using the Scanpy package in Python (v.1.4.5)[71]. Cells with more than 10% of all reads coming from mitochondrial sequences were removed, as were cells with fewer than a minimum of 200 genes or more than a maximum of 3,000 genes. Genes that were detected in fewer than two cells were not considered. Cell-type labels taken from the original published studies were always retained when available. Cell types in the bone marrow dataset were manually annotated after Louvain clustering on the top 1,000 highly variable genes following a previously described pipeline[66,72]. Cell-type clusters with fewer than 10 total cells were not included in subsequent analyses. For Circos-style plots, the ShinyCircos package was used to display the integrated single-cell RNA and interaction matrix data. Linkages on the Circos-style plots are drawn where two cell types express an interacting cell-surface protein pair above a threshold requiring a minimum of 10% of single cells in a cell-type cluster to have at least one mRNA read detected for the surface protein. For visualization, ubiquitous interactions are not displayed, but users can explore different visualization criteria on our interactive website. Signalling analysis was done using the NicheNet package in R (v.1.0.0) with the ligand–target matrix constructed off the immune receptor interactome (known plus novel)

described in this manuscript using all default settings for custom model construction with no parameter optimization[26]. Differential expression testing between diseased and paired reference tissue was done with the Seurat package (v.3.1.5)[73] using a non-parametric Wilcoxon rank sum test. Genome-wide multiple testing correction was applied.

## Immune activation differential expression

Cell types with proteomics measurements in both activated and resting states were used for differential expression calculations. The DESeq2 package in R was used to model expression counts and compute Wald test statistics[74]. From these results, the set of differentially expressed cell-surface proteins upon immune activation were determined by setting a fold change threshold of greater than 2. Alternatively, we also compared using adjusted *P* value thresholds, which gave similar results (that is, Extended Data Fig. 6). Each protein was mapped to its measured binding affinity for the interaction(s) it participates in. For proteins with multiple interactions, this ambiguity was addressed by including the affinity of every binding interaction as separate points. These affinity values for each cell type were then grouped on the basis of whether their corresponding proteins were upregulated or downregulated upon immune activation. The affinities of upregulated and downregulated interactions were compared by Welch's *t*-test.

## Mathematical model based on cell-binding kinetics

Details of the kinetic model and derivations of equations can be found in the Supplementary Equations. For each blood immune cell type to be modelled, published parameters about its physical geometry, proportions and protein expression were compiled. When cell types were to be matched with experimental data containing less subtype resolution than the proteomics expression dataset, expression values were estimated as the weighted average of all of a cell type's constituent subtypes, weighted by their measured proportions in blood (for example, if total CD56+ NK cells were measured in the experiment, the 'NK dim' and 'NK bright' subtypes measured in the proteomics would be proportionally averaged in the model). Absolute protein counts per cell from the expression data were converted to average protein density per surface area by assuming that all protein is present on an approximately spherical cell surface. Per quantified interaction, a relative equilibrium density of bound protein molecules was calculated through the Michaelis–Menten equation[75]. To determine the relative connection affinities of different cell types, the sums of all interactions calculated were compared for different pairs of cells.

## Binding-perturbation model using differential equations

Although the core kinetics model can calculate relative cellular affinities, by formulation it cannot on its own predict the outcomes of specific perturbations to proteins (for example, its predictions from removing a surface protein would all uniformly be decreases in binding, or predictions from a strengthening in affinity would all be increases). Thus, when making perturbation predictions, the relative cellular affinities were passed as parameters to a system of differential equations based on the law of mass action. Specifically, all cell types were assumed to collide and form connections at a constant rate, and the dissociation rate for that cell–cell bond was inversely proportional to the relative affinity determined by the core kinetics model. As initial conditions, all cells were assumed to be unbound at frequencies that match literature-reported values for human blood. Numerical integration proceeded until equilibrium was reached. Calculations were performed using the PySB package in Python (v.1.11)[76]. Our initial perturbation studies have simulated removing particular surface proteins by setting the expression values for that protein to zero across all cell types.

## Network centrality calculations

Counts of binarized interactions following integration of expression datasets with the interaction table were converted into a weighted undirected network graph using the igraph package in R (v.1.2.5). Eigenvector centralities were calculated for each cell type in the graph. Each tissue with a single-cell resolution dataset available was computed separately. To compare myeloid cell populations to other lineages, we performed two-sided Welch's *t*-tests on the centrality metrics. *P* value corrections for the multiple tissues tested were done by the Benjamini–Hochberg procedure. For these and all other box plots, the central box displays the 25th, 50th and 75th percentiles, with whiskers extending to 1.5 times the interquartile range.

## Spatial transcriptomics analysis

Lymph-node spatial transcriptomic data were downloaded from 10x Genomics and processed using a standard cleaning pipeline in Python with the Scanpy package. Measured spots on the array were constrained to have between 4,000 and 36,000 total transcript counts, with fewer than 20% of reads derived from mitochondrial sequences, and at least 2,000 different genes detected. Genes were constrained to be detected in at least five spots. A radius of 150 units on the spatial coordinates was empirically determined to encompass only the immediately connected neighbouring spots to a given spot's centre, and used when determining neighbour relationships. Iterating through all protein pairs in the interaction network, spots were marked for whether they detectably expressed at least one count of either a single protein of the pair, both proteins or neither. The number of instances in which both proteins of an interacting pair were detected in physically connected spots (either directly adjacent spots or the same spot containing cells expressing both, owing to how the resolution of spots means that each spot on average contains more than one cell) was tabulated, as was the number of physically connected spots in which both expressed only the same protein but not an interacting pair. To calculate colocalization scores for each pair of protein-coding genes in a receptor–ligand interaction, the fraction of pairings that are 'interaction-capable' (that is, having physically neighbouring expression of the receptor and ligand) was calculated over the entire lymph-node tissue section. To test our experimentally found interaction list against a null hypothesis of randomly paired cell-surface proteins, we took the same proteins as in our true interaction network and randomly permuted which were paired together. The colocalization scores from this permuted null distribution were compared against both the scores from our literature-curated interaction list and our empirically discovered interactions. Statistical testing consisted first of an omnibus one-way ANOVA between these three sets of colocalization scores, followed by a post-hoc Tukey's honestly significant difference test.

## In situ hybridization of lymph nodes

Fresh, unfixed tissue samples were flash-frozen in OCT using dry-ice-cooled isopentane and morphology was checked by haematoxylin and eosin staining. For RNAScope, 10-μm-thick cryosections were cut onto SuperFrost Plus slides, fixed for 15 min with chilled 4% paraformaldehyde (PFA) followed by 90 min in room temperature in 4% PFA, then dehydrated through an ethanol series (50%, 70%, 100% and 100% ethanol). Slides were then processed using the RNAScope 2.5 LS multiplex fluorescent assay (ACD, Bio-Techne) on the Leica BOND RX system (Leica) with protease III treatment. Initially, RNAScope positive and negative control probes were tested on sections, before proceeding to probes of interest with fluorophores opal 520, opal 570 and opal 650 at 1:1,000 concentration. All probes were previously established and are commercially available under the catalogue numbers 546188-C3 (*JAG1*), 845158-C1 (*VASN*), 491518-C4 (*VISTA*; also known as *VSIR*), 457368-C1 (*HLA-E*), 460048-C2 (*PLXNA4*), 418328-C1 (*APP*), 442598-C1 (*CNTN1*), 601738-C3 (*MCAM*) and 601998-C4 (*CD45*; also known as *PTPRC*). These were then imaged at 20× magnification on a Perkin Elmer Opera Phenix High Content Screening System with water immersion.

## Isolation of human PBMCs

Blood buffy coat from a healthy donor was obtained by the Blutspende Zurich, under a study protocol approved by the cantonal ethical committee Zurich (KEK Zurich, BASEC-Nr 2019-01579). To obtain PBMCs, the sample was diluted 1:1 in PBS (Gibco) and cells were isolated with a Histopaque-1077 density gradient (Sigma-Aldrich) according to the manufacturer's instructions. Subsequently, cells at the interface were collected, washed once with PBS and resuspended in RPMI 1640 + GlutaMax medium (Gibco) supplemented with 10% human serum (Chemie Brunschwig). Immune cells were seeded and cultured in CellCarrier 384 Ultra, clear-bottom, tissue-culture-treated plates (PerkinElmer) at a density of $2 \times 10^4$ cells per well in 50 µl per well and incubated at 37 °C and 5% $CO_2$. Cell number and viability was determined by use of a Countess II Cell Counter (Thermo Fisher Scientific).

## Leukocyte protein perturbation and fixation

Isolated leukocytes were incubated with purified recombinant proteins tetramerized around neutravidin (Thermo Fisher Scientific 31000) at doses of 80 pmol (1.6 µM) to 200 pmol (4 µM) per well. Additional negative controls of buffer-only, a tag-only tetramer and the elution material from the supernatant of mock-transfected cells were included. All proteins for which a novel interaction was identified and that could be sufficiently expressed to provide a 4 µM concentration were included, along with CD209, CD58, ICAM1 and SIRPA as previously characterized controls. Assay plates were created in a fully randomized layout and prepared by an Echo 555 Liquid Handler. Four assays were done with five replicates per condition per assay, including 4 h and 24 h time points with and without the addition of 1 pg µl⁻¹ LPS. The assay was stopped by fixing and permeabilizing the cells with 20 µl per well of a solution containing 0.5% (w/v) formalin (Sigma-Aldrich), 0.05% (v/v) Triton X-100 (Sigma-Aldrich), 10 mM sodium (meta)periodate (Sigma-Aldrich) and 75 mM L-lysine monohydrochloride (Sigma-Aldrich). After a 20-min incubation at room temperature, the fixative-containing medium was aspirated by use of a HydroSpeed plate washer (Tecan). The cells were then blocked (50 µl per well) with PBS supplemented with 5% fetal bovine serum (Gibco) and photobleached for 4 to 24 h at 4 °C to reduce background fluorescence by illuminating the fixed cells with conventional white light LED panels.

## Immunostaining and imaging

For immunohistochemistry staining, all primary antibodies were diluted 1:300 in PBS with 6 µM DAPI (Sigma-Aldrich) for nuclear detection. Antibodies used were anti-CD3 Alexa Fluor 647 (BioLegend, Clone UCHT1), anti-CD4 FITC (BioLegend, clone SK3), anti-CD8 PE (BD Biosciences, clone SK1), anti-CD19 FITC (BioLegend, clone SJ25C1), anti-CD56 PE (Beckman Coulter, clone N901), anti-CD16 PE (BioLegend, clone 3G8), anti-CD14 Alexa Fluor 647 (BioLegend, clone HCD14) and anti-CD20 PE (BD Biosciences, clone 2H7). Per well, 20 µl of the antibody cocktail was added and incubated for 1 h at room temperature. For imaging, a PerkinElmer Opera Phenix automated spinning-disk confocal microscope was used and each well of a 348-well plate was imaged at 20× magnification with 5 × 5 non-overlapping images, covering the whole well surface. The images were taken sequentially from the bright-field (650–760 nm), DAPI/nuclear signal (435–480 nm), GFP signal (500–550 nm), PE signal (570–630 nm) and APC signal (650–760 nm) channels. Raw .tiff files were exported for analysis.

## Image processing and quality filtering

Cell detection and single-cell image analysis was performed using Cell-Profiler (v.2)[77]. Nuclear segmentation was performed by thresholding on DAPI intensity. Cellular outlines were estimated by a circular expansion from the outlines of the nucleus. In addition, a second and larger expansion from the nuclei was performed to measure the local area around each single cell (local cellular background). Standard CellProfiler-based intensity, shape and texture features of the nucleus, cytoplasm and the local cell proximity were extracted for each measured channel. Raw fluorescent intensities were $\log_{10}$-transformed and normalized towards the local cellular background as previously described[25].

## Cell-type image classifier

An 8-class 71-layer deep convolutional neural network with an adapted ResNet architecture[78] was implemented using 48 × 48 × 5 input images in MATLAB's Neural Network Toolbox (v.R2020a). For all morphology classifiers (B cell, NK cell, T cell) a 2-class 39-layer deep convolutional neural network (CNN) with an adapted ResNet architecture was used. Input images of 48 × 48 × 3 were used, in which all images contained the DAPI and the bright-field channel whereas the third channel contained the respective channel with the lineage marker. In the case of dendritic cells, the absence of all other lineage markers was used. In all CNN classifications, 48 × 48 pixel sub-images around each nuclei centre were generated. Cells closer than 24 pixels to the border of an image were excluded from all classifications. Network training, evaluation and classification were performed as previously described[29].

## Image processing for cell interactions

For extracting cell–cell interactions from image data, a simplified version of a previously published[25] method was used. Cell–cell interaction analysis was conducted over all different image sites within the same well. Cells were scored as interacting if their nuclear centroids were within a Euclidean distance of 40 pixels. To calculate the interaction score of a cell with type A interacting with a cell of type B, we first calculated specific interactions and total interactions per well. We define specific interactions as the total count of type B cells within the defined radius around a cell of type A. Total interactions are considered as the total count of all interacting cells in that well. To calculate the final interaction score, specific interactions were divided by the product of (the fraction of type A cells of all cells) × (the fraction of type B cells of all cells) × total interactions.

## Pharmacoscopy data processing and analysis

For graphical displays, observed cell state and cell–cell interaction frequencies were first normalized against their respective controls. Recombinant protein conditions were normalized against an average of the control wells for each respective time point, dose and background immune activation. Protein controls consisted of a buffer-only mock treatment well, a well with carryover from purifying an empty transfection and a well stimulated with only the protein epitope tags. As previously described for pharmacoscopy experiments, normalizations were calculated as the observed value minus the control average, divided by the maximum of those two values to give a metric bounded from [−1, 1] with 0 representing no change relative to control. For statistical analysis, the raw measurements across all available control conditions were compared against all replicates of each corresponding perturbation condition by Welch's $t$-test. To adjust for multiple testing, the Benjamini–Hochberg test was used and a 10% false discovery rate threshold was set for delineating significant effects. These calculations are shown both for each individual dose applied, as well as when all doses of a given protein treatment were aggregated. The same overall analysis procedure was done for both the cell state frequencies and the cell–cell interaction datasets, primarily differing in which time point (24 h or 4 h, respectively) was chosen for focused statistical analyses. The non-classical CD16-positive monocyte population was omitted from all final plots as the extremely low numbers of these cells that were detected in each experiment led to inconsistent and often non-finite effect sizes (for example, changes from 0 cells found to 1 cell found between condition and control).

## Model comparison to microscopy interaction data

The cell types measured in the pharmacoscopy experiments were all incorporated into the system of differential equations in the model to

determine both baseline cell–cell interaction frequencies and to predict changes when single surface proteins were removed from the model. The model-calculated frequencies of cells in unbound and all possible interacting paired configurations were processed into interaction scores by following the same equations used to process the microscopy image data. Data from the 4-h time point with no background LPS stimulation were used for all protein conditions. Because our recombinant proteins could either trigger or suppress a given receptor pathway, we compared the absolute magnitudes of the normalized interaction scores to quantify the extent of perturbation. For the final analyses, we restricted the set of proteins to compare to only those for which the model predicted that perturbing that protein would produce meaningful changes in cell–cell interactions (defined as the model prediction maximum for a condition being no lower than one-fifth of the median perturbation prediction maximum across the full set of proteins for which we gathered experimental data). The top third of cell pairs that were predicted to have the greatest perturbation magnitude according to the model were contrasted with the remaining cell pairs that were predicted not to be strongly changed after perturbation. A one-sided $t$-test was then done to determine whether the observed changes in interaction score after the addition of recombinant proteins in our experiment for the cell pairs that were predicted by our model to be perturbed were significantly greater than the baseline interaction score changes of all other cell pairs that were not predicted to change. $P$ values were adjusted using the Benjamini–Hochberg procedure.

## Access to human tissue

Human lymph nodes were obtained from deceased transplant organ donors by the Cambridge Biorepository for Translational Medicine (CBTM) with informed consent from the donor families and approval from the NRES Committee of East of England – Cambridge South (15/EE/0152). This consent includes the generation of open-access genetic sequencing data and publication in open-access journals in line with Wellcome Trust policy. CBTM operates in accordance with the guidelines of the UK Human Tissue Authority. Blood samples from anonymized healthy donors were provided by the Blutspende Zurich, under a study protocol approved by the cantonal ethical committee Zurich (KEK Zurich, BASEC-Nr 2019-01579).

## Reporting summary

Further information on research design is available in the Nature Research Reporting Summary linked to this paper.

## Data availability

The data files required to generate the figures and analyses in this paper are provided in the GitHub repository accompanying this manuscript: https://github.com/jshilts/shilts-et-al-2022-immunoreceptors.

## Code availability

The custom code files used for this study are publicly available in the GitHub repository accompanying this manuscript: https://github.com/jshilts/shilts-et-al-2022-immunoreceptors.

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

**Acknowledgements** This research was principally funded through the Wellcome Trust (grant 206194). Y.S. and B.S. were funded through the Swiss National Science Foundation

(grant PP00P3_163961). We thank L. Parts for discussions on mathematical modelling; A. Wilbrey-Clark for coordinating in situ hybridization experiments; I. Bronner for assistance with robotics; and M. Quail and J. Parkhill for access to laboratory facilities.

**Author contributions** J.S. and G.J.W. designed the study. J.S. developed the SAVEXIS technique. J.S. prepared all recombinant proteins, with assistance from N.M.-S. during screening. J.S., N.M.-S. and Z.-S.C. conducted the receptor interactome screen measurements. J.S. analysed screen data. J.S. performed cell-based binding assays. F.G. performed SPR. J.S. constructed the mathematical model of cell binding. S.P. performed in situ hybridization experiments. J.S. processed and analysed all transcriptome expression datasets. S.T. and R.V.-T. supervised expression analysis. Y.S. performed pharmacoscopy assays. Y.S. and J.S. analysed the imaging data. B.S. supervised the pharmacoscopy analysis. G.J.W. supervised all other experiments. J.S. wrote the manuscript with comments from all authors.

**Competing interests** S.T. has received remuneration in the last three years for consulting and membership of scientific advisory boards from Foresite Labs, GlaxoSmithKline, Biogen, Qiagen and Transition Bio, and is an equity holder of Transition Bio. The remaining authors declare no competing interests.

**Additional information**
**Correspondence and requests for materials** should be addressed to Jarrod Shilts or Gavin J. Wright.

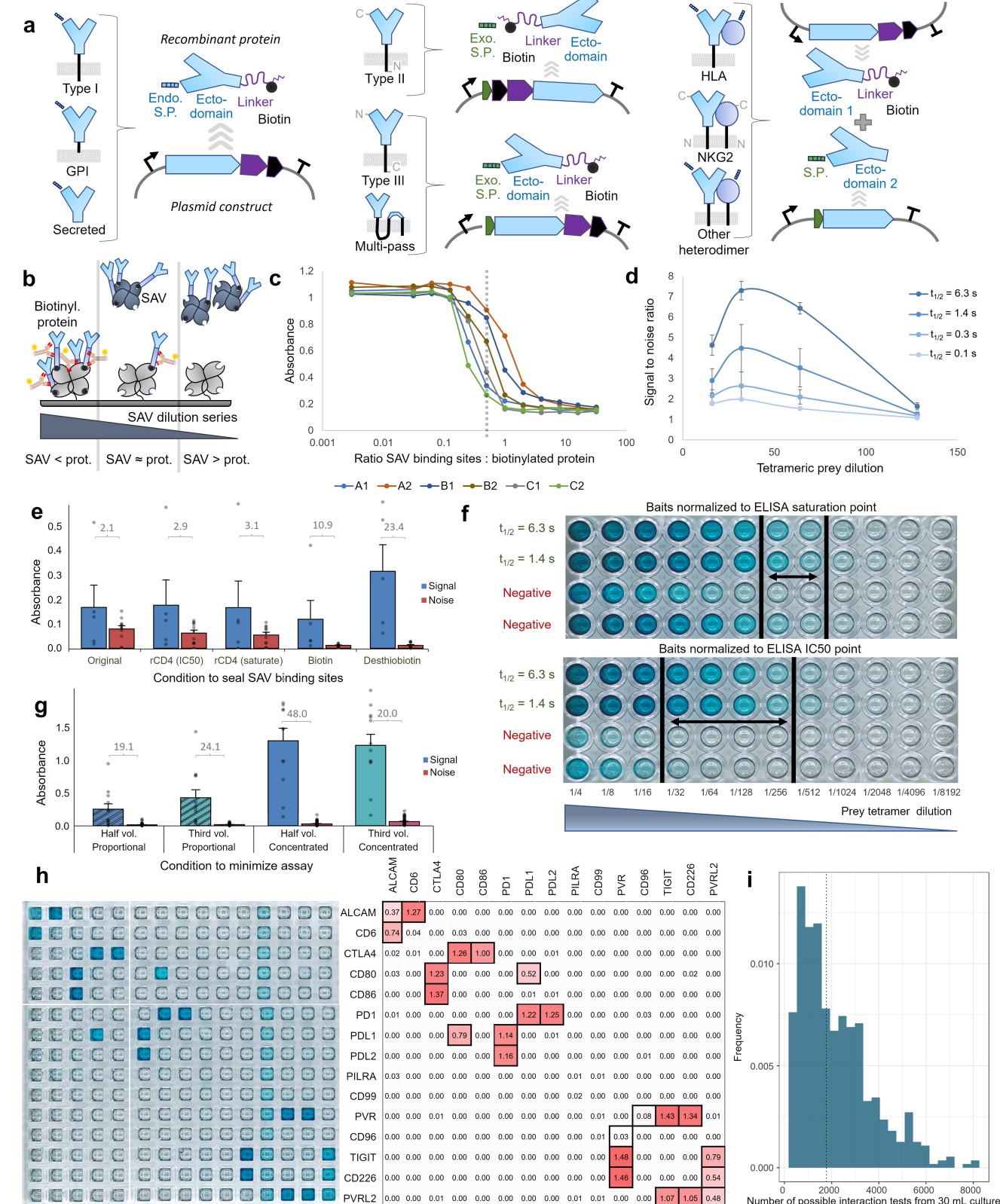

**Extended Data Fig. 1 | See next page for caption.**

**Extended Data Fig. 1 | Developing SAVEXIS. a**. Design of expression vectors and recombinant protein constructs for SAVEXIS screening of divergent architectural classes of cell-surface proteins. For heterodimers, the exact formulation of each chain will depend on the receptor subunit's topology (e.g. using the Type I vector for integrins, and Type II for CD94/NKG2). **b**. Empirically gauging streptavidin multimerization stoichiometry by ELISA. Schematic of the procedure for measuring tetramerization around streptavidin by titrating soluble streptavidin (SAV) against a fixed concentration of biotinylated protein before transferring to a streptavidin-coated plate for an ELISA. The measured dilution at which signal ceases represents the optimal tetramerization stoichiometry. **c**. Calculated stoichiometric equivalence points of 6 example biotinylated proteins incubated with streptavidin. The 4:1 stoichiometric equivalence point inferred mathematically based on molecular mass calculations is indicated on the x axis as "1". A dashed line indicated the empirically measured median equivalence point. A set of 6 *Danio rerio* Jam proteins were measured, with the average indicated by a dashed line. **d**. Titrating prey concentrations identifies a common prey activity with optimal sensitivity. For 4 known zebrafish receptor–ligand interactions of varying affinity (colour shades, ordered by known binding affinity expressed as dissociation half-life), the ratio of raw absorbance signal for the specific interaction against non-specific interactions (y axis) is measured across prey concentrations (y axis). Error bars represent the standard error of the mean. n = 6 technical replicates. **e**. Applying soluble desthiobiotin greatly enhances the assay signal to noise ratio by sealing unoccupied biotin-binding sites. Measurements of a set of 3 example interactions compared without applying any sealing step in between bait and prey incubations ("Original"), applying biotinylated rCD4 tag at either IC50 or plate-saturation concentrations, or applying a molar equivalent of saturation with either biotin or desthiobiotin. The ratio between mean signal to noise is indicated above each condition. Error bars represent the standard error of the mean. n ≥ 5 independent wells. **f**. Immobilizing lower quantities of bait protein can reduce off-target signals. Example interaction assays with baits saturating each well (top) or baits at half-maximal dilutions (bottom). A dilution series of prey is applied across each column. The top two rows are specific interactions (binding half-lives listed in green), and the bottom two are known non-interactors. **g**. Assay miniaturization into 384-well plates can retain and enhance performance. Different assay miniaturization strategies to adapt from 96-well format to 384-well format are indicated along the x axis, including reducing by a half or a third all volumes and protein amounts proportionally (left two), or only reducing volumes while concentrating the samples so the total protein quantities applied are conserved (right two). The ratio between mean signal to noise is indicated above each condition. Error bars represent the standard error of the mean. n ≥ 12 independent wells. **h**. Representative screening example for human receptors. The appearance of the raw screening plate (left) is shown alongside absorbance values following median polish normalization (right). Bold borders indicate interactions that are expected based on literature publications. One protein, corresponding to a construct later found to be incorrectly labelled, is omitted. **i**. SAVEXIS consumes small enough quantities of protein that thousands of assays are possible from small input sizes. For a sample set corresponding to all measured proteins from the leukocyte surface receptor library, the amount of interaction tests that could be performed for each protein based on its expression yield was calculated. As a typical case illustration in which protein is purified from a 30 mL cell culture, the median number of interaction tests possible is indicated by a dashed line.

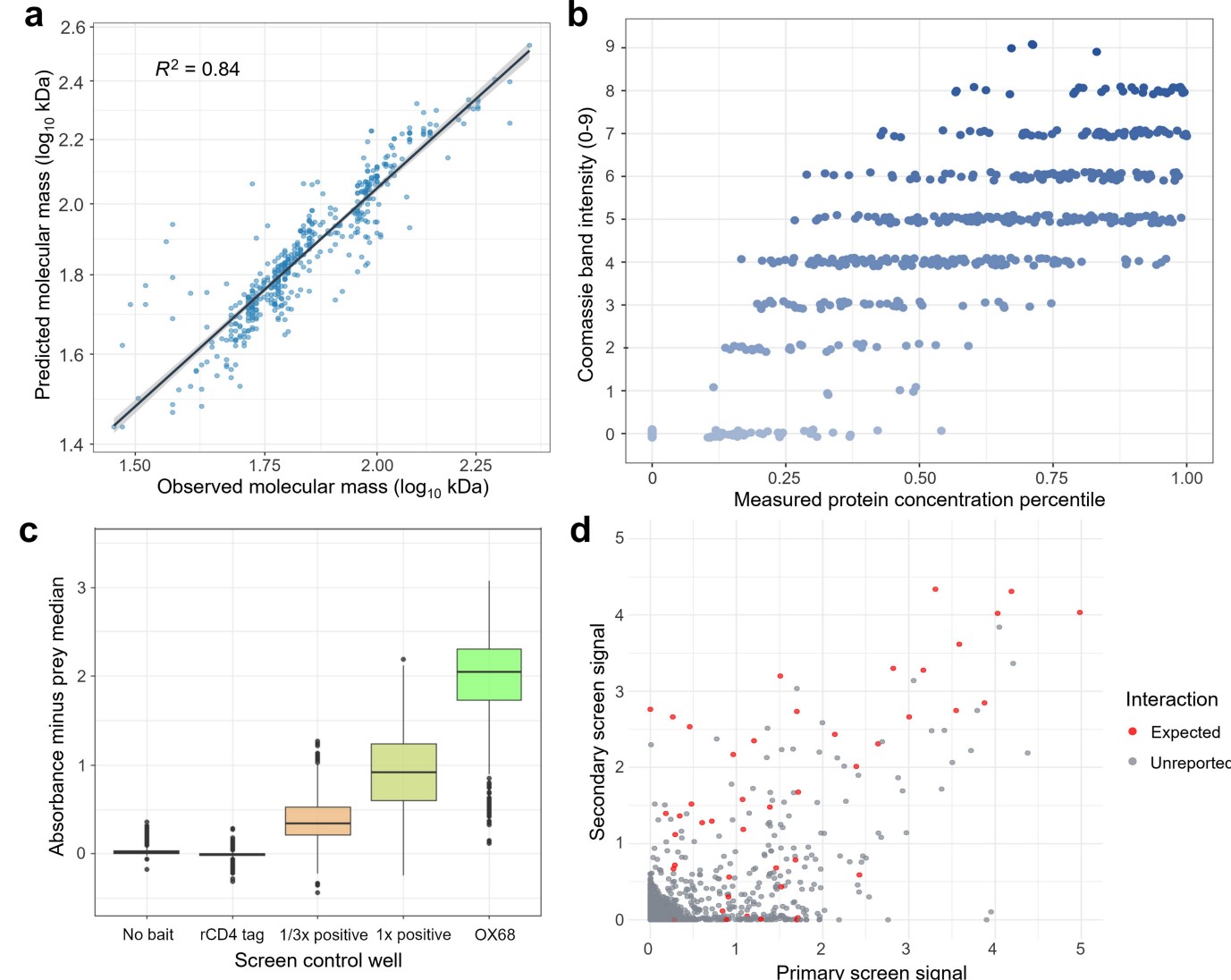

**Extended Data Fig. 2 | Interaction screen quality controls testify to robust measurements. a**. Recombinant proteins produced at scale match their expected molecular masses. Observed molecular masses from denaturing protein gel electrophoresis (x axis) are compared against computationally predicted molecular mass. Predictions were made by taking the known masses of each amino acid in the protein after processing of its signal peptide, with 2.5 kDa added per predicted N-linked glycosylation site. Shading indicates the 95% compatibility interval for the least-squares linear regression fit. Full images for all gel electrophoresis samples are provided in Supplementary Fig. 1. **b**. Quantitative protein concentration measurements by Bradford assay agree with qualitative estimates of protein concentration based on densitometry of Coomassie-stained protein gels. Measured concentration percentiles (y axis) are compared against discretized expression categories based on staining intensity (y axis and colour shade). **c**. Control wells included on every screening plate indicate high consistency across the primary interaction screen. Boxplots of plate measurements for negative control wells (blank baits and tag-only rCD4 baits), positive control wells (the known interaction between *P. falciparum* P12 and P41 at either a 1x or 3x dilution), and loading control wells (OX68 antibody) that capture prey proteins by recognizing their rCD4 tag. n = 1,262 wells per condition. **d**. Positive interactions from the primary screen were reproducible in a secondary screen with independently produced and measured proteins. For each protein–protein pair measured, the processed signal in the primary screen (x axis) is correlated against the signal in the secondary screen (y axis). Pairs previously described as being interactors are denoted by red colouration.

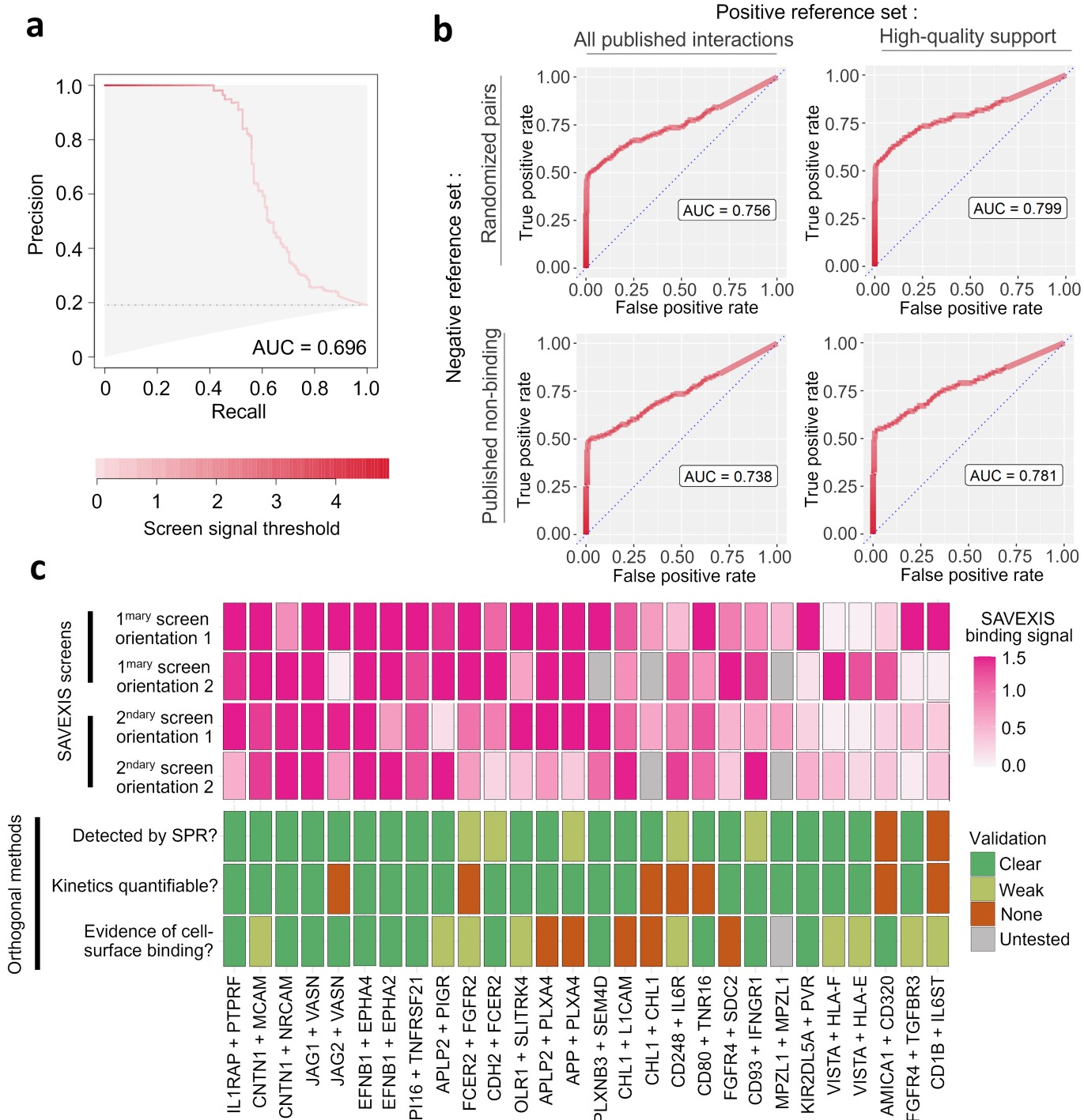

**Extended Data Fig. 3 | Interactome validation summaries. a.** Precision-recall curve corresponding to Fig. 1d. Colour shading indicates the cut-off for the summed screen signal across both bait–prey orientations. The performance of a random classifier is shown by the dotted line, and grey shading indicates the valid range between perfect performance and a random classifier. This curve only considers proteins for which expression was detectable, and defines a positive set based on previously published interactions and a negative set based on randomized interaction pairs. **b.** ROC curves of screen performance are consistent across possible definitions of positive and negative sets. These curves consider all proteins regardless of whether they were detectably expressed. Columns provide different positive reference sets, and rows delineate possible negative reference sets. High-quality support refers to interactions with experimental support by SPR, isothermal titration calorimetry, analytical ultracentrifugation, or a co-crystal structure. **c.** Overview of evidence for newly identified interactions. Data from both measured bait–prey orientations in the primary and secondary screen are indicated in pink. Results from cell-binding experiments and SPR are categorized for each interaction along a simple qualitative scale of green to red for ease of comparison. This includes whether a binding response was detectable in SPR equilibrium experiments, if the binding response in SPR experiments was sufficiently quantifiable that 1:1 binding models could be fit, and whether gains in cell-surface binding were observed in cell-based assays when the counter-receptor was overexpressed. The full experimental results that are summarized here can be found in Extended Data Fig. 4.

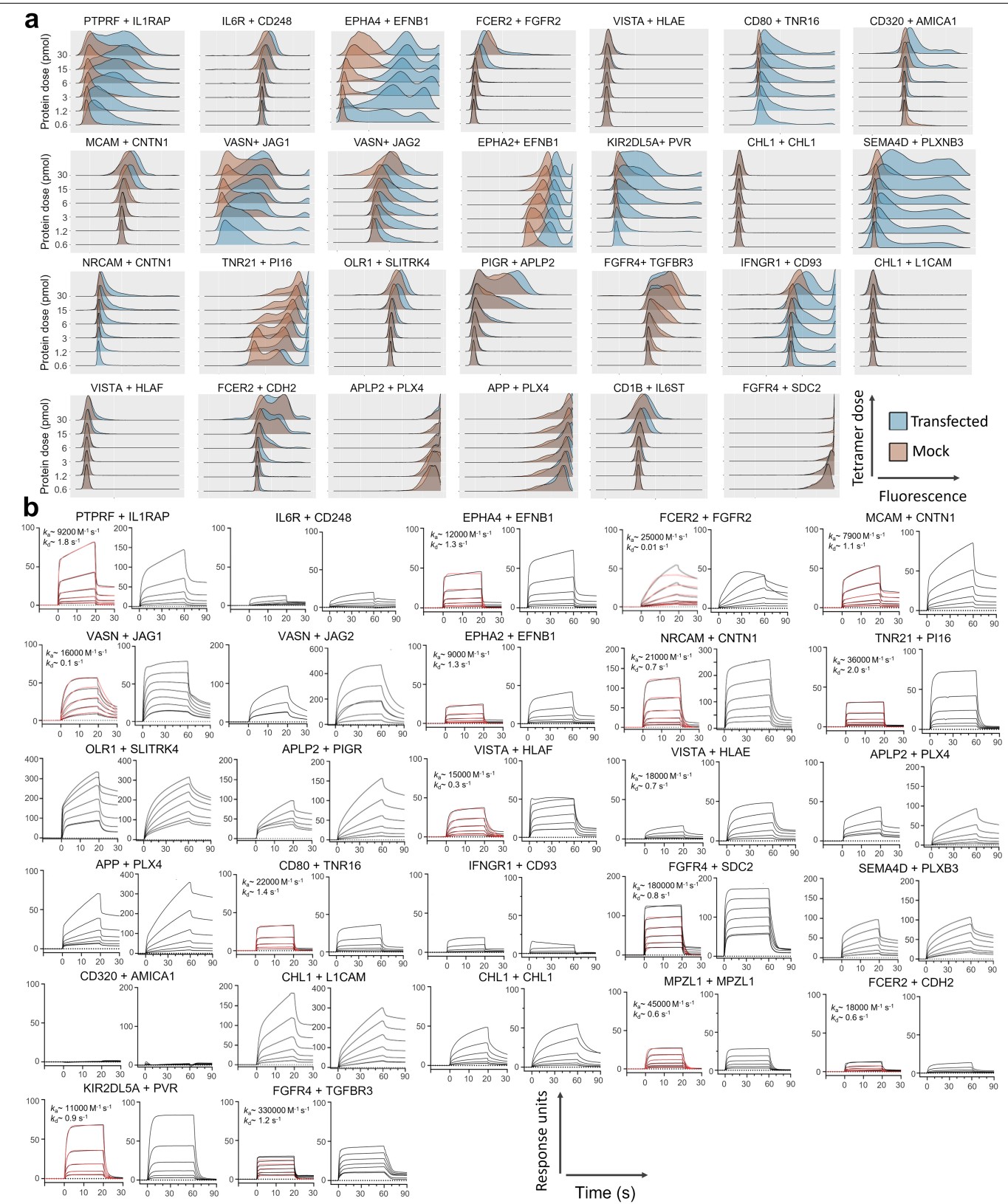

**Extended Data Fig. 4 |** See next page for caption.

**Extended Data Fig. 4 | Orthogonal binding assays to confirm each interaction. a**. Full set of cell-binding traces, extended from Fig. 2a. For each interaction pair, the protein named on the right of the title was transfected into human cells and the protein named on the left was tested for binding as a fluorescently linked recombinant protein tetramer. Tetramer binding to cells (x axis) was measured by flow cytometry at different tetramer concentrations (y axis). The traces in blue are cells overexpressing the indicated receptor, and red shows binding to the mock-transfected control cells. Because cDNAs may express to widely varying levels or not at all, and some proteins may bind to endogenously expressed HEK293 surface proteins, some experiments give inconclusive binding data. For example, HLA-F is known to be predominantly sequestered intracellularly[79,80], whereas soluble APP is known to already have strong baseline binding activity to cell lines[81]. **b**. Full set of SPR sensorgrams, extended from Fig. 2b. For each interacting pair, the sensorgram on the left side shows kinetic binding measurements, and the sensorgram on the right shows equilibrium binding measurements. The protein used as the analyte is named on the left and the immobilized ligand is on the right. Association and dissociation constants from a 1:1 binding model fit (red line) are displayed where applicable on the kinetic traces. All analytes were resolved by gel filtration immediately prior to use in binding experiments to reduce the influence of protein aggregates, which otherwise can dominate binding kinetics (Supplementary Fig. 2). Some analytes show clear evidence of two-phase binding kinetics such as PTPRF and MCAM.

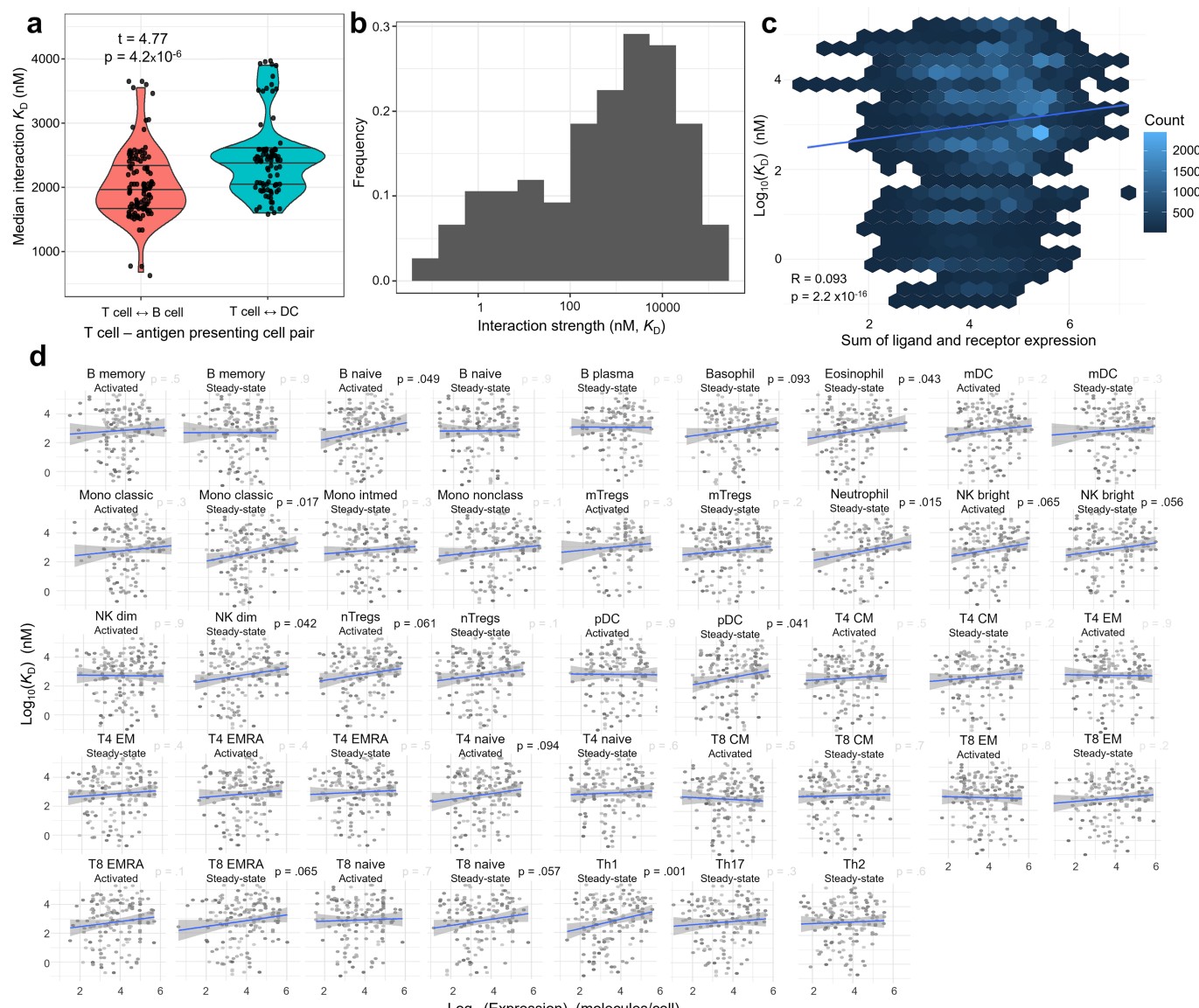

**Extended Data Fig. 5 | Distribution of immune receptor binding affinities.**
**a**. Antigen-presenting cells vary in their receptor contacts with circulating T lymphocytes. Each dot is a cell–cell pair comprising T cells (including all CD4 and CD8 subsets) with either a B cell (red distribution) or dendritic cell (blue distribution). Statistics are overlaid for a Welch's t-test. **b**. The overall distribution of immune receptor binding affinities is centred in the range of micromolar dissociation constants. All quantified immune interactions in our network are plotted as a log-scale distribution of binding dissociation constants ($K_D$). **c**. Cell surface receptor–ligand binding affinities weakly correlate with protein expression level. For all immune cell types measured by proteomics and all protein interaction pairs measured in our study, the equilibrium binding constant (y axis) is correlated to the summed expression of the two proteins. Weaker interactions (corresponding to higher $K_D$ values)

are associated with higher expression. Owing to the number of points, a hex plot is shown with the number of unique combinations of cell type and protein pairs represented by colour shading. A least-squares linear regression line is overlaid in blue. **d**. Within individual cell types, a surface protein's expression level weakly and variably correlates with its binding affinity. Instead of all pair combinations, each cell type measured by proteomics is individually presented. One point is placed for each detectable protein with an interaction in that cell type. In cases in which a single protein has multiple interactions of varying affinities, one point is drawn per binding interaction that protein participates in. Least-squares linear regression fit lines and 95% compatibility intervals are overlaid for each cell type. p-values for the Pearson correlation fit are accentuated for conditions where p < 0.10.

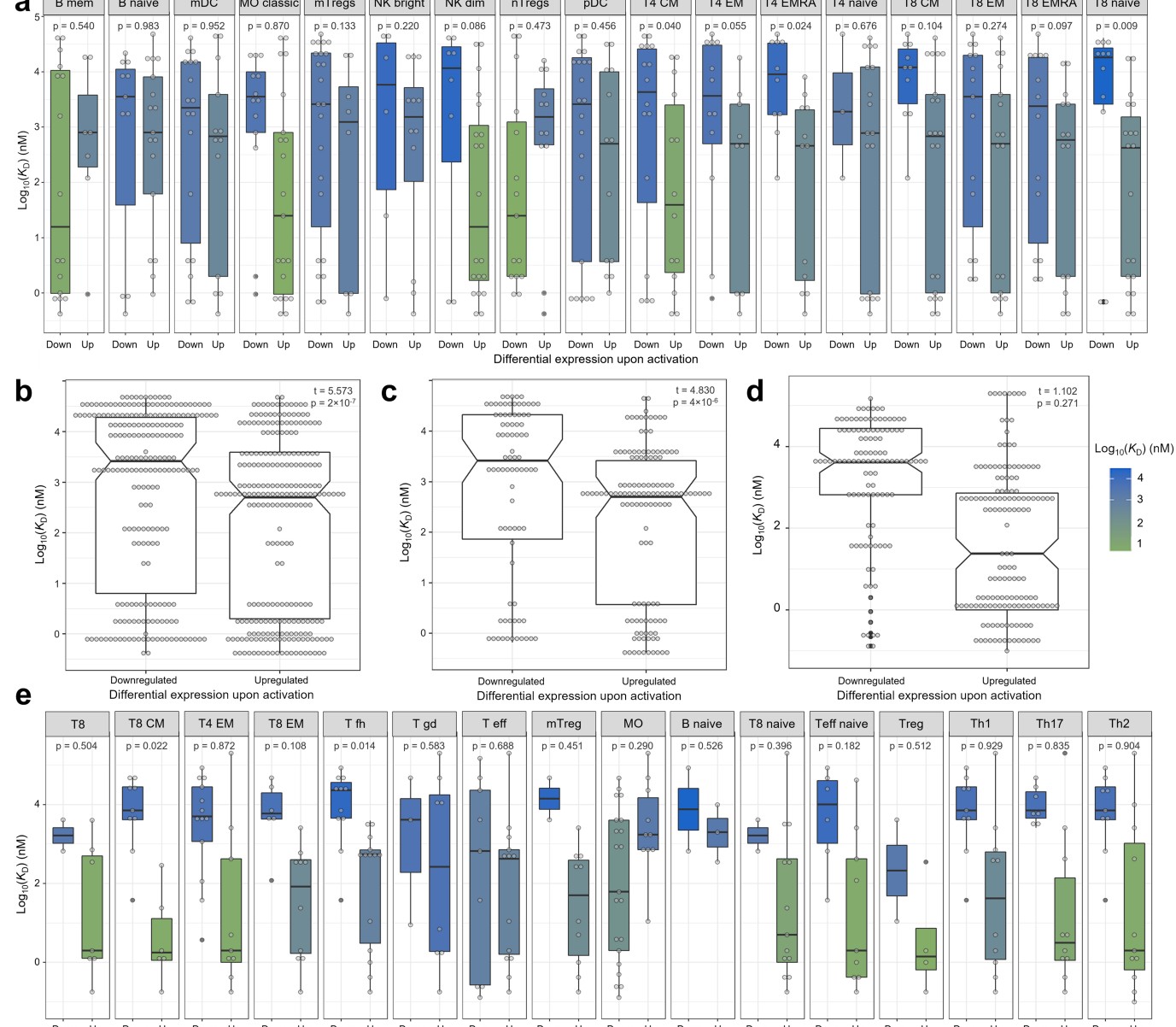

**Extended Data Fig. 6 | Average receptor affinity increases after immune activation. a**. Extended version of Fig. 2e, showing all cell subtypes identified in the original proteomics study[1] instead of higher-level cell-type categories. Each point represents the strength of a protein's interaction for proteins that are either differentially downregulated upon activation ("Down") or differentially upregulated upon activation ("Up"). To be classified as differentially expressed, the protein must have more than a 2-fold change upon activation. Data are shown as Tukey boxplots with Holm-corrected p-values calculated from a two-sided Welch's t-test. **b**. Combined analysis of all cell types for which paired activated and resting expression data is available. As in panel a, differential expression was defined by more than 2-fold changes upon activation (n = 4 blood donors). Statistics are overlaid for a two-sided Welch's t-test.

**c**. Same as panel b except setting as the criterion for differential expression that the protein must have a corrected p-value below 0.05 across the 4 proteomics replicates available per condition. Statistics are overlaid for a two-sided Welch's t-test. **d**. Same analysis as panel b performed on an independent dataset of RNA-seq measurements on sorted and stimulated immune cell populations[87]. Statistics are overlaid for a two-sided Welch's t-test (n = 4 blood donors). **e**. Same analysis as panel a, performed on an independent dataset of RNA-seq measurements (Calderon et al. 2019). Every leukocyte subpopulation measured in activated and resting states is show as a separate box. Data are shown as Tukey boxplots with Holm-corrected p-values calculated from a two-sided Welch's t-test.

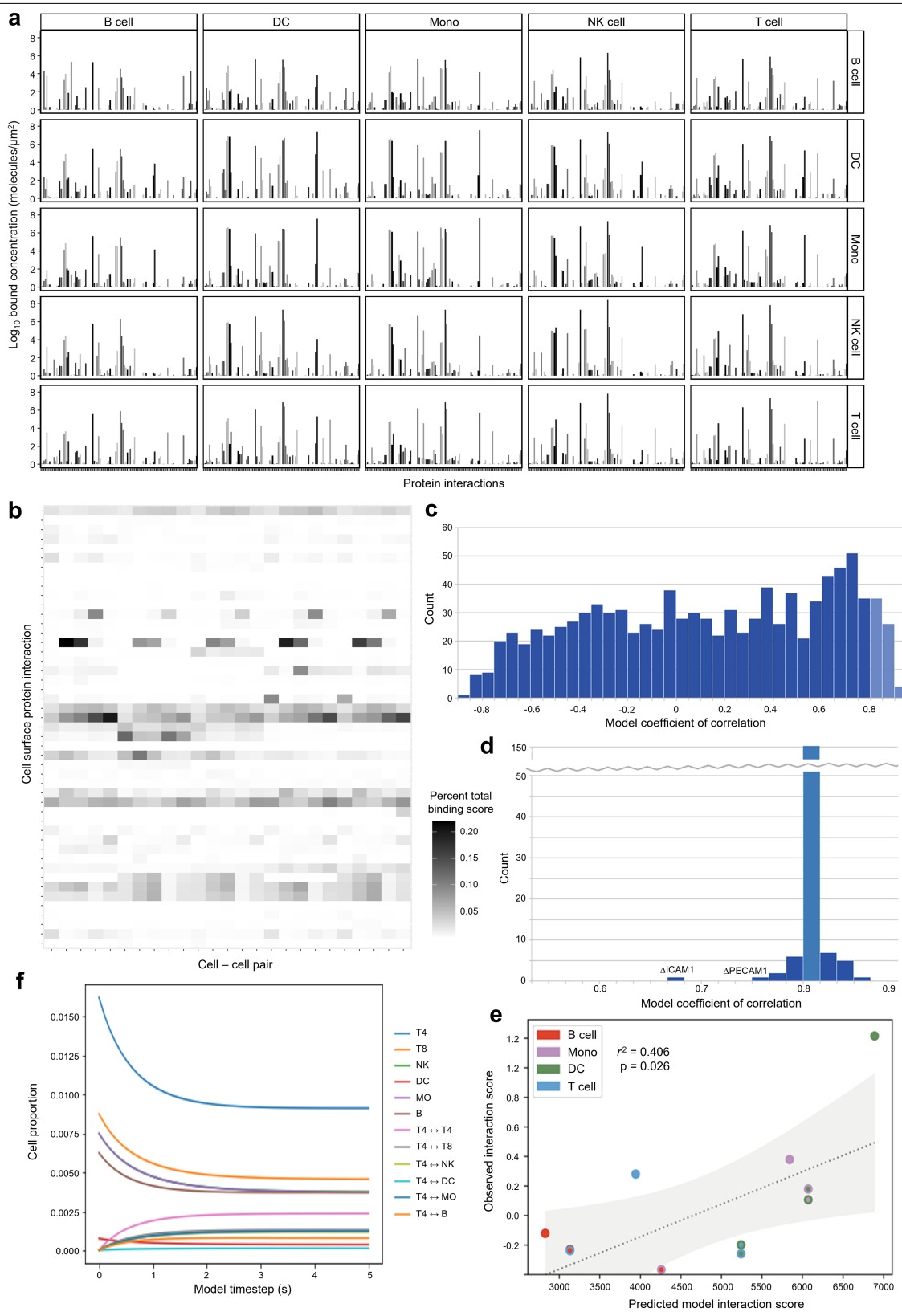

**Extended Data Fig. 7** | See next page for caption.

**Extended Data Fig. 7 | Internal representations of the quantitative cellular interaction model and additional validations. a**. Interaction spectra showing how individual receptor interactions (x axis) contribute to the overall connection strength between cell pairs. The model output per pair, which corresponds to the calculated density of proteins in a bound configuration at a theoretical equilibrium, is shown on the y axis in $\log_{10}$ scale. **b**. Relative contributions of different protein–protein interactions to a cell pair's overall connection scores. The colour shading indicates the percent of the total calculated interactions between a given cell pair (x axis) that are attributable to each specific protein–protein interaction (y axis). **c**. Null distribution of correlations to published data when the model's cell-surface protein interaction pairs are randomized. Histogram bins representing correlations equal to or greater than the fit of the true model are shaded in light blue after 1,000 random permutations were performed. **d**. Distribution of model correlation coefficients to previously published leukocyte binding data following a complete leave-one-out analysis of each surface protein. The histogram bin that includes the observed correlation in Fig. 2g is shaded in lighter blue. The two proteins leading to the greatest change in correlation (ICAM1 and PECAM1) are labelled. **e**. Model fits remain robust on independently measured datasets of leukocyte cellular contacts. Following an analogous approach to Fig. 2g, the kinetic model's predictions for baseline leukocyte interaction rates were compared to empirical data generated during our pharmacoscopy experiments. Only negative control conditions treated with PBS instead of recombinant protein were included, with interaction s cores calculated following the same methodology as the previously published study[25] described in Fig. 2g. Shading indicates the 95% compatibility interval of the least-squares linear regression fit. **f**. Differential equation model output simulating cell pairs reaching a binding equilibrium. Each colour is either a cell type in an unbound state or a cell pair. The absolute proportions in blood (y axis) were tracked over an arbitrary time scale (x axis) until equilibrium is reached.

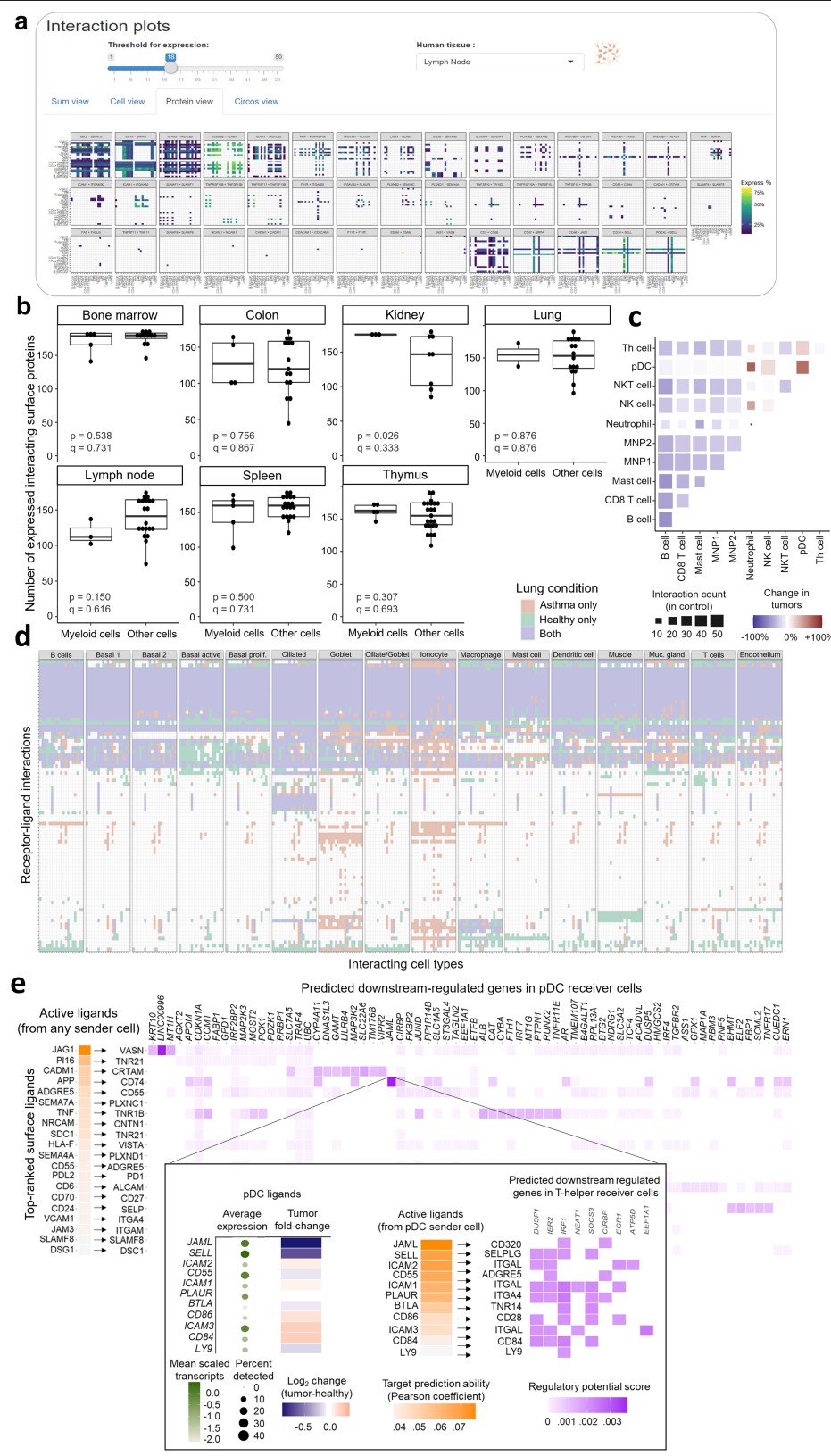

**Extended Data Fig. 8 |** See next page for caption.

**Extended Data Fig. 8 | Integrating single-cell expression atlases with cellular interactions. a**. Interactively searching immune interactions in human tissues through our web tool. This screenshot depicts the core features of the website, including a drop-down menu to select one of eight different tissue datasets, adjustable sliders for expression cut-off thresholds when determining interactions, and multiple tabs offering different kinds of views for both cellular and molecular interaction types. **b**. Myeloid-lineage cells do not express any greater quantity of interaction-capable surface proteins than non-myeloid cells. Each dot represents a specific measured cell type in the indicated dataset, which are grouped into categories as myeloid or non-myeloid (x axis). The absolute number of cell-surface proteins for which at least one interaction is annotated with another cell-surface protein (y axis) is compared between categories. p-values from a two-sided Welch's t-test are shown alongside their corresponding false discovery rate-corrected q values. Tissues match those shown in Fig. 3c. **c**. Changes in cellular interaction frequencies among immune cells isolated from cancerous versus healthy kidney implicate cellular contacts with potential relevance to pathology. The total number of interactions detected in a single-cell sequencing dataset of paired healthy kidney and kidney tumours are compared across immune populations. **d**. Comparisons of interactions detected in paired samples of healthy and diseased tissue can suggest functional targets. Human lung tissue from healthy donors and patients with asthma were processed in identical ways through our web tool's functions. At the indicated expression cut-off which requires the mRNA encoding an interacting surface protein be detected in at least 5% of cells from a given cell type, interactions between cells were categorized based on whether they were present in both healthy and asthmatic samples (purple), only healthy (green), or only asthma (red). Although more qualitative than differential expression based tests, this approach may have utility in conducting more sensitive exploratory analyses of interaction sets. **e**. Intercellular signalling pathways in tumour-infiltrating immune cells inferred by NicheNet analysis. Two analyses are shown on a single-cell RNA-seq dataset of immune cells isolated from kidney for cell-surface signals that differentially regulate gene expression in plasmacytoid dendritic cells (pDCs) and helper T cells within kidney tumours when compared to adjacent healthy kidney tissue. In the larger box, intercellular signals being received by pDCs are shown (left), matched to genes inferred to be regulated by those signals. In the smaller inset box, the finding that tumour pDCs upregulate JAML is expanded by analysing pDC communication specifically with helper T cells. Targets in our gene regulatory analysis were filtered to exclude those which recurred non-specifically in more than half of all cases.

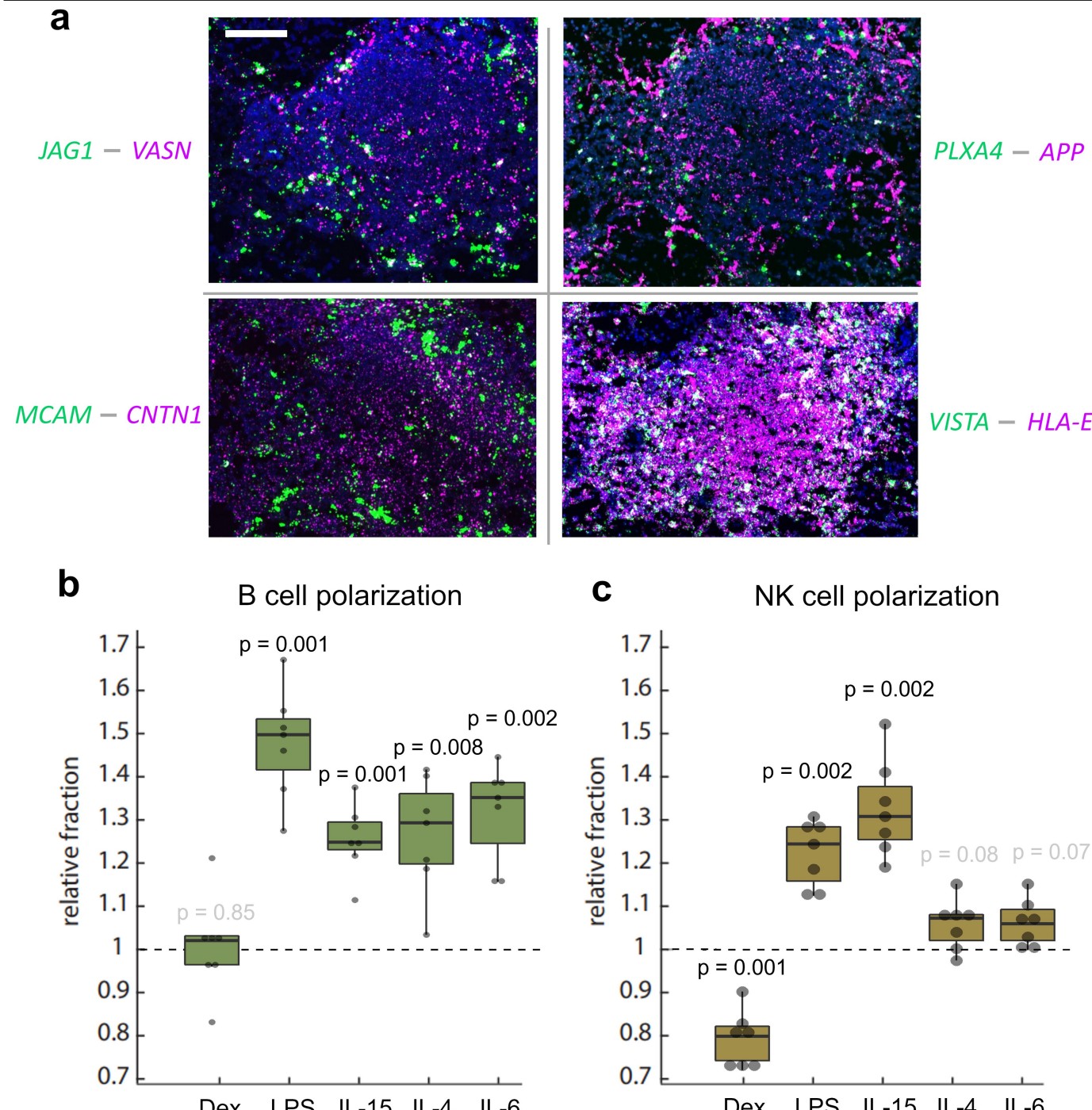

**a**

JAG1 — VASN

PLXA4 — APP

MCAM — CNTN1

VISTA — HLA-E

**b** B cell polarization

**c** NK cell polarization

**Extended Data Fig. 9 | Microscopy readouts identify that receptor binding partner transcripts are colocalized in human lymph nodes, and that polarization of B and NK cells serves as a marker of classical activation pathways. a**. Single-molecule fluorescent in situ hybridization in human lymph node demonstrates that cells transcribing genes encoding surface proteins found to physically interact in biochemical assays are also physically colocalized in the lymph node. A lymphoid follicle is shown for 4 different transcript pairs encoding proteins that directly interact. Each experiment was repeated on tissue from two donors. The scale bar is 100 μm and applies to all images. **b**. Polarization of B cells relative to PBS-treated controls after treatment with cytokines and other immunomodulatory molecules.

As expected IL-4 and IL-6 are strong B-cell specific activators[82,83], whereas IL-15 and LPS activate both NK and B cells[84]. Data are shown as Tukey boxplots with Holm-corrected p-values calculated from a one-sided t-test. n = 7 blood donors. **c**. Polarization of NK cells relative to PBS-treated controls after treatment with cytokines and immunomodulatory molecules. As has been reported for NK cell activation, IL-15 invokes the strongest activation of NK cells[85], whereas there are no effects from IL-4 and IL-6. The NK cells are also inhibited by the steroid dexamethasone (Dex), consistent with known pharmacology[86]. Data are shown as Tukey boxplots with Holm-corrected p-values calculated from a one-sided t-test. n = 7 blood donors.

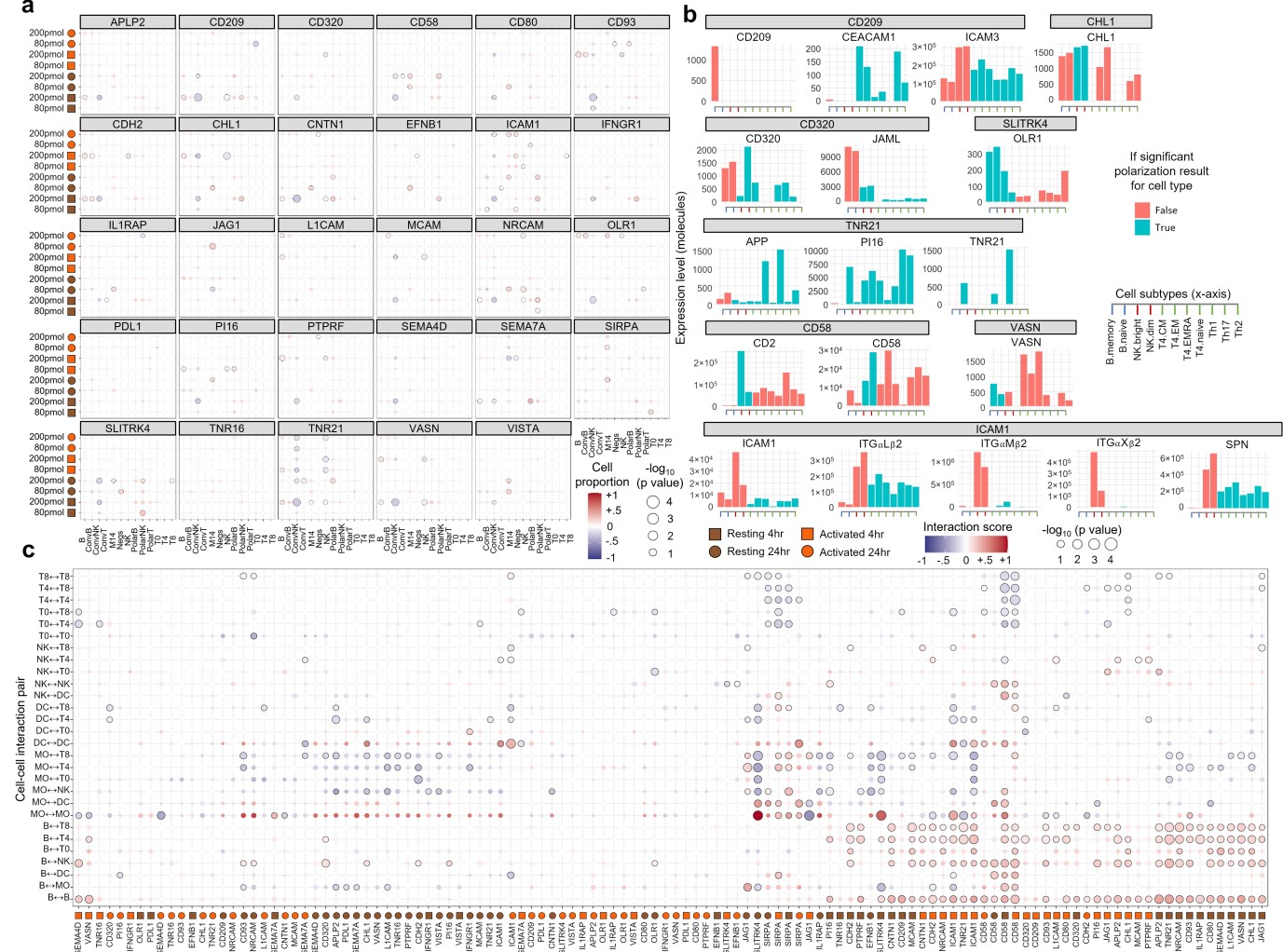

**Extended Data Fig. 10 | High-content microscopy datasets show human leukocyte phenotypes after infusion of purified proteins. a.** Full dataset of changes to cellular state proportions, extended from Fig. 4b. The proportions of measured cell types and cell states (x axis) are compared across different protein doses, timepoints, and with or without background LPS activation (y axis). Points are sized by their adjusted p-values, and shaded to show relative change compared to controls. n = 5 wells. **b.** Observed immunomodulatory phenotypes match the expression profiles of the cell types expressing the applied protein or its identified binding receptor. For every statistically significant phenotype upon protein application that is described in Fig. 4b (grey top bars), the protein itself and all its identified interaction partners (columns) had their expression in lymphocyte populations (x axis) compared according to quantitative proteomics measurements (y axis). The lymphocyte populations are coloured blue if they match the cell type for which the significant phenotype was found. **c.** Clustering on the full set of cell-to-cell connectivity changes identifies recurring modules of cellular shifts. All protein conditions and timepoints (y axis) were hierarchically clustered by the complete linkage method. As above, changes in each cell pair's interactions (y axis) are indicated by the provided colour scale and sized based on their adjusted p-values. n = 10 wells.

# Reporting Summary

## Statistics

For all statistical analyses, confirm that the following items are present in the figure legend, table legend, main text, or Methods section.

| n/a | Confirmed | |
|---|---|---|
| ☐ | ☒ | The exact sample size (*n*) for each experimental group/condition, given as a discrete number and unit of measurement |
| ☐ | ☒ | A statement on whether measurements were taken from distinct samples or whether the same sample was measured repeatedly |
| ☐ | ☒ | The statistical test(s) used AND whether they are one- or two-sided *Only common tests should be described solely by name; describe more complex techniques in the Methods section.* |
| ☒ | ☐ | A description of all covariates tested |
| ☐ | ☒ | A description of any assumptions or corrections, such as tests of normality and adjustment for multiple comparisons |
| ☐ | ☒ | A full description of the statistical parameters including central tendency (e.g. means) or other basic estimates (e.g. regression coefficient) AND variation (e.g. standard deviation) or associated estimates of uncertainty (e.g. confidence intervals) |
| ☐ | ☒ | For null hypothesis testing, the test statistic (e.g. *F*, *t*, *r*) with confidence intervals, effect sizes, degrees of freedom and *P* value noted *Give P values as exact values whenever suitable.* |
| ☒ | ☐ | For Bayesian analysis, information on the choice of priors and Markov chain Monte Carlo settings |
| ☒ | ☐ | For hierarchical and complex designs, identification of the appropriate level for tests and full reporting of outcomes |
| ☒ | ☐ | Estimates of effect sizes (e.g. Cohen's *d*, Pearson's *r*), indicating how they were calculated |

*Our web collection on statistics for biologists contains articles on many of the points above.*

## Software and code

Policy information about availability of computer code

| | |
|---|---|
| Data collection | Optical absorbance measurements were collected on a Tecan Spark plate reader. Surface plasmon resonance data were collected on a BIAcore 8k instrument. Flow cytometry data were collected on Becton-Dickinson LSR Fortessa flow cytometer. Microscopy images were taken using a Perkin Elmer Opera Phenix confocal microscope. |
| Data analysis | All custom analysis code is written in R and Python, which are open projects without license restrictions. R is version 4.0.3 and Python is version 3.7.4. The following packages were also used. For R: Seurat (version 3.1.5), shinyCircos (version 1.0.0), PRROC (version 1.3.1), NicheNet (version 1.0.0), DEseq2 (version 1.30.0), igraph (version 1.2.5). For Python: Scanpy (version 1.4.5), PySB (version 1.11). FlowJo (version 10.6.1) was used for drawing gates on flow cytometry data. For microscopy image analysis, both CellProfiler (version 2) and MATLAB Neural Network Toolbox (version R2020a) were used. |

For manuscripts utilizing custom algorithms or software that are central to the research but not yet described in published literature, software must be made available to editors and reviewers. We strongly encourage code deposition in a community repository (e.g. GitHub). See the Nature Portfolio guidelines for submitting code & software for further information.

## Data

Policy information about availability of data

All manuscripts must include a data availability statement. This statement should provide the following information, where applicable:

- Accession codes, unique identifiers, or web links for publicly available datasets
- A description of any restrictions on data availability
- For clinical datasets or third party data, please ensure that the statement adheres to our policy

All data files are available in the Github repository associated with this manuscript (https://github.com/jshilts/shilts-et-al-2022-immunoreceptors). Accession codes are provided by the UniProt database (https://uniprot.org/).

# Field-specific reporting

Please select the one below that is the best fit for your research. If you are not sure, read the appropriate sections before making your selection.

☒ Life sciences          ☐ Behavioural & social sciences          ☐ Ecological, evolutionary & environmental sciences

For a reference copy of the document with all sections, see nature.com/documents/nr-reporting-summary-flat.pdf

# Life sciences study design

All studies must disclose on these points even when the disclosure is negative.

| | |
|---|---|
| Sample size | For automated microscopy experiments, a sample size of 5 per concentration per protein condition was selected as the largest number of replicates that could be fit onto the screening plates run for the platform. This number is equal to or greater than sample sizes previously published as sufficient for this method, e.g. DOI 10.1038/NCHEMBIO.2360. For binding screens, two replicates for each bait-prey orientation were done based on calculations for ensuring any rare false-positive signals on an individual replicate could be corrected, as informed by our earlier false positive rate benchmarking. |
| Data exclusions | No binding screen measurements were omitted. For specific analyses such as measuring protein-protein interaction signals from screening independent of known carbohydrate-binding lectin receptors, the criteria used for defining that excluded group are explained in the methods. Automated microscopy microscopy measurements of non-classical monocytes were excluded because so few cells were detected that they were not possible to analyze (e.g. cell counts jumping between 0 and 1 resulting in undefined fold-changes). |
| Replication | All of the novel interactions we describe in our study have been replicated by multiple independent approaches, as described in the text and summarized in the Extended Data figures. Binding assays measured two separate orientations for each protein, with two of these complete binding assays being done for every interaction. Every positive signal identified was then verified by binding onto human cell lines and through biophysical approaches verifying the binding was saturable and measuring the kinetics of binding. |
| Randomization | The order of all proteins in the systematic binding assays were randomized by a computer before screening. The positions of every condition in the automated microscopy experiment was also randomized on each plate. |
| Blinding | Investigators were blinded to the identities of samples during interaction screening. For other experiments such as surface plasmon resonance measurements, the samples could not be blinded. |

# Reporting for specific materials, systems and methods

We require information from authors about some types of materials, experimental systems and methods used in many studies. Here, indicate whether each material, system or method listed is relevant to your study. If you are not sure if a list item applies to your research, read the appropriate section before selecting a response.

## Materials & experimental systems

| n/a | Involved in the study |
|---|---|
| ☐ | ☒ Antibodies |
| ☐ | ☒ Eukaryotic cell lines |
| ☒ | ☐ Palaeontology and archaeology |
| ☒ | ☐ Animals and other organisms |
| ☐ | ☒ Human research participants |
| ☒ | ☐ Clinical data |
| ☒ | ☐ Dual use research of concern |

## Methods

| n/a | Involved in the study |
|---|---|
| ☒ | ☐ ChIP-seq |
| ☐ | ☒ Flow cytometry |
| ☒ | ☐ MRI-based neuroimaging |

# Antibodies

| Antibodies used | For leukocyte immunostaining, the following antibodies raised against human epitopes were used: anti-CD3 AF647 (Biolegend, Clone UCHT1, cat. #300416, lot #B284504), anti-CD4 FITC (Biolegend, Clone SK3, cat. #344604, lot #B244280), anti-CD8 PE (BD Biosciences, Clone SK1, cat. #345773, lot #106349), anti-CD19 FITC (Biolegend, Clone SJ25C1, cat. #363008, lot #B290869), anti-CD56 PE (Beckman Coulter, Clone N901, cat. #A07788, lot #49), anti-CD16 PE (Biolegend, Clone 3G8, cat. #302008, lot #B290852), anti-CD14 AF647 (Biolegend, Clone HCD14, cat. #325612, lot #B260484), anti-CD20 (BD Biosciences, Clone 2H7, cat. #555623, lot #8260745). For protein normalization ELISAs, the following antibodies were used: anti-rat Cd4 (Clone OX68, purified from a hybridoma provided by Neil Barclay, University of Oxford), and anti-mouse IgG alkaline phosphatase (Sigma, cat. #A9316). |
|---|---|
| Validation | Every antibody used was validated for specificity to their human cell-surface targets by their manufacturers, and all pharmacoscopy antibodies have been established for use as a panel by a previous study (Severin et al., 10.1101/2021.12.03.471105). The antibodies against CD3, CD8, CD14, and CD20 were also validated by their manufacturers for immunocytochemistry, while all remaining antibodies were validated by their manufactures for flow cytometry. The OX68 antibody has further had its specificity validated by prior studies (e.g. 10.1186/1471-2091-6-2). |

# Eukaryotic cell lines

Policy information about cell lines

| Cell line source(s) | Both HEK293 cell lines were graciously provided by Yves Durocher (National Research Council, Canada). |
|---|---|
| Authentication | HEK293 cell lines were not authenticated before this study. |
| Mycoplasma contamination | All cell lines were regularly tested for mycoplasma (Surrey Diagnostics, UK) and found to be negative all throughout these experiments. |
| Commonly misidentified lines (See ICLAC register) | Our cell lines are not listed as commonly misidentified. |

# Human research participants

Policy information about studies involving human research participants

| Population characteristics | Blood donors were healthy Swiss residents, with hemoglobin levels and blood pressure readings within normal reference ranges. The characteristics of the deceased organ donors for lymph node specimens were not recorded. |
|---|---|
| Recruitment | Anonymous organ donors were identified by the Cambridge Biorepository for Translational Medicine (CBTM). Blood donations were collected by Blutspende Zürich. |
| Ethics oversight | Tissue collection was overseen by the Cambridge Biorepository for Translational Medicine (CBTM) with full approval from the National Research Ethics Service Committee East of England - Cambridge South (15/EE/0152). CBTM operates in accordance with UK Human Tissue Authority guidelines. All tissues were taken from deceased organ transplant donors with informed consent from the donor families. Blood collection was overseen by the cantonal ethical committee of Zurich (KEK Zurich, BASEC-Nr 2019-01579) with consent from the blood donors. |

Note that full information on the approval of the study protocol must also be provided in the manuscript.

# Flow Cytometry

## Plots

Confirm that:

☒ The axis labels state the marker and fluorochrome used (e.g. CD4-FITC).

☒ The axis scales are clearly visible. Include numbers along axes only for bottom left plot of group (a 'group' is an analysis of identical markers).

☒ All plots are contour plots with outliers or pseudocolor plots.

☒ A numerical value for number of cells or percentage (with statistics) is provided.

## Methodology

| Sample preparation | HEK293 cells were incubated with fluorescent tetramers loaded with a recombinant protein extracellular domain. Only a single fluorochrome was use d, hence some of the points listed above are not applicable. |
|---|---|
| Instrument | BD LSRFortessa Flow Cytometer |
| Software | FlowJo (version 10.6.1) was used for drawing gates. The R package ggcyto (version 1.17.0) was used for drawing histogram plots of the gated data. |

| Cell population abundance | No sorting was performed. All cells began as homogeneous HEK293 cultures, ensuring their purity. At least 10,000 cells were measured for each condition. |
|---|---|
| Gating strategy | Cell-sized events were gated based on SSC-A / FSC-A as shown in the supplementary example gates. Singlet cells were selected based on FSC-W / FSC-A. |

☒ Tick this box to confirm that a figure exemplifying the gating strategy is provided in the Supplementary Information.

