## [Peer Review File · Nature]

Manuscript Title: A physical wiring diagram for the human immune system.

Reviewer Comments & Author Rebuttals

Reviewer Reports on the Initial Version:

Referees' comments:

Referee #1 (Remarks to the Author):

In this manuscript Shilts et al. provide a systematic and quantitative view of the cell-surface proteins that enable immune cells to dynamically wire their interactions. This was achieved using a high-throughput surface receptor protein-protein interaction (PPI) assay called SAVEXIS. The authors then independently validated the biophysical parameters for 28 new interactions not previously identified using any PPI method.

To validate these 28 new PPIs, each of them was assessed by orthogonal approaches, such as by testing protein binding to receptors displayed on the human cell surface by cDNA transfection followed by rounds of surface plasmon resonance (SPR). Importantly, all of the 28 newly identified interactions were validated by at least one additional PPI method. From this integrative interactome mapping approach the authors then constructed a mathematical model that predicts the behavior of collections of leukocytes from first principles.

Specific points:

1. Their high-throughput SAVEXIS screening approach works very well and their SPR data shows direct saturable binding. Their system is scalable and can be applied in many different areas of biology (e.g., to map new host-pathogen interactions). However, the authors should better explain in a revised manuscript how exactly this method can be applied to type I, type II and type III integral membrane proteins as well as secretable proteins.
2. Figure 3 – interactome validation summaries – the authors have to better explain the ROC curves of their screen performance (panel b) as well as overview evidence for 28 newly identified interactions which are NOT clearly visible from panel c.
3. The authors should better explain their mathematical model that integrates quantitative proteomics expression, binding kinetics and published cell parameters to summarize the contributions of individual PPIs to a given cellular interaction. As a non-mathematician, it is very difficult to understand this part from the current version of this manuscript.
4. Figure 3e: could the authors provide in situ hybridization images of JAG1 and VASN?

Referee #2 (Remarks to the Author):

Review of Shilts et al "A physical wiring diagram for the human immune system"

Shilts et al start with a large biochemical screen of immune surface receptors. A full set of annotated receptors is classified into 6 structural groups for biotinylation and conjugation to tetrameric streptavidin. With the purified components an all vs all sort of elisa assay was performed and very specific results were obtained. This is evidenced by the fact that more than half of the PPIs in the network remain isolated pairs of interacting proteins and that almost all (25/28) PPIs could be validated with at least one of three attempts: quantitative repetition, SPR or in cell binding experiments. It is clear that the vast majority of PPIs found in the screen are "expected", i.e. find support in the literature. This for sure is a unique, systematic high-quality approach to immune receptor PPIs.

An important further aspect of the PPI approach is that quantitative data, apparent affinities, are obtained for the (new?/ vs literature) PPIs. A mass equilibrium model that uses receptor expression and the PPI affinities is set up to predict cell-cell interactions. This develops into an interesting line of results presented as an interactive atlas of immune cell connectivity. Single cell expression data of 11 (immune) cell-types in four tissues reveal differential cell-cell connectivity (physical connection maps) together with receptor expression and PPI connectivity that can be interactively queried. The key finding here is that myeloid cells behave as hub in the cell-cell connectivity maps. A second data set, derived from transcriptomics of human lymph node confirms physical localisation of interacting leukocytes in vitro. This dynamic cell-cell contact maps can be a very useful resource and demonstrate an innovative way of combining expression and PPI information towards biological insight.

The last part is based on high-content microscopy for measuring leukocyte activation, where their purified protein constructs are used as perturbation. Put simply, soluble ligands are used to perturb cell-cell interactions and polarisation. This is a screen as such, however the authors state that the results are in line with the results seen for leukocyte interaction changes. Anyway Figure 4 e shows that cell-cell interactions match their predictions for a set of 8/9 proteins in tendency. At this point I have to apologize and state that immune cell-cell contacts are not my core expertise, so I feel only medium confident in judging the effects of receptor proteins on leukocyte activation. Anyway, leveraging the high-quality biochemical PPI screen into an integrated view of immune cell-cell contracts is a major advance.

Points for consideration:

- *) Figure 1c is of insufficient quality, so that neither PPIs nor literature information can be read, the network is fine however and shows the PPIs.
- *) PPI data: in essence the authors elude to only a single PPI involving the orphan receptor VISTA.
- *) Literature PPIs: The literature curated PPIs remain somehow ill-defined throughout, even though I think the comparison is very much supporting the approach "captured a majority of all previously reported interactions", numbers would be good.
- *) How much do the new PPIs (only 28 from many) contribute to the cell-cell PPI model? What happens if the newly found PPIs are omitted from the model?
- *) The cell-cell interaction prediction model needs to be scrutinized with respect to PPI variation. How robust are cell-cell contact predictions based on mass action of sets of expressed receptors? Figure 2g summarizes predictive value of the model, however no error estimates are given. Figure 8 c clearly suggest that some interactions are more some are less important / specific.
- *) Figure 4 is very descriptive. For example, CD58 inhibits most cell-cell contacts and therefore completely abolishes what? In contrast VISTA inhibits 2 cell-cell type contacts only in resting conditions after 24 hours. Does this relate to the interactions with HLA/E or is it more complicated?
- *) The manuscript, even in its short from, can maybe more explicitly point towards findings rather than saying: "Among our other findings are ", "we have been able to derive other interesting biological properties of the network", "allows multiple kinds of analysis". The reader is under the impression to miss/not understand something important what is the meaning of a quilt plot?

Referee #3 (Remarks to the Author):

Using a high-throughput screening method, Shilts and colleagues expand the known leukocyte surface protein interactions by about 20%, including ligands for a previously orphan receptor. They validated their findings using surface plasmon resonance, which allowed them to also quantify the strength of the binding. Pooling this acquired data with existing literature, they created a substantial resource of quantitative surface protein interactions in the human immune system, for which they also created an online interface.

Major points:

- The authors propose a simple model for quantifying the strength of the cellular interaction in terms of the quantity of interacting molecules expressed and the strength of their interaction. The validation was coarse using published literature (Fig. 2g) and experiment (Fig. 4e). This is one of the most substantial points of the ms, it could be highlighted more and the authors could strive to offer a stronger validation.
- The authors found several interesting biological leads using the model (e.g. immune cell activation on average decreases their potential interaction strength; DC & NK cells show increased interaction strength in kidney tumors etc) though all of these findings are interesting; their biological significance is unclear and they would require substantial further investigation.
- In the last part of the manuscript the authors use a novel high-content microscopy approach for validation – this is the most extensive validation in the paper however, however the approach is the main subject of an independent work.
- The online resource is not online yet, so no way to assess it.

Minor points:

1. The panel in Fig. 3A should not be introduced as revealing 'connectivity patterns' or 'organizing principles'. It merely shows the interactions, and, to a human-reader, this panel is almost non-informative.
2. Illegible, minute fonts are very prevalent throughout the figures.

Author responses : Nature manuscript 2021-06-10485

We thank the reviewers for their valuable time to comment on our paper. We are appreciative of their positive and constructive feedback that has allowed us to improve both the quality of our results and the clarity of our manuscript. Below we provide a point-by-point response to each comment we received, together with details of how we have performed further experiments and analyses to address each and incorporate them into the revised version of the manuscript.

For convenience, we have also summarized the major changes and additions to the manuscript in the non-exhaustive listing below :

- performed 8 additional validation tests to support our mathematical model, including verifying statistical assumptions, characterizing robustness, and benchmarking against new experimental data (featured in the revised or new Fig 2g, Fig S7c, Fig S7d)
- improved our mathematical model's accuracy by integrating feedback from the reviewers (revised Fig 2g, revised Fig 4e)
- completed new single-molecule RNA hybridization experiments on human lymphoid tissue sections to confirm our earlier analysis of colocalized cell pairs expressing receptors that we found to interact *in situ* (new Fig 3e and new Fig S9a).
- applied a new method for integrating our receptor interaction network with downstream signaling pathways inferred from public databases (new Fig S8d)
- included new analyses for explaining functional changes seen in our pharmacoscopy data, as well as independent validation of the shift we observed in activated immune cells' receptor affinity profile (new Fig S6d, S6e).
- expanded the description of our modeling work and interaction network curation, with 3 additional pages at to the Supplementary Text, an additional schematic (new Fig S1a), and new table (Supplementary Table 3)
- updated all figures in the manuscript to enlarge their fonts and ensure all parts of the figures satisfy *Nature's* figure guidelines (every main and extended data figure adjusted).

We believe that it is worth emphasizing that the central findings of our manuscript are centered around the receptor interactions that we have discovered and characterized. We are grateful for the comments by the reviewers to polish and reinforce our analyses stemming from these core discoveries, which have significantly improved the quality of our paper.

Referee #1 (Remarks to the Author):

In this manuscript Shilts et al. provide a systematic and quantitative view of the cell-surface proteins that enable immune cells to dynamically wire their interactions. This was achieved using a high-throughput surface receptor protein-protein interaction (PPI) assay called SAVEXIS. The authors then independently validated the biophysical parameters for 28 new interactions not previously identified using any PPI method.

To validate these 28 new PPIs, each of them was assessed by orthogonal approaches, such as by testing protein binding to receptors displayed on the human cell surface by cDNA transfection followed by rounds of surface plasmon resonance (SPR). Importantly, all of the 28 newly identified interactions were validated by at least one additional PPI method. From this integrative interactome mapping approach the authors then constructed a mathematical model that predicts the behavior of collections of leukocytes from first principles.

We thank the reviewer for their thorough and positive appraisal of our manuscript, including noting the care we have taken in screening, validating, and quantifying each new interaction in our interactome.

Specific points:

1. Their high-throughput SAVEXIS screening approach works very well and their SPR data shows direct saturable binding. Their system is scalable and can be applied in many different areas of biology (e.g., to map new host-pathogen interactions). However, the authors should better explain in a revised manuscript how exactly this method can be applied to type I, type II and type III integral membrane proteins as well as secretable proteins.

We thank the referee for their positive comments about our method. It is a great suggestion to add additional detail on the versatility of the approach, since we agree the scalability and broad-applicability of SAVEXIS are two important strengths that perhaps were not sufficiently emphasized in the original manuscript. Although we are limited in text space, we have expanded Extended Data Figure 1 to feature a new schematic that details how the method can be applied to all the different membrane protein architectures that the reviewer asked about.

Given these constraints, we should add that it is our intention to submit a more lengthy account of practical step-by-step guidance for implementing the method in a suitable forum such as *Nature Protocols* with the aim of making this technique more widely

accessible. As the reviewer correctly points out, it is very likely that this versatile approach will have applications in many different fields. Indeed we are now regularly employing this technique in our laboratory, with recent success in projects studying surface proteins both in the brain and on pathogen species. We believe the expanded explanations we have made for these revisions are a first step toward making the method easier to adopt.

Figure R1. Schematic showing plasmid construct design to produce soluble ectodomains for different architectural classes of membrane protein (now part of the revised Extended Data Figure 1). Type I transmembrane proteins, GPI-anchored proteins, and secreted proteins are encoded as complete ectodomains including their endogenous signal peptide and fused at their c-terminus with recombinant tags including the biotin acceptor site. Type II proteins have a reversed orientation, beginning with an exogenous signal peptide, tags, and then the full ectodomain fused at the c-terminus. Type III proteins are encoded similarly to type I proteins except with an exogenous signal peptide. Multi-pass transmembrane proteins employ the same design except only the largest contiguous extracellular region is included. Designs for multi-subunit receptor complexes involve co-expressing the two chains of the receptor. Only one of these chains is tagged and biotinylated as in the prior designs, while the other chain is left untagged. The exact constructs used for each chain reflects the topology of each subunit: for example, HLA complexes which involve a type I subunit and a secreted subunit are produced both with the type I/GPI/secreted construct, while NKG2 complexes are comprised of type II subunits and thus use the type II construct.

2. Figure 3 – interactome validation summaries – the authors have to better explain the ROC curves of their screen performance (panel b) as well as overview evidence for 28 newly identified interactions which are NOT clearly visible from panel c.

We have expanded our methods section to describe in more detail how the ROC curves are calculated, and their interpretation as a graphical summary of screen performance against benchmarks of previously-published interactions. In brief, ROC and precision-recall curves provide a method to comprehensively summarize how well an assay (which gives a quantitative signal over a continuous range) accords with a binary benchmark (which gives only discrete categories of ‘interacts’ or ‘does not interact’). It has been frequently used for benchmarking large-scale interactome assays (see Braun *et al.* 2009 “An experimentally derived confidence score for binary protein-protein interactions” and other studies following the guidelines it set for the field such as Trepte *et al.* 2018 and

Trigg *et al.* 2017). The red shading shows each threshold at which the quantitative screen data could be converted into a binary classification, with the accompanying position on the x and y axes showing how well the screen, at that threshold, covered interactions in the positive reference set and discriminated them from interactions in the negative reference set. Because there are no universally agreed guidelines for how to set these reference sets in the literature (for example, “is a negative reference set defined based on publications claiming non-binding or inferring non-binding by randomly selecting pairs”, and “is a positive reference set based only on any literature claim or only publications with independently-validated or high-quality support?”), In Extended Data Figure 3b, we show all of these possible permutations in an effort to be as comprehensive as possible. Regardless of the benchmark set, our screen consistently showed excellent performance characteristics, recapitulating most known interactions at very low false positive rates.

We have also modified Extended Data Figure 3c in several different ways to try to make the summary data more visually clear. First, we expanded the text labels to replace abbreviations and add additional words to descriptions so that it is more obvious what each category refers to. Second, we segmented each row of validation measurements into groupings with subheadings to explain the source of each measurement. Third, we have expanded the legend to this figure panel to elaborate more on the different data sources being summarized in this panel.

Revised Extended Figure 3. Panel c has been modified to incorporate the reviewer’s suggestions to make the labels and classifications clearer, with changes to all text labels and the visual grouping of results.

3. The authors should better explain their mathematical model that integrates quantitative proteomics expression, binding kinetics and published cell parameters to

summarize the contributions of individual PPIs to a given cellular interaction. As a non-mathematician, it is very difficult to understand this part from the current version of this manuscript.

This is an excellent suggestion to make our explanation of the mathematical model more intuitive so that it is more widely accessible. In our revised manuscript, we have added 3 pages of text and diagrams as a preface before our more formal Supplementary Text on the derivations of equations and mathematical details. The first of these diagrams (Figure R2) illustrates how the core intuition of the model is that we can combine measurements of surface protein concentrations (from previously public expression datasets and cell surface area measurements) with our systematic interaction network and binding quantification (from our study and curation). The law of mass action provides a simplistic yet principled basis by which to aggregate these values into predictions about intercellular contacts.

Figure R2. New illustration showing the basic principles of calculating cellular interaction scores through our modeling approach (added to Supplementary Text). Prior to the more formal definitions of equations, this illustration provides a simplified overview of how the model combines binding and expression data.

The extended preface we have added also illustrates how this more mathematically-guided approach avoids issues inherent to prior more coarse attempts at calculating receptor interactivity from expression and protein-protein interaction data (Figure R2). In the hypothetical example we illustrate, a system of 3 cell types could have identical numbers of complementary receptor-ligand pairs between them, and based on simple counts of expression values without factoring in affinity, a wrong estimation for the ranking of cellular contacts would be made.

Figure R3. New illustration in our revised manuscript to provide a more intuitive representation of the factors integrated in our mathematical model (added to Supplementary Text). As a visual representation of the relative roles of expression level and binding affinity, we provide an example that shows how affinity and expression can be combined to produce more accurate predictions than more rudimentary alternative approaches.

4. Figure 3e: could the authors provide *in situ* hybridization images of JAG1 and VASN?

We appreciate this constructive suggestion to reinforce the findings we report in our manuscript by an alternate method. In our revised manuscript we have added new single-molecule fluorescent *in situ* hybridization data that we generated on human thoracic lymph node tissue sections. Following the advice of the reviewer, we began by testing JAG1 and VASN, as they were the example that our earlier spatial transcriptomics analysis displayed.

Both probes generated clear signals, demonstrating two important observations about the interacting pair we identified: first, JAG1 and VASN are typically expressed in independent cells, which verifies that the spatial colocalization we measured in spatial transcriptomics was not simply a byproduct of coexpression within a single cell type (Figure R4). Second, the independent cells that are highly JAG1 positive and VASN positive both localize to particular regions of the tissue, in close enough physical proximity to verify that it is possible these cells could physically interact *in vivo*. The regions most enriched with

bordering JAG1 and VASN expressing cells correspond to the highly immune cell rich follicles of the lymph node, as would be expected for an interaction that has a role in regulating leukocyte behavior (new main Figure 3e).

Figure R4. Single-molecule RNA hybridizations detects JAG1 and VASN in distinct but directly-adjacent immune cell populations within human lymph nodes. a. Hematoxylin and eosin stain of a human thoracic lymph node section. b. RNAscope control probes (yellow, red) and DAPI staining (blue) show the expected staining patterns and a lack of staining artifacts. c. Low-magnification staining of an entire lymph node section for JAG1 and VASN. d. High-magnification staining including JAG1, VASN, CD45 (as a pan-leukocyte marker) and DAPI.

New main figure 3e. Single-molecule RNA hybridization on human lymph node defines regions where newly-identified interaction partners are expressed in spatially-neighboring cells. A single lymphoid follicle enriched in *CD45*⁺ leukocytes is magnified (top), showing zonation of *JAG1* and *VASN* expressing cells into the corona and germinal center respectively (left). An inset (right) highlights a region of bordering cells, showing *JAG1* and *VASN* are expressed on distinct but physically-neighboring cells.

We were able to extend these results even further by expanding our measurements to 3 more pairs of genes encoding cell surface proteins that we found to physically interact. For not only *JAG1* - *VASN* but also *MCAM* - *CNTN1*, *PLXA4* - *APP*, and *VISTA* - *HLAE*, we could see cells expressing each half of the pair in nuclei belonging to cells in direct contact with one another, despite variation in the exact pattern of where cells localize (Figure R5). The other 3 probe pairs that were selected for this analysis were not selected based on prior expression data, but simply because of pragmatic reasons that these probes happened to be leftover at the Institute from unrelated past projects. We therefore consider it significant that despite these protein pairs originating from large-scale biochemical measurements, it is possible to observe each among colocalized immune cells within the human body. We would like to emphasize that because leukocytes are highly migratory cell types, it would be expected that the expression of one side of an

interaction pair is not always correlated with the neighboring expression of the other side of the pair. However, as we observe, these interactions appear physiologically possible and are not merely pairings of dubious biological significance, such as if one side of the interaction is almost only found in brain tissue and the other restricted to heart muscle. These data have now been added to Extended Data Figure 9.

Figure R5. A panel of transcripts corresponding to 4 different novel interactions all show spatially co-occurring cells within immune cell enriched compartments of the human lymph node. A magnified view of a single lymphoid follicle is shown for 4 different pairs of transcripts. The scale bar is 100 μ m and applies to all images.

Referee #2 (Remarks to the Author):

Review of Shilts et al “A physical wiring diagram for the human immune system”

Shilts et al start with a large biochemical screen of immune surface receptors. A full set of annotated receptors is classified into 6 structural groups for biotinylation and conjugation to tetrameric streptavidin. With the purified components an all vs all sort of elisa assay was performed and very specific results were obtained. This is evidenced by the fact that more than half of the PPIs in the network remain isolated pairs of interacting proteins and that almost all (25/28) PPIs could be validated with at least one of three attempts: quantitative repetition, SPR or in cell binding experiments. It is clear that the vast majority of PPIs found in the screen are “expected”, i.e. find support in the literature. This for sure is a unique, systematic high-quality approach to immune receptor PPIs.

An important further aspect of the PPI approach is that quantitative data, apparent affinities, are obtained for the (new?/ vs literature) PPIs. A mass equilibrium model that uses receptor expression and the PPI affinities is set up to predict cell-cell interactions. This develops into an interesting line of results presented as an interactive atlas of immune cell connectivity. Single cell expression data of 11 (immune) cell-types in four tissues reveal differential cell-cell connectivity (physical connection maps) together with receptor expression and PPI connectivity that can be interactively queried. The key finding here is that myeloid cells behave as hub in the cell-cell connectivity maps. A second data set, derived from transcriptomics of human lymph node confirms physical localisation of interacting leukocytes in vitro. This dynamic cell-cell contact maps can be a very useful resource and demonstrate an innovative way of combining expression and PPI information towards biological insight.

The last part is based on high-content microscopy for measuring leukocyte activation, where their purified protein constructs are used as perturbation. Put simply, soluble ligands are used to perturb cell-cell interactions and polarisation. This is a screen as such, however the authors state that the results are in line with the results seen for leukocyte interaction changes. Anyway Figure 4 e shows that cell -cell interactions match their predictions for a set of 8/9 proteins in tendency.

At this point I have to apologize and state that immune cell-cell contacts are not my core expertise, so I feel only medium confident in judging the effects of receptor proteins on leukocyte activation. Anyway, leveraging the high-quality biochemical PPI screen into an integrated view of immune cell-cell contracts is a major advance.

We are grateful for the reviewer’s detailed assessment of the “major advance” and “unique, systematic, high-quality approach” contributed by our work. We also thank the reviewer for their careful consideration of the “very useful” resource we provide.

Points for consideration:

*) Figure 1c is of insufficient quality, so that neither PPIs nor literature information can be read, the network is fine however and shows the PPIs.

We appreciate that these visualizations are limited in what they can display at one time. Our motivation for including panel c in Figure 1 was not primarily for the purpose of reading out individual PPIs, but rather to display the data generated from the screening method. These binding matrix grids are often considered standard for large-scale binary interactome studies including ones previously published in *Nature* (e.g. see Figure 2A of Özkan *et al.* 2013, Figure 2A of Visser *et al.* 2015, and Figure 1A of Smakowska-Luzan *et al.* 2018), and are shown to communicate two important features of the larger dataset. First, they demonstrate the sparsity of the interactome. A small fraction of all the binary measurements produce any binding signal, which is a hallmark of high-quality extracellular interaction networks and shows the low false-positive rate of the technique. In other words, it visually shows the frequency of the interactions that were *not* detected, as well as the ones that were. As the reviewer notes, a valuable feature of our approach is the “very specific results” achieved against known benchmarks, which we believe this form of visualization helps communicate. Second, this style of plot shows the consistency of the reciprocal binding measurements between bait and prey orientations, which is immediately visible as the striking symmetry across the $x = y$ diagonal axis. That is to say, the same receptor-ligand interactions are frequently detected using this approach irrespective of whether the interacting proteins are presented as either the immobilized bait or soluble prey. We agree that there also should be visualizations which better capture individual details of the interactome, which is why we included the network diagram in panel e.

*) PPI data: in essence the authors elude to only a single PPI involving the orphan receptor VISTA.

In the first paragraph Results section describing our PPI results, we chose to mention VISTA specifically by name because of the recent exceptional interest around this protein and because that PPI result for VISTA was, in itself, an answer to the long-standing quest to discover a physiological/non-tumor ligand. In subsequent Results sections however, we make specific references to several other PPIs we identified. Unlike VISTA where the binding itself was noteworthy, for these other PPIs additional sources of information that we collected (such as expression profiles, disease associations, or functional readouts in primary cell assays) were relevant to explaining why these PPIs were noteworthy, and that is why we chose to mention them in subsequent sections. In the Discussion, we include further elaboration into individual PPIs, viewed in the full context of not only the binding measurements but the full functional data and analysis contained in our study. We agree that there are many exciting PPIs that are worth discussing in detail out of our

findings, although due to space constraints we have had to limit ourselves to these few passages.

*) Literature PPIs: The literature curated PPIs remain somehow ill-defined throughout, even though I think the comparison is very much supporting the approach “captured a majority of all previously reported interactions”, numbers would be good.

This is another excellent suggestion when considering how our analysis forms a valuable survey of the current literature. In our revised manuscript, we have now added a new supplementary table (Supplemental Table 3) which details all the literature-curated interactions. As well as listing these interactions, we include fields quantifying binding kinetics with each available parameter (along with ranges and error estimates where applicable), and important notes such as which interactions depend on specific glycosylation patterns. All of these entries are associated with references to the primary literature sources.

Following the reviewer’s recommendation, we have also calculated a precise percentage of previously-claimed interactions re-identified using our approach to be 53%. As explained in our response to Reviewer #1’s comment about Extended Data Figure 3, there are multiple ways to define a reference set of previously-claimed interactions. Irrespective of which definition of a positive interaction is used, we consistently identified a majority (>50%) in the positive reference set. For this 53% calculation, we selected the most commonly-used definitions, where the positive reference set is based on any literature claim and the negative reference set is based on randomly permuted pairs.

*) How much do the new PPIs (only 28 from many) contribute to the cell-cell PPI model? What happens if the newly found PPIs are omitted from the model?

This is an interesting question to consider, which we can formally frame in two different ways: a) “how much do the newly-discovered PPIs in our study contribute to our ability to predict leukocyte adhesion”, and b) “how much does our study as a whole, including newly-discovered PPIs and literature curation, contribute to our ability to predict leukocyte adhesion”?

We have now analyzed both of these questions in detail. For part (a), just as the reviewer predicted, it is not the newly-discovered PPIs by themselves which are the main drivers of the model’s overall predictions. Removing these 28 new interactions results in a 0.8% change in the r^2 fit to the previously published leukocyte contact frequency data shown in Figure 2g. As the reviewer correctly pointed out, this is not unexpected. First, these are, as the reviewer writes, only “28 out of many” and thus it would be surprising if they alone dominated the output compared to the over 100 other interactions in our network which happened not to be novel. Second, surface protein interactions with the highest affinities are both the most likely to contribute to the model predictions and the most likely to have

been discovered by previous studies. Third, as we demonstrate in our new analyses which addresses the subsequent comment by Reviewer #2, these model predictions are robust to interactions being removed (as will be described in subsequent section, figures R7-R8), and thus even removing a handful of strong adhesive receptors is insufficient to drastically alter predictions against the backdrop of so many other contributing proteins.

When evaluating the advances we have made with our study however, part (b) of the question is also interesting to consider. Very few prior PPI databases include annotations of binding kinetics at all, and thus our detailed literature curation and empirical measurements provide the most complete database to date of this class of PPIs. The most complete previously-existing database that we have found is the IntAct database. However, when this database is queried with our set of leukocyte cell-surface proteins, only 29 PPIs with K_D information are returned. By contrast, our literature curation alone identified 114 PPIs with K_D information (>3x the number in the most complete prior database), not including the additional 23 PPIs from newly-discovered interactions we were able to derive quantitative binding parameters for through our study's SPR measurements.

When our model relied solely on the PPIs and affinity measurements available in the much less complete IntAct database, we observed an unusual bifurcated pattern in the predictions (Figure R6). Curious about this strange result, we investigated it in detail by decomposing the contributions of the constituent PPIs from IntAct. Despite being the most complete existing database, IntAct failed to curate several of the most important adhesive receptor interactions (e.g. the ICAM and integrin families, as well as others such as PECAM1 and JAM proteins). Instead most of the model's predictions came from a single protein: HLA-G. This is partly due to a mistake in the IntAct database that overestimates the binding affinity of HLA-G to LILR receptors by a factor of 1000. When this single protein is removed from the IntAct list, the combined interaction scores of all the measured cell types drops from 9132 bound molecules/ μm^2 to 2318 bound molecules/ μm^2 . Notably, HLA-G is not primarily reported to be an adhesive receptor, and thus even this poorer fit on the IntAct data is mostly a fluke of chance driven by PPIs known through the literature to not be the major drivers of leukocyte adhesion. When these 4 HLA-G -- LILR PPIs are removed, all accuracy is erased (Figure R6, inset). By contrast, our own model's output (Figure R6, left) when viewed even before running any validations agreed well with the existing knowledge that the field of immunology has accumulated over the years, with well-established major regulators of leukocyte adhesion (e.g. ICAMs, integrins) all comprising the largest contributors to cell adhesion. This contrasts greatly with the biological implausibility of outputs generated using public databases available prior to our study. We would emphasize that this highlights the limitations of relying on large databases without the painstaking manual curation effort and measurements we have invested in our study.

Figure R6. Comparison of our full model’s fit to previously-published leukocyte interaction data to a fit generated from a version of our mathematical model that only includes surface receptor interactions previously curated in the IntAct database. As in main figure 2g, the scatter plots show pairs of interacting cell types are represented by the two colors of each point on the plot. Pearson regression statistics are printed next to each legend. The two shaded regions on the IntAct-based model are to highlight how its predictions are only coarsely qualitative and fall into two general bins of ‘high’ and ‘low’. Below the scatter plots, bar plots show the top 15 proteins contributing to model output when using our dataset (left) compared to the output using IntAct as an interaction dataset (right). The scatterplot fit to published data when LILR receptors are omitted from the IntAct list is shown as an inset.

As a final note to add in relation to question (a), we believe it is worth emphasizing that even if our new interactions may not themselves be central drivers of the cellular connectivity seen in Figure 2g, our predictions on the consequences of experimentally perturbing these newly discovered interactions in Figure 4 demonstrate their utility in the model. We could not, of course, have made accurate predictions about SEMA4D, VISTA, et cetera if their interaction partners and kinetics were unknown. Beyond this, there is a crucial distinction between what theoretically could have been done before, and what actually was been done before, and to our knowledge no prior study has attempted anything similar to our model on a data source like IntAct.

*) The cell-cell interaction prediction model needs to be scrutinized with respect to PPI variation. How robust are cell-cell contact predictions based on mass action of sets of expressed receptors? Figure 2g summarizes predictive value of the model, however no error estimates are given. Figure 8 c clearly suggest that some interactions are more some are less important / specific.

This is an excellent observation by the reviewer, since it is true that the constituent PPIs in the model do vary considerably in their specificity and contribution to the overall cellular interaction scores. Interested in this question, we performed several analyses to scrutinize the model's sensitivity to individual PPIs and proteins. Please also refer to our reply to Reviewer 3 below, which features even further layers of validation we have now added for our mathematical model.

To test how our model responds to PPI variation, we first systematically removed one PPI at a time to determine how robust the predictions are. The vast majority of individual interactions can be omitted from the model with minimal consequence: 89% of PPIs result in less than a 1% change in the coefficient of correlation (r), 93% with less than a 2% change, and all but 2 individual PPIs can be omitted with less than a 5% change in the coefficient of correlation when fit to the published leukocyte-to-leukocyte contact data (Figure R7). We find it noteworthy that overall cellular connectivity is remarkably robust to individual cell-surface PPIs. There is considerable redundancy built into the proteome, and most major adhesive receptor families contain several orthologs with overlapping interaction specificities (for example, the ICAM, JAM, and NECTIN receptor families are highly interconnected and different receptors in those families retain sufficient homology to conserve binding partners). No one interaction dominates the model predictions. We therefore interpret this result not merely as a reflection of the resiliency of our modeling strategy, but of biological resiliency ingrained in these receptor pathways.

Figure R7. Distribution of model fits to published data following a complete leave-one-out analysis of all the constituent protein interactions in the model. The correlation was calculated when each of the 137 constituent PPIs were removed one at a time. The bin encompassing the same coefficient of correlation as the full model with no interactions left out is shaded a lighter shade of blue.

To further characterize the model’s robustness, we also considered how robust the model would be to removing individual proteins rather than interactions. As expected, because proteins can participate in multiple interactions, the influence on the coefficient of correlation post-fitting was slightly larger, although 90% of proteins could still be omitted with less than a 1% change in correlation. We manually inspected the two proteins with the largest influence on the fit: ICAM1 and PECAM1. As would be anticipated based on its known distribution on leukocytes, omitting PECAM1 caused an under-estimate of the strength of monocyte interactions. ICAM1 has a much more pronounced effect on the model’s predictions, yet as we show in our experimental results in Figure 4, this predicted major role for ICAM1 matches the empirical data we generated when perturbing ICAM1 with recombinant protein. We again interpret this based on our experimental evidence as more of a reflection of the underlying biological dependencies rather than an outcome intrinsic to our modeling approach.

Figure R8. Model fits to published data after systematically leaving out one surface protein at a time. The bars of height 1 corresponding to omitting ICAM1 and PECAM1 are indicated by text labels, alongside scatter plots and linear regression lines showing the raw fits when those respective proteins are omitted. The bar encompassing the same coefficient of correlation as the full model with no interactions left out is shaded a lighter shade of blue. The y-axis of the histogram is truncated so that rare deviations are visible.

Lastly, we thank the reviewer for pointing out that we did not include an error estimate in our original version of Figure 2g. It has now been revised to include shading to indicate

the 95% confidence interval along the linear regression line of best fit. Note that, in large part because of helpful suggestions by each of the Reviewers for additional validation steps, we identified a couple minor mistakes in our original code (including inputting the cell diameter rather than radius into a formula and entering an incorrect value from the literature), and so the exact positions of points on the graph have been updated compared to the initial submission. Of note, these corrections improved both our model's correlation to previously-published measurements (Figure 2g), and predictions on the consequences of experimental perturbations (Figure 4e). We thank the reviewers again for their suggestions in identifying these points.

Revised main figure 2g. Grey shading indicating the 95% compatibility interval of the linear regression has been added onto the previously-made scatterplot and this replaces the original figure in our revised manuscript.

Revised main figure 4e. Updated model results predicting the cellular interaction changes upon perturbation. This version replaces the original figure in our revised manuscript.

*) Figure 4 is very descriptive. For example, CD58 inhibits most cell-cell contacts and therefore completely abolishes what? In contrast VISTA inhibits 2 cell-cell type contacts only in resting conditions after 24 hours. Does this relate to the interactions with HLA-F/E or is it more complicated?

We appreciate the reviewer's point that in these kinds of large-scale studies, our goal of being as systematic and thorough as possible in investigating interactions across all major leukocyte populations in blood at once means that, by necessity, we must present relatively high-level summaries of quite intricate datasets. With that point granted, we would like to stress that the data in Figure 4 are nevertheless not "descriptive" in the traditional sense of being purely passive descriptions of a biological system. Instead all of the data shown in Figure 4 are the results of specific experiments designed to biochemically characterize leukocyte interactions, using defined recombinant protein reagents that individually perturb specific receptors. Rather than passive descriptions, these data are driven by the hypothesis that we can deconvolute the mechanisms behind intercellular connectivity into the contributions of individual PPIs. Indeed, our modeling results at the end of Figure 4 demonstrate that we can quantitatively formulate these hypotheses and test them using perturbation experiments. Similarly, our experiments designed to test for surface proteins that modulate lymphocyte activation in Figure 4 go beyond traditional 'descriptive' approaches of merely inferring receptor signaling based on counting kinase or other motifs in receptor cytoplasmic domains, and instead empirically test for the cellular activation responses.

As the reviewer implies, it is possible to pick out individual results contained within our larger summary figures to extract functional stories around individual receptors. In the case of CD58 which is mentioned by the reviewer, our observation that infusing recombinant CD58 downregulates interactions between T cells and a range of other cell populations matches well-established findings that the CD2-CD58 PPI is important for T cell adhesion (e.g. Dustin and Springer 1991, Rabin *et al.* 1993). The accompanying changes seen where B cell contacts rise is likely to be a compensatory effect, because our interaction scores are normalized such that it would be expected that if certain cell pairs show greatly decreased interactions, then that will free up unpaired cells to participate in other cellular interactions. From this previous research, we know that the decrease in T cell adhesive contacts with other leukocytes when CD2-CD58 interactions are blocked is causally linked to decreases in T cell activation (which is a trend faintly visible in our own T cell activation measurements, although for that previously-known result our comparatively small dataset with stringent multiple testing correction did not reach statistical significance, Welch's $t = -2.1$, unadjusted $p = 0.053$, adjusted $p = 0.26$).

The reviewer's question about VISTA/HLA-F is also an informative case to consider. Although of course far less is known about this interaction because our study marks the first description of it, the expression profiles of VISTA and HLA-F offer a reasonable basis to account for our experimental observations in the microscopy assay. The two significant

cell interaction changes in the VISTA condition were between classical monocytes and lymphocyte populations (with monocyte--CD8 T cell and monocyte--NK cell contacts significantly decreased, while CD4 T cell contacts also decreased but not past the significance threshold). As the reviewer noted, these changes are only observed in a background of leukocytes in their resting state but disappear upon low-level LPS stimulation. All of these distinctive features match the profile of where and when VISTA and HLA-F are expressed (Figure R9). Classical CD16⁻ monocytes are by far the dominant source of VISTA, explaining why classical monocyte interactions are most perturbed. This in itself is of some interest to consider in future studies, given that attention on VISTA has, so far, predominantly been focused on its role as a checkpoint for T cells while comparatively few studies have investigated monocyte contributions to VISTA function. The HLA-F ligand we identified for VISTA likewise matches the phenotypic profiles we observed: CD8 T cells and NK cells have high expression, while it is lower but still present on CD4 T cells and low on other monocytes. The tumor ligand for VISTA recently identified by Wang *et al.* (2019), IGSF11, is not detected at all in any of these cell populations. Lastly, there is the question of why we only observed phenotypes when immune cells were in steady-state. Again, the expression data provides an elegant explanation for this finding, as VISTA is dramatically downregulated upon immune activation (Fig. R9).

Figure R9. Expression levels of both VISTA and HLA-F obtained from a proteomics atlas match the context-dependencies seen in our cell interaction measurements following VISTA perturbation. Protein expression data from Rieckmann *et al.* 2017 are plotted for the given cell types in both steady-state conditions and after *in vitro* activation. Individual points are shown for each blood donor, as well as overlaid bar charts showing the mean and standard deviation error bars. No peptides matching IGSF11 were detected in the mass spectrometry proteomics dataset, and thus only VISTA and HLA-F are shown.

While performing a full accounting of every significant effect we observed and its specific context would be a very lengthy endeavor, we hope the two cases we have provided above that address the reviewer's examples of CD58 and VISTA illustrate how the data we

generated can fit into more tailored hypotheses about individual receptors. The effort we invested to make our datasets interactive and accessible was with this aim in mind, so that readers can extract from our larger study any cell population or surface protein of interest to them.

*) The manuscript, even in its short form, can maybe more explicitly point towards findings rather than saying: “Among our other findings are “, “we have been able to derive other interesting biological properties of the network”, “allows multiple kinds of analysis”. The reader is under the impression to miss/not understand something important what is the meaning of a quilt plot?

We appreciate the reviewer’s advice on the writing of the manuscript and have changed these passages accordingly. We agree that although our intent with that phrasing was to underscore that the analyses we performed are not the only analyses possible, these passages now read more clearly when they are pointed explicitly at the particular findings we describe. For the labels on the website, we have also replaced the use of non-standard terms like “quilt” in favor of the more explicit labels of “cell view” and “protein view”.

Referee #3 (Remarks to the Author):

Using a high-throughput screening method, Shilts and colleagues expand the known leukocyte surface protein interactions by about 20%, including ligands for a previously orphan receptor. They validated their findings using surface plasmon resonance, which allowed them to also quantify the strength of the binding. Pooling this acquired data with existing literature, they created a substantial resource of quantitative surface protein interactions in the human immune system, for which they also created an online interface.

We thank the reviewer for their insightful comments about the “substantial” findings and resources for the community contributed through our study.

Major points:

- The authors propose a simple model for quantifying the strength of the cellular interaction in terms of the quantity of interacting molecules expressed and the strength of their interaction. The validation was coarse using published literature (Fig. 2g) and experiment (Fig. 4e). This is one of the most substantial points of the ms, it could be highlighted more and the authors could strive to offer a stronger validation.

We agree our modeling results are worth highlighting more in the manuscript. In our revision, we have therefore expanded the Discussion to further elaborate our quantitative analyses of immune receptor affinity. Based on this comment and another similar excellent suggestion from reviewer 1, we also revised the Supplemental text to add 3 pages of additional explanation on the model and its utility in more broadly-accessible language (see also figures R2 and R3 which were added to the Supplemental text for this same purpose). We further overhauled Extended Data Figure 7 to include additional description and validations of our modeling approach.

As suggested, we have also striven to provide an even stronger validation of the model beyond the data already presented in the initial submission. In response to a question by Reviewer 2 above, we have scrutinized the model's robustness and validated that its predictions are stable against fluctuations in the exact inputs (see Figures R6, R7, R8, and revised 2g). Beyond this, we will now describe additional validations that we have performed to confirm the model's rigor. The first set of these validations evaluated the model with respect to its statistical grounding. If the predictions of the model are truly specific to the context being evaluated, then randomizing the internal representations of the model (such as its underlying PPI network or cell types on which each protein is expressed) should result in "null" distributions that significantly depart from the observed cell behavior.

To test this, we began by randomly permuting which proteins in our PPI network were considered as interactors. In doing this controlled "interaction shuffling" we made sure to preserve the original structure of the true PPI network: the same proteins as found in the true PPI network were used in the randomization, the randomization was done with replacement to recapitulate the observed structure in our PPI network, and affinity values were taken exactly from the true PPI network except now assigning them to receptor-ligand pairs at random. All other steps of the model were preserved. When 1000 random PPI permutations were fit against our published data benchmark, we observed a relatively uniform distribution of correlation coefficients centered around 0, as would be expected if the true PPI network we measured is essential for prediction accuracy (Fig. R10). Only approximately 6% of randomized networks resulted in correlations as strong as was observed when fit on our true data. We consider this to be quite low considering the amount of internally correlated variables retained after permuting the PPIs (e.g. if an important adhesive protein, when permuted, happens to land on another protein with a similar expression profile of cell types, then a correlation similar to the true fit would emerge). These analyses are now included in Extended Data Figure 7.

Figure R10. Random permutation of interacting protein-protein pairs while keeping all other data sources constant is sufficient to eliminate the model's fit to published data. This histogram displays the results of 1000 permutations where all aspects of the model were kept the same except for shuffling which protein pairs were listed as interacting. Randomization was done within the matrix of true interacting proteins, as opposed to all proteins on the leukocyte surface. Linear regression correlations to published data, following the same procedure as generated main text Figure 2g, were calculated for each permutation. Histogram bins equal to or greater than the observed correlation are shaded light blue.

While this establishes a specificity for the PPI input into the model, we next also considered the specificity of the cell type expression data input. We randomly permuted the cell type labels of our protein expression matrix, while keeping all PPIs and other features constant. Because there are only $4! = 24$ permutations, we evaluated all possible permutations. Again this generated a null distribution centered cleanly at 0, demonstrating a lack of internal biases that could explain our fits in the paper as having occurred by chance (Fig. R11).

Figure R11. Randomization of cell type labels in the model results in a null distribution centered at zero correlation to observed data. The cell type labels corresponding to surface

protein expression measurements in the model's input were randomly shuffled to generate all possible permutations. The correlation of these permuted models to published data (i.e. main Figure 2g) was quantified. The bin on the far right edge of the histogram corresponds to permutations that are almost identical to the true labels except for conservative within-class changes, while the bin on the far left edge corresponds to those same permutations but inverted to generate a negative correlation instead of positive.

These validations, together with our robustness analysis, emphasize the validity of our biophysical modeling approach. We also appreciate that the word “validation” can be interpreted more broadly as also referring to using new separate datasets, and verifying the fits to those as well. In the case of our work, we would stress both the difficulty of acquiring separate datasets given a paucity of techniques for directly measuring intercellular contacts, as well as the important fact that our model already aggregates disparate and independent data sources to reach its predictions and make its fits. To build our model, we have used protein expression data collected years ago from patients in Germany, combined with PPI networks and binding affinity measurements generated by us here as well as by collecting measurements from dozens of prior laboratories who have reported affinity data in the literature. Then the fit of the model is tested against both published imaging data from Austria and newly-acquired experimental perturbation data we collected with our coauthors in Switzerland. All of these disparate sources are integrated in a principled way using biophysical equations, thereby avoiding common concerns around ‘overfitting’ that can plague purely statistical models.

Despite the difficulties associated with acquiring a new independent dataset against which to validate our model fits, we have strove to nevertheless do this for our revision in the best way we could realistically achieve. Although no other existing dataset comes close to the depth of the previously-published study we compare against in Figure 2, we were able to re-purpose data from control experiments we had performed to generate a more limited yet independent measurement of baseline rates of the physical contacts between human leukocytes. These measurements were done on cells from a single blood donor using the pharmacoscopy technique we employ in Figure 4 of our paper. In spite of the variability intrinsic to this smaller dataset and the different data collection and analysis methods used, compared to the previously-published imaging study, our model still produced an accurate fit to this independent dataset (Fig. R12).

Figure R12. An independently-generated microscopy dataset measuring baseline leukocyte cell-to-cell interaction rates agrees with model predictions. Following an identical procedure to main Figure 2g, we correlated our model's predictions about cellular interactions with direct measurements done by microscopy, in this case from a new (but much smaller) dataset of leukocytes originally measured for use as negative controls. The methods of cell preparation and contact score calculation differ from the previously-published Vladimir *et al.* study (2017), which accounts for differences seen on the y-axis of this plot compared to Figure 2g.

Finally, we considered the highest bar of independent validation to be the generation of a new kind of prediction beyond what was previously performed and demonstrates the model's broad generalizability. Again, this is a very difficult task when so few empirical methods are available to measure intercellular contacts between human leukocytes. Employing our pharmacoscopy technique however, we could make a first attempt at this kind of validation by making measurements of the baseline rate of physical contacts involving innate lymphoid NK cells - a cell type that we previously had not considered in our model fit because no NK cell measurements were available in that previously-published study. Despite experimental limitations (e.g. these data are from 5 technical replicates on a single donor, in contrast to the previously-published imaging study which aggregated over 1,400 measurements to produce a reliable baseline), our model could still produce predictions that significantly fit to this leukocyte dataset including NK cells (Fig. R13).

Figure R13. Predictions from the model can be extrapolated to new cell types not previously considered, based on a small microscopy study including NK cell interactions. Model predictions were made for a considerably larger set of leukocyte interactions including NK cell populations. These were then correlated to newly-provided microscopy measurements of leukocyte interaction rates. Although considerably more cell-cell pairs were measured, the dataset is also considerably smaller than the published study used in Figure 2g and consequently more variable.

- The authors found several interesting biological leads using the model (e.g. immune cell activation on average decreases their potential interaction strength; DC & NK cells show increased interaction strength in kidney tumors etc) though all of these findings are interesting; their biological significance is unclear and they would require substantial further investigation.

We are gratified to see the reviewer's interest in the findings we report in our study. We fully agree that many of these discoveries would be fascinating to pursue in future publications as detailed single-point stories, although we concur with the reviewer's assessment that there are factors such as limitations to existing technologies and time considerations that would make substantial *in vivo* work beyond the scope of this study focusing on discovering and characterizing the overall interactome of immune receptors. We have carefully evaluated what would be practically achievable, both in the context of revisions and imagining future long-term projects building off our work reported here. To consider pragmatically the example given about immune cell activation shifting potential interaction strength, we have held discussions with other laboratories about the possibility of experimentally dissecting and expanding on this result, but a true test would require a complete *in vivo* model (ideally human, as this is the entire subject of our manuscript)

where it would be possible to artificially increase or decrease receptor affinity with greater precision than any technology we are aware of can currently offer.

Although experimentally dissecting this process may prove prohibitive, there are fascinating clues from the human genetics literature that support the biological significance of our finding about affinity switches in immune activation. In our analysis, we observed a preference for lower-affinity surface interactions prior to activation, which is then exchanged for stronger interactions in an inflamed state. In terms of significance, this could imply that cellular adhesion, when too strong, could interfere with immune responses *in vivo*, in spite of how simple *in vitro* assays might make it appear that stronger adhesion always monotonically facilitates activation. Indeed, it has been previously hypothesized that low affinity interactions are advantageous during the steady-state “scanning” stages when it is the role of the highly migratory immune cells to identify stimuli and switch to higher affinity interactions once reacting to an identified stimulus (Martinez and Evavold 2015). In fact this has been demonstrated by mouse models where adhesive immune receptors were genetically modified to enhance binding (Semmrich *et al.* 2005, Martinez *et al.* 2020). The experimentally-caused excessive binding affinity prevented immune cells from efficiently responding to pathogenic stimuli. Other studies have theorized that transient B and T cell contacts *in vivo* are important for efficient sampling of antigens (Sinai *et al.* 2014), providing one interpretation for why we observe a preference for low-affinity receptors in their basal state.

By contrast, once in an inflammatory state, a lack of sufficiently high-affinity interactions may be associated with pathological responses or persistent inflammation. For example, three genetic variants that have appeared strongly associated with systemic lupus erythematosus (SLE) in genome-wide association studies are variants in ITGAM, which as our model quantifies, is an important high-affinity adhesion receptor. Studies into these clinically-significant ITGAM variants have found they reduce the affinity of binding to ITGAM's surface partners, and this decreased adhesion may be causally linked to the persistent autoimmunity seen in SLE (MacPherson *et al.* 2011). Intriguingly, agonists that promote integrin adhesion have shown promise as therapeutics to dampen these persistent inflammatory responses (Hafeez Faridi *et al.* 2017). This could hint that the affinity shift we observed may relate to the proper initiation (lower affinity) and termination (higher affinity) of immune responses, which again we agree would make a fascinating future study that builds off our initial findings here, once such a study becomes more technically feasible. We have extended our Discussion section in our revised manuscript to make the potential biological significance of these results more apparent.

As a precursor to such a study, one step we have been able to take was to further confirm this finding by repeating our analysis on an independently-generated data source. We acquired expression data from a recent study that sorted human immune cell populations, performed stimulations to activate the cells, and then identified differentially expressed genes using RNA-seq (Calderon *et al.* 2019). Despite coming from an independent cohort of blood donors, sorting for different cell subpopulations than our original analysis,

measuring RNA instead of protein, and using different calculations for determining differential expression, the results were remarkably consistent with our original findings (Fig. R14). These data are now included in our revised Extended Data Figure 6. Please note that in the process of performing this validation, we also corrected a mislabeling present in the original figure where the x-axis category labels were flipped. This has now been fixed on all plots.

Figure R14. An independent transcriptomic dataset faithfully replicates the finding that immune cell activation is accompanied by a broad shift in receptor binding affinity profiles. The same analysis as is reported in Extended Data Figure 6 and main Figure 2e was repeated for a separate expression dataset (Calderon *et al.* 2019 bulk RNAseq measurements instead of the Rieckmann *et al.* 2017 mass spectrometry proteomics measurements used previously, which also differs in which cell populations were investigated and the stimulation methods used). The binding affinities of receptors significantly upregulated upon activation (labeled “Active” on the x-axis) were contrasted to the receptors downregulated upon activation (labeled as “Resting”). This was done both stratified across the measured cell types (top) and aggregated across all cell types (bottom).

Next, we sought to expand on the “significance” and biological interpretation of our results in Figure 4. To do this, our strategy was to match the cellular interaction changes we observed upon adding recombinant surface proteins against the changes seen after stimulating leukocytes with characterized therapeutic molecules that have known and well-established mechanisms of action. Our original data from perturbing leukocytes with purified recombinant surface proteins revealed 3 main clusters of cellular interaction changes, as we documented in Extended Data Figure 10. By similarly measuring the leukocyte behaviors to stimuli of known influence on the immune system using the same

assay, we could cluster these characterized immunomodulatory drugs and molecules into analogous groupings (Fig. R15). The actions of recombinant SEMA4D, SEMA7A, and VISTA after 24 hours showed similar changes in cellular connectivity as observed when leukocytes have completed reacting to inflammatory stimuli such as fungal Zymosan or Staphylococcal Protein A. These factors cause an increase in the contacts within monocyte populations at the expense of reduced monocyte contacts with lymphocytes compared to controls. These effects are known to emerge from signaling within monocytes, which agrees with the high expression levels of VISTA and SEMA4D and their ligands in monocytes. A second cluster that includes recombinant CNTN1, SIPRA, and CD58 exhibited a distinct pattern where B cell contacts with all cell types except classical monocytes were elevated, while there was a modest decline in monocyte contacts generally. This matched the result of incubating cells with known stimulatory cytokines such as IL-1 β and TNF- α . A final cluster broadly consisting of enhanced B cell interactions with no accompanying major shifts in other populations was shared between the proteins APLP2, CHL1, JAG1, and CD93 and anti-inflammatory corticosteroid medications such as cyclosporine. Using these data, we can enhance our biological interpretation of these potentially useful recombinant protein constructs and, perhaps more importantly, begin to build an understanding of how the patterns of immune cell interactions are connected to known functional pathways.

Figure R15. Comparison of observed clusters of cellular interaction changes in response to recombinant protein perturbation to known immunomodulatory molecules reveals shared groupings. We performed pharmacoscopy measurements on a panel of drugs and other

molecules with known mechanisms of action on immune cells. Three general groupings could be drawn which roughly parallel what was observed from our recombinant protein conditions, as previously reported in Extended Data Figure 10c. Within clusters, the characterized drugs and molecules shared fairly consistent modes of action, making it possible to assign commonalities between the actions of these known therapeutic and immunomodulatory molecules and the potentially therapeutic proteins we investigated in our study.

Finally, in our revisions we also worked to demonstrate how our new receptor interactions can serve as inputs into computational tools, such as we show in Figure 3, in order to generate more detailed functional hypotheses about pathways that may be of interest to specific investigators.

Quite recently, computational methods have been invented to integrate i) single-cell expression data, ii) cell surface receptor-ligand interactions, iii) intracellular phosphorylation and other signal transduction cascades, and iv) gene regulatory networks, in order to infer complete pathways ensuing from cell-to-cell communication. We applied the NicheNet method (Browaeys *et al.* 2019) to further investigate the single-cell RNAseq dataset of immune cells isolated from kidney tumors that the reviewer highlighted (Fig. R16 - R17). We set up a customized version of NicheNet's internal data structures that was constructed out of the novel cell surface interactions we discovered and their corresponding signaling predictions. From this, using the tool's analysis pipeline we could i) identify surface ligands that are differentially-expressed in the tumor microenvironment, ii) after linking those ligands to their receptors on other cells, construct a downstream signaling network that converges on known transcriptional regulators, and iii) link target genes differentially expressed in the receptor-bearing cell to the action of that receptor's putative signaling pathway via a transcriptional regulation model. From this we can generate detailed hypotheses about the significance of newly-discovered receptors on explaining the transcriptional profiles of immune cells in the tumor environment.

We focused on the classic cellular interaction between tissue-resident dendritic cells (in the case of the kidney dataset, pDCs) and CD4+ helper T-cells. Our first analysis asked if there are external signals that pDCs receive in the tumor but not in healthy kidney tissue that cause them to differentially present surface ligands. Among several modules of signaling active in tumor pDCs, we observed one including signaling downstream of the CD74 receptor on pDCs which is predicted to strongly regulate presentation of JAML (Fig. R16). We were intrigued by this result because our study uncovered that JAML (also known as AMICA1) participates in a novel interaction with CD320. We then hypothesized that this change in JAML expression on pDCs may influence the communication between pDCs and T cells in the tumor.

Figure R16. NicheNet analysis identifies novel cell surface interactions that potentially regulate dendritic cell surface ligands in the tumor microenvironment. As an illustration of how more detailed significance can be extracted out of our datasets and methods, we applied a custom version of the NicheNet method populated with our immune receptor interactome to single-cell RNAseq data of immune cells isolated from kidney tumors and healthy reference tissue. External signaling pathways predicted to be differentially active in tumor-resident pDCs are listed on the left, with downstream target genes affected by the signaling shown in the matrix on the right. We note the example of JAML, which has the single highest prediction scores of any downstream target on the pDCs.

To interrogate our hypothesized pathway, we then ran a tailored NicheNet analysis of only intercellular communication between pDCs and helper T cells. This first revealed the direction of the observed change in JAML levels on tumor pDCs as a significant decrease. While JAML is abundant on pDCs isolated from healthy sections of kidney, within the kidney tumor JAML expression drops approximately by half. We could then computationally infer the consequence of this drop in JAML on communication with nearby T cells. NicheNet links significant changes seen in IRF1 and CIRBP signaling pathways in T cells that infiltrate the tumor to the loss of JAML stimulation by pDCs. The CD320 receptor for JAML we identified provided the critical link between these two phenotypes, as CD320 present on the T cells is attributed by NicheNet’s gene regulatory model to explain the changes in IRF1 and CIRBP via the influence of pDCs (Fig. R17).

Figure R17. Gene regulatory models can link differential expression of a novel receptor - ligand pair to inferred downstream signaling consequences in immune cells. Following from the analysis done in Figure R16, we first show the differential expression of surface ligands on pDCs extracted from kidney tumor biopsies, in particular with striking changes to JAML levels. Focusing on the well-characterized interaction between pDCs as antigen presenting cells and CD4+ helper T cells, we again used NicheNet to calculate the most probable signaling consequences of that differential JAML expression on the activity of the T cells which bear the CD320 receptor we identified for JAML. Targets in our gene regulatory analysis were filtered to exclude those which recurred non-specifically in more than half of cases.

Both IRF1 and CIRBP are characterized gene regulatory factors in the context of CD4+ T cells, with IRF1 acting as a transcription factor to turn on helper T cell differentiation (Kano *et al.* 2008) and CIRBP as an RNA-binding translational regulator linked to inflammatory responses (Lujan *et al.* 2018). Independent analyses have found IRF1 in particular as a marker of cancer prognosis and the exhaustion of tumor-infiltrating T cells, which raises an intriguing hypothesis that the JAML-CD320 interaction we discovered could have relevance in explaining tumor immune rejection.

We hope this provides an instructive case study on the utility and significance of the findings and resources we present in our manuscript. As shown in our single-cell transcriptome analysis here and in the originally-submitted manuscript, it is possible to make computational inferences of complex processes between multiple cell types occurring in their context of the human body, all of which otherwise would be experimentally intractable. This analysis of pDCs and T cells in kidney tumors is just one example out of many that future investigators may extract out of our findings. While current computational analysis tools may admittedly be coarse, as they continue to be refined and enhanced, our study provides a strong foundation for their application to immune cells. As in the JAML-CD320 example, the receptor-ligand interactions we describe in our interactome may provide the missing information to draw clinically-meaningful associations.

- In the last part of the manuscript the authors use a novel high-content microscopy approach for validation – this is the most extensive validation in the paper however, however the approach is the main subject of an independent work.

The reviewer is correct that the novel high-content microscopy technique we apply in the last section of our manuscript is the primary subject of another manuscript which describes the technical details of developing the assay, as we have cited. However, this should not be interpreted as meaning the present study has not contributed any new extensions of the method that go beyond that initial descriptive study of the technique. To help make these distinctions more clear for the reviewers of our present manuscript, we have received the agreement of all authors on that separate technique paper to post it as a preprint, which you can now find on Biorxiv:

<https://www.biorxiv.org/content/10.1101/2021.12.03.471105v1>

A first noteworthy advance described in the present manuscript that goes beyond the initial technique description is that we present for the first time multiplex measurements of lymphocyte activation that span all 3 major lymphocyte populations at once: T cells, B cells, and NK cells. The study by Severin *et al.* had only investigated T cell activation markers. Extended Data Figure 9 shows benchmarking for classifiers that detect NK and B cell activation. We would therefore expect future publications based on this microscopy method to cite both manuscripts together as forming the foundation of the approach and its application.

Second, we would like to emphasize the additional technical work we performed to adapt the method for use with bespoke recombinant proteins as perturbations. Previously the pharmacoscopy method has been employed for small molecule drugs and a limited number of peptide-based therapeutic molecules. Diverse panels of large recombinant proteins had not been put into the assay. A first requirement was therefore to tune the protein purification protocol to reach sufficient stringency to eliminate contaminants that could influence the behavior of the immune cells. We achieved this by a combination of lowering Ni-NTA resin bed volumes to eliminate spare binding capacity, higher-stringency washing, and dialysis with a wide-pore membrane.

Subsequently, we titrated the quantity of recombinant protein required to perturb receptor interactions. Given the range of binding strengths that surface receptors display, we also considered the format in which the recombinant proteins are presented: either as monomers or as tetramers clustered around neutravidin to enhance weak binding through avidity. Although a dose as small as 32 pmol of protein was sufficient to elicit some responses for a relatively high-affinity receptor such as ICAM1, there was a clear dose-dependent relationship in the magnitude of changes (Fig. R19). Equally importantly, higher doses of the control samples did not elicit any signs of major off-target contamination. Thus we selected the highest available doses for our final assay. These experiments also clearly revealed an advantage to multimerizing recombinant proteins for use as functional

reagents. Approximately 2 to 3 fold lower doses of tetramer could elicit the same response as the comparable quantity of purified monomer. We considered it probable that this encouraging trend would be even more pronounced for receptors with even lower affinity than ICAM1, and hence all proteins in our study were multimerized prior to their addition onto cells. These additional tests on ICAM1 we performed to establish the method also replicate the findings we report in our main manuscript and show its dose-dependence.

Figure R19. Pilot testing identified optimal recombinant protein presentation methods and concentrations to elicit robust responses in pharmacoscopy measurements of leukocytes. Prior to the experiments reported in our main Figure 4, we applied a range of recombinant ICAM1 concentrations and formats (i.e. presentation either as a monomer or multivalent tetramer). As a well-characterized adhesive factor, ICAM1 served as a positive control to gauge the extent to which changes in cellular interaction could indeed be observed. Median pharmacoscopy interaction scores are plotted on the top for each condition tested, while dose series for select cell-cell pairs are shown on the bottom.

Although these technique optimizations are secondary to the main message of our manuscript, they nevertheless provide an important contribution that enables this high-content microscopy technique to be used to investigate a range of endogenous human proteins. Now that we have adapted the technique to handle any purified recombinant protein, our laboratory has already made plans to use this approach for future projects that investigate effects of pathogen proteins on the human immune system, which could reveal their immunoregulatory targets. In summary, we would like to reassure the reviewer that we are using the very latest high-content microscopy technologies for assaying leukocytes, and although we do not make any claim that these developments are a major

component of this current manuscript, we have extended its use in ways that we believe will be of interest to other researchers.

- The online resource is not online yet, so no way to assess it.

We sincerely apologize if the web server was not available or having issues at the time the reviewer tried to access it. There was a short delay between when our manuscript was submitted and when the server went public, so depending on the time the URL was accessed it may have appeared unavailable. We and our collaborating labs have since made repeated use of this online resource for projects on immune interaction niches, so we are confident it is currently accessible. We also provide the full code and data files for the online resource with our submission, in case reviewers would like to run and inspect the tool in more detail.

<https://www.sanger.ac.uk/tool/immune-interaction/immune-interaction/>

Minor points:

1. The panel in Fig. 3A should not be introduced as revealing 'connectivity patterns' or 'organizing principles'. It merely shows the interactions, and, to a human-reader, this panel is almost non-informative.

We agree with the reviewer's suggestion that the language we use in introducing Figure 3 could be made more precise. In the revised manuscript, we have removed the generic phrases 'connectivity patterns' and 'organizing principles'. In their place, we agree with the reviewer that panel A of the figure is simply best-described as a resource or atlas. Terms like "pattern" we now reserve for panel C, where we specifically extract a single recurring motif seen in how immune cells interact across tissues. We appreciate that the density of information in the panel can make it taxing to digest on its own, which was a major motivation for our web tool which allows users to make more customized or magnified views of cell populations or proteins of interest. Since the figure panel must be a single static image rather than that more dynamic web view, we opted to make it a visual summary of all the features we encapsulate, which can then be narrowed by readers.

2. Illegible, minute fonts are very prevalent throughout the figures.

In our revisions we have revised every figure in the entire manuscript to ensure all fonts are at least the minimum size set out by *Nature* or larger. We appreciate the reviewer's patience in evaluating our earlier figure versions for the initial submission.

References :

- Braun, P. et al. An experimentally derived confidence score for binary protein-protein interactions. *Nat Methods* 6, 91–97 (2009).
- Browaeys, R., Saelens, W. & Saeys, Y. NicheNet: modeling intercellular communication by linking ligands to target genes. *Nat Methods* 17, 159–162 (2020).
- Calderon, D. et al. Landscape of stimulation-responsive chromatin across diverse human immune cells. *Nat Genet* 51, 1494–1505 (2019).
- Dustin, M. L. & Springer, T. A. Role of lymphocyte adhesion receptors in transient interactions and cell locomotion. *Annu Rev Immunol* 9, 27–66 (1991).
- Faridi, M. H. et al. CD11b activation suppresses TLR-dependent inflammation and autoimmunity in systemic lupus erythematosus. *J Clin Invest* 127, 1271–1283.
- Kano, S. et al. The contribution of transcription factor IRF1 to the interferon-gamma-interleukin 12 signaling axis and TH1 versus TH-17 differentiation of CD4+ T cells. *Nat Immunol* 9, 34–41 (2008).
- Lujan, D. A., Ochoa, J. L. & Hartley, R. S. Cold-inducible RNA binding protein in cancer and inflammation. *Wiley Interdiscip Rev RNA* 9, (2018).
- MacPherson, M., Lek, H. S., Prescott, A. & Fagerholm, S. C. A systemic lupus erythematosus-associated R77H substitution in the CD11b chain of the Mac-1 integrin compromises leukocyte adhesion and phagocytosis. *J Biol Chem* 286, 17303–17310 (2011).
- Martinez, L. et al. A Genetic Model of Constitutively Active Integrin CD11b/CD18. *J Immunol* 205, 2545–2553 (2020).
- Martinez, R. J. & Evavold, B. D. Lower Affinity T Cells are Critical Components and Active Participants of the Immune Response. *Front Immunol* 6, 468 (2015).
- Özkan, E. et al. An extracellular interactome of immunoglobulin and LRR proteins reveals receptor-ligand networks. *Cell* 154, 228–239 (2013).
- Rabin, E. M. et al. Inhibition of T cell activation and adhesion functions by soluble CD2 protein. *Cell Immunol* 149, 24–38 (1993).
- Rieckmann, J. C. et al. Social network architecture of human immune cells unveiled by quantitative proteomics. *Nat. Immunol.* 18, 583–593 (2017).
- Semrlich, M. et al. Importance of integrin LFA-1 deactivation for the generation of immune responses. *J Exp Med* 201, 1987–1998 (2005).
- Severin, Y. et al. Multiplexed high-throughput immune cell imaging reveals molecular health-associated phenotypes. doi:10.1101/2021.12.03.471105 (2021)

Shilts *et al.*

Shao, Y.-J. et al. IRF1-mediated immune cell infiltration is associated with metastasis in colon adenocarcinoma. *Medicine (Baltimore)* 99, e22170 (2020).

Sinai, P. et al. T/B-cell interactions are more transient in response to weak stimuli in SLE-prone mice. *Eur J Immunol* 44, 3522–3531 (2014).

Smakowska-Luzan, E. et al. An extracellular network of Arabidopsis leucine-rich repeat receptor kinases. *Nature* 553, 342–346 (2018).

Trepte, P. et al. LuTHy: a double-readout bioluminescence-based two-hybrid technology for quantitative mapping of protein-protein interactions in mammalian cells. *Mol Syst Biol* 14, e8071 (2018).

Trigg, S. A. et al. CrY2H-seq: a massively multiplexed assay for deep-coverage interactome mapping. *Nat Methods* 14, 819–825 (2017).

Visser, J. J. et al. An extracellular biochemical screen reveals that FLRTs and Unc5s mediate neuronal subtype recognition in the retina. *Elife* 4, e08149 (2015).

Wang, J. et al. VSIG-3 as a ligand of VISTA inhibits human T-cell function. *Immunology* 156, 74–85 (2019).

Reviewer Reports on the First Revision:

Referees' comments:

Referee #1 (Remarks to the Author):

The authors have addressed all raised points to my satisfaction.

Referee #2 (Remarks to the Author):

Revised version of "A physical wiring diagram for the human immune system"

The authors invested into the revised version of the report and provide substantial additional analyses and clarification.

Specific comment to PPI robustness of the ODE model. I think the analyses are informative, single PPI removal and node removal approach ... What I find really convincing is the argument that their PPI data together with literature curation (case b, page 12 in the rebuttal letter) for the modelling bring a major advance. This is well shown in comparison to the INTACT data set. I thus I think that Figure R6 is a very good extension to Figure 2g (now revised). It is of course up to the editor and ultimately to the authors whether or not to include it as a supplemental figure. Congratulations to the work! I suggest to go ahead.

Referee #3 (Remarks to the Author):

- The new connection of the clustering analysis from Extended Data Fig. 10C to the changes elicited by common soluble immunomodulators, which is shown in Fig. R15 should be made part of the extended data. This goes potentially in the right direction in terms of finding biological interpretation of the data shown; however, it stops short of it. What is the interpretation of the relationship between the soluble perturbation and the recombinant protein perturbation if they fall in the same class? Is this to say that the soluble signal elicits the expression of the respective ligand/receptor in the most perturbed cell types?
- Figures R6, R8 and R13 need to be part of the manuscript/extended data, which either they currently are not, or I must have overlooked them due to the absence of respective pointers in the rebuttal letter.
- The presumed co-localization of cells (that are claimed to be neighbouring each other more often than by random) is not obvious from Fig. 3E or Fig. R5 and no statistical analysis offered.
- Figure 2D caption reads 'Immune cell subsets utilize different distributions of binding affinities when communicating with other cell types.'; however, no statistical test of whether the distributions actually are different is offered. In fact, they seem remarkably similar. Also, the distributions are clearly log normal, thus median would be a more appropriate characteristic of the central tendency than average (i.e. mean) currently used in the color scale.
- The analysis done in Fig. 2E needs to be better described. I understand K_d is a property of a ligand-receptor pair, not that of a receptor. What do then the authors mean when they say 'the affinities of down-regulated receptors are compared to the affinities of up-regulated receptors'? Also, the current labeling of the categories on the x-axis is misleading and evokes that what is shown is the distribution of K_d 'for all receptors' in the respective states, while what is shown is merely the distribution of K_d 'of the receptors' down- and up-regulated on activation. This needs to be corrected by making the x-axis read 'Down-regulated' and 'Up-regulated', or similar. Are the affinities weighted here by the magnitude of change in expression level? If not, why?
- Fig. 3D left panel shows JAG1-VASN expression, but the caption does not make any mention of this. More importantly, it seems to want to suggest that JAG1-VASN are expressed in neighboring cells more often than it would be by random, but no statistical test for this is provided, e.g. by

permuting the identity of spots. Also, how was JAG1-VASN chosen? This is important to know as multiple hypothesis correction might be necessary to apply for the permutation test mentioned above given the many options.

- Fig. 3D right panel needs to include Prior+Novel. Is it any different to Prior only? If not, what is the message of the panel? Would it not be more appropriate to make a quantitative estimation of interaction strength between each of the spots using Prior links and Prior+Novel links and show whether this metric predicts colocalization significantly better if using also the Novel links?

- For Fig. 3C, more statistical testing might be necessary. How was myeloid lineage chosen for this comparison? How does such graph look like e.g. for lymphoid vs. other? As a statistical test, given that it is unclear that the centrality values are normally distributed, what might a be more appropriate than a t-test would be a bootstrapped p value - taking all possible choices of four/five (depending on how many myeloid cell types were identified in that tissue) cell types out of all the cell types and asking what proportion of times the mean centrality comes higher than for the myeloid subset.

- For Fig. 3A, from reading the main text, caption and Methods, it is unclear how the links in circos plots are derived, so this needs to be better stated. Are the circos plots weighted? Is it that the interaction strength was calculated using the presented model that integrates the ligand-receptor interaction strength and expression levels? If it was not weighted, why?

Nature manuscript 2021-06-10485
Shilts *et al.*

Author responses : Nature manuscript 2021-06-10485

We greatly appreciate the careful effort by the reviewers to help us enhance the clarity, depth, and rigor of our manuscript. We are pleased to see the positive appraisals of the quality and significance of our work. In this revision, we address the remaining technical questions of the reviewer to improve the understandability of the paper and compare alternate statistical methodologies.

Referee #1 (Remarks to the Author):

The authors have addressed all raised points to my satisfaction.

We thank the reviewer again for their beneficial contributions to the manuscript during peer review and their positive assessment of the manuscript.

Referee #2 (Remarks to the Author):

Revised version of “A physical wiring diagram for the human immune system”

The authors invested into the revised version of the report and provide substantial additional analyses and clarification.

Specific comment to PPI robustness of the ODE model. I think the analyses are informative, single PPI removal and node removal approach ... What I find really convincing is the argument that their PPI data together with literature curation (case b, page 12 in the rebuttal letter) for the modelling bring a major advance. This is well shown in comparison to the INTACT data set. I thus I think that Figure R6 is a very good extension to Figure 2g (now revised). It is of course up to the editor and ultimately to the authors whether or not to include it as a supplemental figure. Congratulations to the work! I suggest to go ahead.

We are grateful for the reviewer’s kind words and thoughtful assistance in improving the manuscript to reach this stage.

Referee #3 (Remarks to the Author):

- The new connection of the clustering analysis from Extended Data Fig. 10C to the changes elicited by common soluble immunomodulators, which is shown in Fig. R15 should be made part of the extended data. This goes potentially in the right direction in terms of finding biological interpretation of the data shown; however, it stops short of it. What is the interpretation of the relationship between the soluble pertubagen and the recombinant protein perturbation if they

fall in the same class? Is this to say that the soluble signal elicits the expression of the respective ligand/receptor in the most perturbed cell types?

We are grateful for the reviewer's positive assessment of the immunomodulator data we included in response to their very constructive comments on the previous round of revisions. We personally would be cautious about interpreting a shared phenotype elicited by a characterized immunomodulatory perturbation and a recombinant protein as direct evidence that the perturbation is acting on the same ligand or receptor as the protein. As a demonstration of this, consider the middle cluster of figure R15, where IL-1 β and TNF- α belong to the same cluster despite their molecular mechanisms acting through different receptors (IL-1R and TNFRSF1A/1B respectively). Instead, we would argue that these data reveal higher-level phenotypic convergences. This means instead of recombinant APLP2 necessarily proceeding through the same receptor as the immunosuppressive drug cyclosporine, the results reveal that both perturbations converge on a shared set of immune cell interaction changes which may reflect a broadly immunosuppressed state. These convergences may reflect shared mechanisms further downstream in each molecule's signaling/effector pathway.

The mechanistic resolution one can achieve in interpreting results like this depends on the granularity of the clustering. In the case of figure R15, we only considered coarse-grained groupings based on how the complex responses we observed naturally clustered into 3 broad functional profiles. As further data is collected using this newly-developed pharmacoscopy technique, it may become possible to make finer-grained distinctions between clusters of immune cell responses. A finer-grained clustering could have the resolution needed to dissect more specific mechanistic details of the signaling pathway in a way that could potentially reveal shared receptors or signaling proteins between the actions of a recombinant protein and a clinically-characterized immunomodulatory drug. We are pleased by the reviewer's desire to make this analysis available to readers, and as explained further in the next comment, we will make figure R15 available through Nature's open peer review option.

- Figures R6, R8 and R13 need to be part of the manuscript/extended data, which either they currently are not, or I must have overlooked them due to the absence of respective pointers in the rebuttal letter.

We are pleased to see the positive appraisal of the analyses we performed for our revisions, and agree that making them available to readers would strengthen the conclusions and rigor of our manuscript. Within the strict space constraints set by the journal's guidelines, we managed to fit figure R8 into Extended Data Figure 6, where it now has its own captioned panel. Despite our best efforts, we were unable to fit the other revision figures into the Extended Data, but in recognition of the value added by these additional analyses, we would like to reassure the reviewer that we will make the entire revision publically available through *Nature's* transparent "Open Peer Review" option.

This will make every figure and analysis (not just R6, R8, and R13), along with the lengthier descriptions we had written, fully accessible.

- The presumed co-localization of cells (that are claimed to be neighbouring each other more often than by random) is not obvious from Fig. 3E or Fig. R5 and no statistical analysis offered.

For the *in situ* RNA imaging presented in figure 3e and its accompanying Extended Data figure 9, we would like to clarify that those figures are *not* claiming receptor-expressing and ligand-expressing cells are “neighbouring each other more often than by random”. Instead the colocalization experiment we perform in figure 3e was done to support the claim as written in the manuscript that “(Fig 3e) [t]his confirms these interactions have the potential to occur between immune cells *in vivo*”.

On the level of a single individual interaction such as is shown in figure 3e, we would not necessarily expect high colocalization, due to the fact that leukocytes are highly migratory and their receptor contacts are often very transient. As useful comparisons, receptor interactions that are well-established such as the PD1-PDL1 or T cell receptor-MHC class II interactions have been measured *in situ* by similar approaches to the one we applied for our novel interactions (e.g. Kim *et al.* “Surveying expression of immune checkpoint markers in the tissue microenvironment” (ACD Bio, 2016) and Goltsev *et al.* Deep Profiling of Mouse Splenic Architecture with CODEX Multiplexed Imaging” (Cell, 2018)). When viewing a single static snapshot of a fixed tissue section, there will be many T cell receptor-expressing T cells and several MHC class II-expressing antigen presenting cells, but only a relatively small fraction of those T cells would be captured in the process of interacting with antigen presenting cells at any one given time. For cell surface interactions in general where the protein pairs are expressed on two different cells, in contrast to cytoplasmic proteins where it is a prerequisite that they must be expressed in the same cell to physically interact, one would expect that colocalization frequencies could vary anywhere from 100% to single digit percentages and still have biological significance. The important finding for interacting cell surface proteins is not the exact colocalization percentage but rather that the cells bearing each surface protein are indeed sufficiently proximal that they can communicate, even if they do so rarely.

This is why we phrased our claim simply to say the two proteins of the interaction pair do co-occur in the tissue among nearby cells, and thus the signaling between them is possible *in situ*. This is significant because the novel interaction pairs we report were identified purely by systematic screening of purified proteins, and so our findings in figure 3e counter a potential critique that the protein pairs we identified may not actually be found close enough together in the human body for the biochemical binding we describe to occur among immune cells *in vivo*.

We do in the manuscript, however, include what we believe to be the most robust approach that is technically feasible for answering the question of whether the collection of novel interactions we report appear statistically more colocalized than would be expected by chance. These are the results displayed in figure 3d, which because they derive from spatial transcriptomics data where all receptors in the human genome are measured, it is possible to compare all of our novel interaction pairs against baselines of previously-known interacting receptor pairs and randomized receptor pairs. We describe this analysis in more detail in response to a subsequent reviewer question asking to clarify the interpretation of these results.

For the sake of completeness however, we nevertheless strove to conduct a quantitative analysis of the single-molecule RNA in situ hybridization images presented in Figure 3, to gauge what it would look like. We applied an existing CellProfiler method to segment the nuclei detected in that lymph node image, assign which cells each single-molecule RNA spot corresponds to, and then calculate colocalization (Figure R20).

Figure R20. Segmentation and automated assignment of single-molecule RNA spots to the human lymph node cells imaged in main figure 3e. The four fluorescence channels that were imaged are shown in the top row. The RNA images are masked to only show detected spots within an identified cell nucleus. Distinct identified nuclei from DAPI staining (far left) are colored in the bottom panel. Distinct identified immune cells with detectable pan-leukocyte *CD45* RNA within a cell nucleus are colored in the middle-left panel. *JAG1* and *VASN* RNA spot outlines (right) are colored based on their adherence to the segmentation criteria according to the CellProfiler defaults (version 4.2.1).

When quantified, this revealed that between 15-30% of *JAG1*-expressing or *VASN*-expressing immune cell nuclei were within a distance of 20 μm away from immune cell nuclei expressing the complementary protein transcript (Figure R21). While the exact values from that quantiation are not readily interpretable for the reasons described above, it does verify that, as was the result claimed for this figure panel, the two components of the interaction are sufficiently proximal that the interaction has the potential to take place *in vivo*.

Figure R21. Quantified distances between *JAG1*-expressing and *VASN*-expressing immune cells in main figure 3e support that the receptor-ligand pair is proximal *in situ*. Each histogram shows the distance between either a *VASN*-expressing cell (top) or *JAG1*-expressing cell (bottom) and the nearest cell expressing the complementary binding partner (*JAG1* or *VASN* respectively). Distances are measured between the centers of the cell nuclei, meaning if cells have an average diameter of ~10 μm then distances of 10 μm would represent cells whose edges are directly touching.

- Figure 2D caption reads 'Immune cell subsets utilize different distributions of binding affinities when communicating with other cell types.'; however, no statistical test of whether the distributions actually are different is offered. In fact, they seem remarkably similar. Also, the distributions are clearly log normal, thus median would be a more appropriate characteristic of the central tendency than average (i.e. mean) currently used in the color scale.

We fully concur with the reviewer's suggestions to improve this figure panel. First, we agree that not only are some of the differences in the binding affinity distributions noteworthy, but also it is noteworthy how the overall shapes of each distribution are, as the reviewer writes, "remarkably similar". To emphasize this better in the text, we have revised the caption to Figure 2d to read "Immune cell subsets utilize related but varying distributions of binding affinities when communicating with other cell types". Following the reviewer's prudent advice, we have also revised Main Figure 2d to display the median as the metric of each distribution's central tendency.

Revised main figure panel 2d. All colors have been recalculated based on median values instead of arithmetic mean values.

Next, we are thankful to the reviewer for pointing out how a statistical analysis would be valuable to test more rigorously whether the shifts in distributions between cell-cell pairs are significant. To correct this oversight, we have added a new table to the manuscript (Supplementary Table 4). To answer the question of whether these distributions differ, we conducted an omnibus statistical test that evaluates if the observed affinities of all cell-cell pairs derive from the same distribution (the null hypothesis) or differing distributions (the alternate hypothesis). Motivated by the reviewer’s astute observation about the log-normal characteristics of our data, we applied the Kruskal-Wallis H test, which is a robust non-parametric version of the common one-way analysis of variance F test. This verified that the distributions between cell pairs do indeed differ ($p \approx 2 \times 10^{-6}$). For the sake of completeness, we also include in the new Supplementary Table 4 the results from *post-hoc* Wilcoxon tests on each possible comparison following adjustment for multiple comparisons. We would like to emphasize that it can be difficult to interpret the biological significance of these kinds of “all-ways” comparisons, which is the reason why in our initial manuscript submission we instead selected a single set of cell-cell pairs

shown is the distribution of K_d 'for all receptors' in the respective states, while what is shown is merely the distribution of K_d 'of the receptors' down- and up-regulated on activation. This needs to be corrected by making the x-axis read 'Down-regulated' and 'Up-regulated', or similar. Are the affinities weighted here by the magnitude of change in expression level? If not, why?

These are excellent suggestions to clarify how the analysis in Figure 2E was performed. First, we have revised our manuscript to change the labels on the x-axis of the figure from “Resting” and “Active” to the more precise phrasing the reviewer suggested of “Upregulated” and “Downregulated”.

This change in the phrasing used for the axis labels has been applied both to the main Figure 2E and all the related panels in Extended Data Figure 6.

We also agree that we could better explain the methodology for how binding constants, which as the reviewer notes are properties of pairs of receptors, were mapped to individual differentially-expressed receptors. In the majority of cases, the receptors we identify as being significantly differentially expressed after immune activation participate in only a single binding interaction, and thus the K_D value that is plotted is simply the K_D of that receptor's sole binding interaction. For the minority of cases where a receptor participates in more than one binding interaction, we address this ambiguity by including separate data points for each of the receptor's interactions. To consider the example of the differentially expressed receptors in Figure 2E : 71% participate in only a single interaction, 91% have binding dissociation constants for all the interactions it participates in varying by no more than a factor of 10^1 , and every receptor except one (96%) have binding constants that vary by no more than 10^2 (Figure R22). Given the range of binding affinities across receptors is greater than 10^6 , we judged that this approach to map individual receptors to their binding pairs was sufficiently consistent to enable our analysis.

The wording on both the figure legend and methods section have been revised to clarify how interaction binding affinities were mapped to receptors.

Figure R22. Receptors differentially expressed upon immune activation predominantly have a single or closely-related interaction binding affinities. For all the receptors found to be differentially expressed in the analysis in main Figure 2e, one point is plotted for each binding interaction that the receptor participates in. Receptors with multiple interactions have boxplots overlaid to show the distribution of binding dissociation constants.

Lastly, in the manuscript we initially submitted, we had not weighted the data points by the magnitude of change upon differential expression. Although we understand the logic of putting greater emphasis on more highly upregulated or downregulated receptors, we felt that the standard approach we took avoids questions of how one would weight the interactions in a way that is not arbitrary or prone to bias. For instance, if we performed a weighted analysis, should the weighting be proportional to the fold-change in expression, the log-fold change, the t-statistic or p-value that also factors in the statistical confidence in the magnitude of change, a metric counting the absolute magnitude of change in protein copy number on the cell, or a combination of all these factors? The question of which statistical test or approach would be used for the final calculations would again be ambiguous given these tests are less standard.

For completeness, in this document we have explored one such weighted analysis to see how its results compare to our simpler unweighted approach. We repeated the analysis in Figure 2E using the *weights* package in R which incorporates weight values for each data point when calculating means and variances for a t-test. We considered the absolute value of the log-fold change to be our weighting criteria. The results were

similar to what we calculated with our more standard unweighted t-test approach (Figure R23). After careful consideration, we have decided to use our initial unweighted results to avoid the concerns mentioned above about how to select weights and tests in a way that is biologically meaningful and less prone to criticism of being based on arbitrary selections.

Figure R23. Reanalysis of the differential expression results in main Figure 2e by weighting each point according to the magnitude of differential expression returns similar results to the original unweighted analysis. Each point is scaled to be proportional to the absolute magnitude of the log₂ fold-change in expression between the indicated immune cell population when activated versus resting. Statistical test values are from weighted t-tests.

- Fig. 3D left panel shows JAG1-VASN expression, but the caption does not make any mention of this. More importantly, it seems to want to suggest that JAG1-VASN are expressed in neighboring cells more often than it would be by random, but no statistical test for this is provided, e.g. by permuting the identity of spots. Also, how was JAG1-VASN chosen? This is important to know as multiple hypothesis correction might be necessary to apply for the permutation test mentioned above given the many options.

We thank the reviewer for pointing out how the JAG1-VASN example image we show in Figure 3D did not have enough context provided in the current manuscript version. We have added a sentence to the figure legend to clarify that the tissue section image showing JAG1 and VASN mRNA spots is simply an example to illustrate how we can take spatial transcriptomics spots and calculate how often neighboring spots contain both members of an interaction pair, in order to produce the boxplot graph on the right. JAG1-VASN is not itself the subject of the boxplot but rather is a single data point of a novel interaction whose colocalization was measured.

We hope that this clarifies why we do not make claims specifically about JAG1-VASN as a single interaction, because this data source cannot reliably be used to answer this question for individual interactions. Instead, the boxplots illustrate how the spatial colocalization of protein pairs in our set of novel interactions (blue) is significantly greater than the colocalization of those same proteins with the pairings randomly permuted

(orange) and further is similar to the colocalization seen in already-established known interactions (green). Thus the claim we are making is not about the colocalization of JAG1-VASN specifically (or any other specific interaction). Rather it is that our set of novel interactions as a whole behaves as would be expected for known interactions already reported to occur *in situ* (i.e. ‘known’ boxplot comparison) and the pairs we identified by our binding screen are distinctly different in their colocalization than purely random pairings (‘random pairs’ boxplot comparison). Because we are not testing the many individual interactions but rather the groups of interactions as a whole, that also accounts for why large-scale multiple testing correction does not apply in this instance. Instead, for the 3 comparisons we made, we performed an ANOVA followed by post-hoc Welch tests, which controls for the family-wise error rate across the 3 comparisons done.

When picking a visual example to illustrate the method, we selected JAG1-VASN because it both 1) was not an outlier case unrepresentative of the larger set of novel interactions, and 2) had a sufficiently high colocalization score between neighboring spatial transcriptomics spots that the colocalization could be visible by a simple visual inspection without requiring numerical analysis to see. The individual colocalization scores of the novel interactions that make up the data points plotted in the blue boxplot of Figure 2D are shown below in Figure R24. As explained above, the measurement for JAG1-VASN was neither too small to be visually noticeable when compared to the colocalization of random pairs (dashed line) nor too large to be an outlier in the way the VSIR interactions would have been were they picked as the example.

Figure R24. Physical adjacency of spatial transcriptomics spots expressing the two proteins of an interaction pair for each novel interaction with RNA expression detected. For each protein pair, the length of the bar corresponds to the percent of the time both components of the pair were detected in physically-connected regions of tissue (e.g. same spot

or directly adjacent spots). A dashed line indicates the median value calculated when permuting proteins into random pairs.

- Fig. 3D right panel needs to include Prior+Novel. Is it any different to Prior only? If not, what is the message of the panel? Would it not be more appropriate to make a quantitative estimation of interaction strength between each of the spots using Prior links and Prior+Novel links and show whether this metric predicts colocalization significantly better if using also the Novel links?

We hope that the elaborations we have made to Figure 3D in response to the reviewer's prior comment will also help to clarify what was asked in this question. The message of the panel is two-fold. First, when you look at the spatial colocalization of the receptors we found to be interacting in our biochemical screen within the context of a human lymph node, the receptors we identified as binding do indeed colocalize with each other to a degree that is statistically indistinguishable to the colocalization seen between pairs of receptors previously shown to bind. Second, the colocalization of the receptors in our set of novel interaction pairs is significantly greater than the colocalization of those receptors in randomly permuted pairs.

This demonstrates that despite our interactions being discovered by biochemical binding assays on purified protein, we can nevertheless detect the receptor pairs found in our novel interactions as being significantly colocalized in primary lymphoid tissue. We would not expect these colocalizations to ever be close to 100% as leukocytes are highly migratory cells and the two cells of an interaction pair need only to colocalize briefly to initiate signaling. This is the reason why we compared to previously-known interactions, to know the baseline colocalization frequencies that would be expected for true receptor binding events already known to occur *in vivo*. The similarity between the 'known' and 'novel' distributions and divergence from the randomly permuted distribution gives us confidence that, on aggregate, the novel interactions we discovered have evidence for occurring *in vivo* within key immune sites such as the lymph node.

- For Fig. 3C, more statistical testing might be necessary. How was myeloid lineage chosen for this comparison? How does such graph look like e.g. for lymphoid vs. other? As a statistical test, given that it is unclear that the centrality values are normally distributed, what might a be more appropriate than a t-test would be a bootstrapped p value - taking all possible choices of four/five (depending on how many myeloid cell types were identified in that tissue) cell types out of all the cell types and asking what proportion of times the mean centrality comes higher than for the myeloid subset.

We appreciate the reviewer's comments about the analysis performed in Figure 3c, and can begin by describing the historical story of where the hypothesis tested in the figure originated. The hypothesis that myeloid cells may be central hubs in immune cell communication networks arose directly from viewing the outputs of the web tool we

created to visualize immune cell interactions in different human tissues. When performing simple visualizations such as the example shown in figure R25 which counts the number of receptor connections between different cell populations, it appeared myeloid cells had either more connections in total or more connections with highly-connected cell populations. These properties, in informal language, are the definition of a “hub” in a communication network. To test this hypothesis, we then conducted a formal analysis of the cell-cell networks as a graph to calculate the eigenvector centralities, which are the results shown in Figure 3c. Therefore, only a single hypothesis was tested, driven by our early observations. To avoid multiple comparison concerns, we have not calculated such graphs for other possible divisions of cells such as lymphoid cells.

Figure R25. Screenshots of outputs from our interactive website showing where the hypothesis myeloid cells may act as interaction hubs originated. Each grid shows the “Sum view” output of the website, where the total number of detected cell-to-cell receptor interactions between two cell populations (x and y axes) are counted and displayed as a heatmap. For consistency, all plots are shown at the website’s default threshold of 10% of cells expressing each protein in an interaction pair to binarize the interaction as ‘detected’. Myeloid cell populations have their names highlighted in orange.

We are grateful for the reviewer’s suggestion on performing an alternate bootstrapped statistical test to try to account for the potentially asymmetric distribution of values in the data. We implemented the test the reviewer described by comparing the mean centrality values of the N observed myeloid cell populations in a given tissue to randomly-selected subsets of N cells out of all the cell populations in that tissue. The p values calculated by this approach were similar to, or lower than, those we originally calculated by Welch’s t-tests (Figure R26).

Figure R26. Bootstrap testing supports the significance of the myeloid centrality results in main figure 3c. For each human tissue measured in main figure 3c, the null distribution of eigenvector centrality scores is shown when a random subset of cell types of equal size to the tissue's true myeloid populations were compared to all other immune cell types in the tissue. A vertical line is shown to denote the observed eigenvector centralities for the myeloid cells in the tissue, along with a p value for the proportion of random subsets as large or larger than the observed.

When performing this bootstrap analysis, we did notice one limitation: tissues varied greatly in the number of possible random combinations (ranging from 98,000 in the leukocyte-rich thymus to 153 in the lungs). The consequence of this is that for 2 of the 7 tissues, it was not possible to achieve the commonly-set minimum standard of 1,000 independent combinations to build the null distribution. To avoid this potential criticism of the approach and because the bootstrap results agreed with the earlier t-testing strategy and were, if anything, lower in p values than the t-tests, we have retained the t-testing p values for the figure.

- For Fig. 3A, from reading the main text, caption and Methods, it is unclear how the links in circos plots are derived, so this needs to be better stated. Are the circos plots weighted? Is it that the interaction strength was calculated using the presented model that integrates the ligand-receptor interaction strength and expression levels? If it was not weighted, why?

We thank the reviewer for pointing out that there were ambiguities in our description of what the links in the circos plot in Figure 3a represent. We have both revised the figure

caption for better clarify and have added two additional sentences to our methods section that explicitly specify the criteria we used for drawing linkages on the circos plot.

Currently no weighting is applied based on our model's calculation of interaction strength. We found the idea of weighting based on our model to be a fascinating suggestion, although there are three pragmatic barriers we encountered when attempting to implement this idea. The first of these is simply visual: since these plots must capture a large number of cell-cell interactions in a single view, the high density of lines would make differences in weights difficult to perceive by eye. Second, our mathematical model can account for the rates of physical associations between immune cells, but receptor interactions may have other important functional roles beyond this (such as initiating signaling). Weighting the linkages based on the model would make certain functionally important kinds of linkages (like PDL1 - PD1, colored lime green) appear minimized simply because their binding is too transient to drive leukocyte adhesion.

The third and final practical barrier we found is that the mathematical model is most suitable for when the values it receives as inputs represent true biophysical measurements, such as the absolute copy number of cell surface protein molecules identified by calibrated mass spectrometry (which is the input data type we currently use in all figures involving the model). Although conceivably one could extend this to RNA-seq data inputs, the biophysical equations in the model would lose their original well-defined interpretations and presumably become loose proportionality values to the true interaction strengths. The fidelity of that proportionality would depend heavily on the correlation of mRNA expression to protein levels and the single-cell protocol's mRNA capture efficiency. Both of these can be especially variable for transmembrane proteins, and thus we have limited ourselves to applying the model's predictions only in the most robust situations where the model's biophysical inputs (e.g. proteins, not mRNA proxies) have been directly measured.

Reviewer Reports on the Second Revision:

Referees' comments:

Referee #3 (Remarks to the Author):

The authors have addressed all raised points to my satisfaction.